

# Truth and Uncertainty. A critical discussion of the error concept versus the uncertainty concept

Thomas von Clarmann[1], Steven Compernolle[2], and Frank Hase[1]

[1]Karlsruhe Institute of Technology, Institute of Meteorology and Climate Research, Karlsruhe, Germany
[2]Department of Atmospheric Composition, Royal Belgian Institute for Space Aeronomy (BIRA-IASB), 1180 Brussels, Belgium

*Correspondence to:* T. von Clarmann (thomas.clarmann@kit.edu)

**Abstract.** Contrary to the statements put forward in "Evaluation of measurement data – Guide to the expression of uncertainty in measurement" (GUM08), issued by the Joint Committee for Guides in Metrology, the error concept and the uncertainty concept are the same. Arguments in favour of the contrary have been analyzed and were found not compelling. Neither was any evidence presented

in GUM08 that "errors" and "uncertainties" define a different relation between the measured and the true value of the variable of interest, nor does this document refer to a Bayesian account of uncertainty beyond the mere endorsement of a degree-of-belief-type conception of probability.

## 1    Introduction

For more than 200 years, error estimation used a more or less unified terminology where the

term 'error' was used, with some caveats, for designating a statistical estimate of the expected difference between the measured and the true value of a measurand (Gauss, 1809, 1816; Pearson, 1920; Fisher, 1925; Rodgers, 1990; Mayo, 1996; Rodgers, 2000, just to name a few). More recently, the Joint Committee for Guides in Metrology (JCGM), on request of the Bureau International de Poids et Mesures (BIPM) presented a contrasting definition how we have to conceive

the term 'error' and have stipulated a new terminology, where the term 'measurement uncertainty' is used in situations where one would have said 'measurement error' before (Joint Committee for Guides in Metrology (JCGM), 2008, this source is henceforth referenced as GUM08). Supplementary material in the context of GUM is found in Joint Committee for Guides in Metrology (JCGM) (2012) and several supplements to GUM08, that are found on the BIPM website

(https://www.bipm.org/en/publications/guides/gum.html). The new concept has been critically discussed by, e.g., Bich (2012), Grégis (2015), Elster et al. (2013) and The European Centre for Mathematics and Statistics in Metrology (2019), and more favorably by, e.g., Kacker et al. (2007). The claim is made that the uncertainty concept can be construed without reference to the unknown and unknowable true value while the error concept cannot (GUM08, p.3 and p. 5). Thus, the dispute be-





tween the error statisticians and the uncertainty statisticians comes down to the question if and how the error (or uncertainty) distribution is related to the true value of the measurand. In this paper we try to shed some light upon this relation which seems to have caused a rift both in the communities of statistics and of empirical sciences. Further, we critically discuss the applicability of the GUM08 recommendations in the context of remote sensing of the atmosphere. Remote sounding employs in-

direct measurements where the measurand is not measured directly but retrieved from the measured signal by the inverse solution of the radiative transfer equation which provides the link between the measurand and the measured signal. In the context of the work undertaken by the activity 'Towards Unified Error Reporting, (TUNER),' a project aiming at unification of error reporting of satellite data (von Clarmann et al., 2020)), this issue is particularly problematic. Without agreement on the

concepts and the terminology of error versus uncertainty assessment, any unification is out of reach.

At the outset we recapitulate the concept of indirect measurements and lay down an appropriate terminology and notation (Section 2). In the subsequent section (Section 3), we analyze the use of the term 'error' by the uncertainty statisticians[1] and will find that it is often not consistent with the use of this term as originally used by the error statisticians. Then we try to find out what the exact

connotation of the term 'uncertainty' is and how it is actually distinguished from the traditional concept of error analysis (Section 4). We shall find that the concept of the 'true value of the measurand' makes up the alleged key difference. That is to say, the uncertainty concept is claimed, contrary to traditional error analysis, to be able to dispense with the concept of the true value. The problem of the true value is that it is neither known nor knowable. In Section 5 we discuss how this affects error

and uncertainty estimation and the relation between the measured and the true value. We find (1) that according to Bayesian statistics (Bayes, 1763) the measured value cannot always be interpreted as the most probable value of the measurand. (2) We further find that nonlinear relationships between the measurand and the measured signal poses problems to the uncertainty analysis because a value in sufficient proximity to the true value should be chosen as linearization point for uncertainty es-

timation and thus must be – at least approximately – known; and (3) we accept that we can never know for certain if the error or uncertainty budget is complete. In the following we investigate the implications of these three problems in turn. First, we investigate under which conditions an error or uncertainty distribution can be understood as a distribution which tells us which likelihood or probability we can assign to a value to be the true value, given a certain measured value (Section

5.2). We shall see that the interpretation of resulting error or uncertainty estimates is completely different in a maximum likelihood versus a Bayesian framework. Second we assess to which degree

---

[1]We use the term 'uncertainty concept' for a concept where it is claimed that error and uncertainty are different entities and that 'uncertainty' can be defined without reference to the true value of the measurand. We use the term 'uncertainty scientists' or 'uncertainty statisticians' for scientists endorsing the uncertainty analysis concept. Conversely, we use the term 'error analysis concept' for a concept which denies a fundamental difference between the traditional concept of error analysis and the uncertainty concept as endorsed by GUM08', and we call 'error scientists' or 'error statisticians' those scientists who endorse the traditional concept of error analysis.





the nonlinearity of the relationship between the measured signal and the target quantity, *viz.*, the radiative transfer equation, poses additional problems (Section 5.3). And third we scrutinize the claim that there will always be unknown sources of uncertainty and that it is thus impossible to relate the

measured value along with its uncertainty estimate to the true value (Section 5.4). After these more general considerations we critically discuss the applicability of the GUM08 concept to indirect measurements of atmospheric state variables (Section 6). There we discuss the problems of measurands that are not well-defined in the sense of GUM08 (Section 6.1), if it is really adequate to report the combined error only (Section 6.2) and if the measurement equation as presented in GUM08 does

optimally support the uncertainty assessment in atmospheric remote sensing (Section 6.3) In this context, we also investigate whether the difference between the traditional concept of error analysis and the uncertainty concept might be linked to a Bayesian versus a frequentist conception of probability. Finally (Section 7) we conclude to which degree the arguments put forward by the Joint Committee for Guides in Metrology (JCGM) are conclusive and what the differences between the

error concept and the uncertainty concept actually are.

## 2   Recapitulation of the concept of indirect measurements

In the case of indirect measurements, e.g., remote sensing, the measurand, i.e., the quantity of interest, $x$, and the measured signal $y$ are linked via a function $f$ as

$$y = f(x; b) + \epsilon, \tag{1}$$

where $b$ represents the parameters of $f$ representing physical side conditions and $\epsilon$ is the actual measurement error in the $y$-domain (Rodgers, 2000). In the case of remote sensing of the atmosphere $f$ is the radiative transfer function. We use vector notation because in remote sensing typically multiple measurands are estimated from multiple measurement signals. For example, $y$ could be a spectral measurement of an ozone emission line in the infrared; $x$ could be a vertical profile of

ozone concentrations, and $b$ could include a vertical profile of temperature, known a priori with some uncertainty and affecting the signal in the ozone line. To obtain information on the measurand, some kind of inverse solution of Equation 1 is required, because the estimate of the target quantity $x$ involves a conclusion from the effect to the cause. More often than not, this inverse problem is ill-posed in the sense of Hadamard (1902), and the direct inversion is impossible and some kind of

workaround is employed. Candidate workarounds are least-squares solutions, regularized solutions and so forth. von Clarmann et al. (2020) summarize the most common methods to solve this kind of problem, along with related error estimation schemes. Here we call this substitute for the genuine inversion $\tilde{F}^{-1}$. Here $F$ is a function representing the true radiative transfer function to the best of our knowledge, i.e., the descriptive radiative transfer law as opposed to the governing but unknown law.

The ˜ symbol reminds us that the inversion is not necessarily a genuine inversion in a mathematical





sense. With this an estimate of the measurand can be obtained[2]:

$$\hat{\boldsymbol{x}} = \tilde{\boldsymbol{F}}^{-1}(\boldsymbol{y}; \boldsymbol{b}) \tag{2}$$

Differences between the estimate[3] $\hat{\boldsymbol{x}}$ and the true value of the measurand $\boldsymbol{x}$ can be due to

1. measurement errors representd by $\epsilon$,

2. erroneous assumptions on the values of the parameter vector $\boldsymbol{b}$,

3. differences between the radiative transfer model $F$ and the true but not exactly known radiative transfer physics $\boldsymbol{f}$, and

4. characteristics of $\tilde{\boldsymbol{F}}^{-1}$, i.e., tricks applied to make a non-invertible inverse problem solvable.

These error sources and recipes to estimate related error components of $\hat{\boldsymbol{x}}$ are discussed in depth in
von Clarmann et al. (2020), drawing upon Rodgers (2000) and Rodgers (1990).

Some complication arises because the true atmospheric state can be represented only by spatially continuous functions while we work with vectors of finite dimension. Here we avoid related difficulties by assuming that the measurand $\boldsymbol{x}$ represents a discretized atmosphere, i.e., it does not represent a point value but some kind of spatio-temporal average.

All this holds *mutatis mutandis* also for scalar quantities where

$$y = f(x) + \epsilon \tag{3}$$

but this distinction has no bearing upon our argument.

When reading the thermometer, we actually read the length of the mercury column, apply inversely the law of thermal expansion, and get an estimate of the temperature:

$$\hat{x} = \tilde{F}^{-1}(y; b) \tag{4}$$

Only in trivial cases, when a measurement device has a calibrated scale from which $\hat{x}$ can be directly read, $F$ is unity. Here the inverse process is effectively pretabulated in the scale. In any case, the availability of $\tilde{F}^{-1}$ allows to statistically estimate the effect of measurement noise $\epsilon$ and systematic effects in $\boldsymbol{b}$ or $\tilde{F}^{-1}$ on $\hat{\boldsymbol{x}}$ (Rodgers, 1990, 2000; von Clarmann et al., 2020).

---

[2]Strictly speaking, it would be adequate to assign a different symbol to the vector of parameters $\boldsymbol{b}$ when it appears in the context of $\boldsymbol{f}$, where it designates the true parameters, and in the context of $\boldsymbol{F}$, where it represents estimates or uncertain a priori knowledge about these parameters. However, we assume that it is clear from the context what is meant.

[3]In the context of direct measurements, 'estimated value' and 'measured value' connote the same thing; in the context of indirect measurements, we use the term 'estimated value' for the value of the target quantity that was inferred from the measured signal.



## 3 The connotation of the term 'error'

Although GUM08 (Sect. 0.2) claims the "concept of uncertainty as a quantifiable attribute to be a relatively new concept in the history of measurement", we uphold the view that it has long been recognized that the result of a measurement remains to some degree uncertain even when a thorough measurement procedure and error evaluation is performed. We recall that even ancient researchers realized that measurement results always have errors. Carl Friedrich (Gauss, 1809) and Adrien-Marie Legendre (1805) formalized the required procedure of balancing imperfect astrometric measurements by least squares fitting, in support of orbital calculations from overdetermined data sets. And there is no reason to believe that earlier investigators were unaware of the fact that they were not working on perfect observational data. Kepler's conclusion concerning the elliptical shape of the orbit of Mars based on the rich observational dataset collected by Brahe would have been impossible without proper implicit assumptions concerning the limited validity of the reported values (Kepler, 1609).

A rich methodical toolbox for error estimation and uncertainty assessment has been developed since then, and a confusing plethora of conventions how to communicate measurement uncertainties exists. Unification of error or uncertainty reporting is overdue but this requires, at a minimum, agreement on terminology and the underlying concepts. While many of the technical terms used are quite clear and often self-explanatory, there is a particularly troublesome terminological issue. It is related to the use of the term 'error' and the underlying concept. According to the Joint Committee for Guides in Metrology (JCGM), the concept of uncertainty analysis should replace the concept of error analysis. Thus, some conceptual and terminological remarks seem appropriate. While on the face of it, this is quibbling about words, actually conceptual differences between the errors and uncertainties are claimed to exist. This issue is discussed in the following.

In the context of measurements, the term 'error' traditionally has two slightly different connotations. The first is the actual difference between the measured or, in the context of indirect measurements, retrieved, value and the true value of the measurand; the second meaning of the term 'error' is a statistical estimate of this difference. Often some attributes are used for clarification and specification, e.g., 'probable error' (Gauss, 1816; Bich, 2012), 'statistical error' Nuzzo (2014), 'error estimation' (Zhang et al., 2010) or 'error analysis' (Rodgers, 1990, 2000; Hughes and Hase, 2010). GUM08 (Annex B.2.19) allows only the connotation 'actual difference', but their use of the terms 'error and error analysis' in the first sentence of their 0.2 or 'possible error' in their 2.2.4 only make sense if the statistical meaning of the term 'error' is conceded. The authors of this paper are not aware of any case where the ambiguity of these connotations of the term 'error' has ever led to any misunderstanding, most probably because it is always clear from the context what is meant.

In the case of 'error', its statistical estimate is mostly understood to be a quadratic estimate and thus does not carry any information about the sign of the error. This entails that the true or most probable value cannot simply be determined by subtracting the estimated error from the measured





value. One of the first major documents, where the term '*error*' has been used with this statistical connotation is, to the best of our knowledge, "Theoria Motus Corporum Celestium" by C. G. Gauss (1809). Since then, the term 'error' has commonly been used to signify a statistical estimate of the

size of the difference between the measured and the true value of the measurand. Seminal books such as "Statistical Methods For Research Workers" by R. A. Fisher (1925) or "Inverse Methods For Atmospheric Sounding - Theory and Practice" by C. D. Rodgers (2000) furnish evidence of this claim about the use of this term. The estimated error is understood as a measure of the width of a distribution around the measured (or estimated) value which tells the data user the probability – or the

likelihood, depending on the statistical framework used – density of a certain value to be measured or estimated if the value actually measured or estimated was the true value. Counterintuitively, in general it does not always provide the probability density that the measured value is the true value. This issue will be discussed in Section 5.2.

     In some cases, uncertainty scientists repudiate the use of the term 'error' to refer to a statistical

estimate. Instead, they claim that the term 'error' only connotes the actual difference between the measured or estimated and the true value of the measurand. E.g., on page 5 in GUM08, the error of a measurent is the "result of a measurement minus a true value of the measurand" (GUM08, p. 5). It may be challenged that this definition is fully adequate, because it suppresses the use of the term 'error' for the statistical estimate of the actual error in the scientific literature since Gauss

(1809) who uses the terms '*error*' and '*incertitudo*' (latin for error and uncertainty, respectively) broadly as synonyms. The only difference, if any, in Gauss' use of these terms seems to be that error refers to estimated values, and uncertainty refers to the true values, saying that they are not known with certainty. In spite of the explicit definition in GUM08, there is no unified stance among GUM08 endorsers as to what 'error' is. E.g., Merchant et al. (2017) uphold that 'error' connotes

only the absolute difference, while Kacker et al. (2007) or White (2016) refer to 'error' as a statistical estimate. Kacker et al. (2007) complain that GUM08 is often misunderstood, and we suspect that the cause for this might be that GUM08 is indeed not sufficiently clear with respect to the differences between the error concept and the uncertainty concept.

     Since the true value is not known, the actual difference between the measured or estimated value

and the true value of the measurand cannot be calculated. This argument is often used to dispraise the traditional concept of error analysis as an obsolete concept. However, history and existing literature tell a different story. Above we have listed numerous sources where the term 'error' connotes a statistical quantity which can be estimated without knowledge of the true value of the measurand.

     The issue of whether the term 'error' should be used also for a statistical estimate cannot be judged

on scientific grounds. It is a matter of stipulation, although in the main body of GUM08 this stipulation is presented as if it was a factual statement ("In this Guide, great care is taken to distinguish between the terms 'error' and 'uncertainty'. They are not synonyms, but represent completely different concepts; they should not be confused with one another or misused.", Sect 3.2, Note 2). The


synonymity of 'error' and 'uncertainty' is thus neither true nor false but adequate or inadequate.
Instead of quibbling about words we will concentrate on the concepts behind these terms.

GUM08 does not only present traditional error analysis in a revised language but suggests that there is more to it. That is to say, the entire concept is claimed to be replaced. We understand that GUM08 grants that the classical concept of error analysis deals with statistical quantities, but these are statistical estimates of the difference between the measured or estimated value and the true value. We take GUM to be saying that the reference of even this statistical quantity to the true value poses certain problems, because the true value is unknown and unknowable. As a solution of this problem, the uncertainty concept is introduced which allegedly makes no reference to the true value of the measurand and is thus hoped to avoid related problems. GUM08 (particularly Section 2.2.4) unfortunately leaves room for multiple interpretations, but our reading is that an error distribution is understood as a distribution whose spread is the estimated statistical error and whose expectation value is the true value, while an uncertainty distribution is understood as a distribution whose spread is the estimated uncertainty and whose expectation value is the measured or estimated value.

GUM08 (p.5) characterizes error as "an idealized concept" and states that "errors cannot be known exactly". This is certainly true but it has never been claimed that errors can be known exactly. Since not all relevant error sources are necessarily known, any error estimate remains fallible but still it is and has always been the goal of error analysis to provide error estimates as realistic as possible. To use the statistical conception of 'error' and conceding the fallibility of its estimated value, it is not necessary to know the true value. It is only necessary to know the chief mechanisms which can make the measured value deviate from the true value and to have estimates available on the uncertainties of the input values to these mechanisms.

Some GUM08 endorsers (e.g., Kacker et al., 2007) try to draw a borderline between error analysis and uncertainty assessment in a way that they associate error analysis with frequentist statistics while uncertainty is placed in the context of Bayesian statistics. Frequentist statistics, we understand, is a concept where the term 'probability' is defined via the limit of frequencies for a sample size approaching infinity. This definition is untenable because it involves a circularity: It is based on the large number theorem, according to which a frequency distribution will almost certainly converge towards its limit. This limit is then associated with the probability. 'Almost certainly' means 'with probability 1'. The circularity is given by the fact that the *definiendum* appears in the *definiens* (See, e.g., Stegmüller, 1973, pp. 27). We concede that many estimators in error estimation rely on frequency distributions. It is, however, a serious misconception to conclude from this that error analysis is based on a frequentist definition of 'probability'. This is simply a *non sequitur*. Frequency-based estimators are consistent with any of the established definitions of probability, and their use does not allow any conclusion on the definition of 'probability' in use.



## 4   The definition of the term 'uncertainty'

According to GUM08, p.2, the uncertainty of a measurement is defined as "a parameter, associated with the result of a measurement, that characterizes the dispersion of the values that could reasonably be attributed to the measurand". GUM08 claims that this context does not make reference to the 'true' value which is unknown anyway and that the uncertainty concept is more adequate because there can always exist unknown error sources which entail that an error budget can never be guaranteed to be complete (GUM08 p. viii). It is stated that the uncertainty concept is not inconsistent with

the error concept [GUM08 p. 2/3]. There are, however, certain inconsistencies and shortcomings, which are discussed in the following.

One of the major purposes of making scientific observations, besides triggering ideas on possible relations between quantities, is to test predictions based on theories on the real world (Popper,

1935). To decide if an observation corroborates or refutes a hypothesis, it is necessary to have an estimate how well the observation represents the true state, because it must be decided how well any discrepancy between the prediction and the observation can be explained by the observational error (e.g., Mayo, 1996). Any concept of uncertainty which is not related to the true state cannot serve this purpose.

On page 3, GUM08 says that the attribute 'true' is intentionally not used within the uncertainty concept because truth is not knowable. In GUM08, p. 59 it is claimed that the uncertainty concept "uncouple the often confusing connection between uncertainty and the unknowable quantities "true" value and "error". The term 'measurand' in their definition, however, is defined as the quantity intended to be measured (Joint Committee for Guides in Metrology (JCGM), 2009) (henceforth

abbreviated GUM09); GUM08, (p.32) says basically the same; GUM09, p. 20, says that the 'quantity' is the same as the 'true quantity value'. Inserting this definition in the GUM08 definition of uncertainty yields that, through the back door, uncertainty still refers to the true value. Thus it is not clear what the difference between the traditional concept of error analysis and the uncertainty concept is. Further, it is stated that systematic effects can contribute to the uncertainty. GUM08 falls

short of clarifying how a systematic effect be understood other than a systematic deviation between the measurement and the true value, the concept GUM08 apparently tries to avoid. In order to justify the attribution of an uncertainty distribution to the systematic effects without relying on frequentist statistics, they invoke the concept of subjective probability. With this it becomes possible to assign an uncertainty distribution to the combined random and systematic uncertainty but still it is not clear

how the systematic effect is defined without reference to the unknown truth.

Subjective probability reflects the personal degree of belief (GUM08, p. 39). Thus, a knowledge-dependent concept of probability is used in GUM08. As discussed in the previous paragraph, this approach has been chosen to allow the treatment of systematic errors as dispersions, although the systematic error does not vary and cannot thus be characterized by a distribution in a frequentist

sense (GUM08 p. 60). If we construe 'estimated error' and 'estimated value' as parameters of a





distribution assigning to each possible value the probability (in a Bayesian context) or the likelihood (in a maximum likelihood context[4]) that it is the true value, no knowledge of the true value is required. This is because, by definition, the subjective probability distribution merely represents the knowledge of the person generating it. In the context of subjective probability it is not clear why the true error or the true value should be needed to generate an error distribution. The values the rational agent believes to be true are sufficient in this case, because the error distribution does not tell us anything about the truth anyway but only about the agent's believe of what truth is. There is nothing wrong with the subjectivist concept of probability, nor do we attack the possibility to combine random and systematic errors in a single distribution. This concept, however, makes the knowledge of the true value and the true error unnecessary, and still the estimated error can be conceived as a statistical estimate of the absolute difference between the measured value and the true value. We consider it untenable and inconsistent to refer to the concept of subjective probability when it comes in handy and to deny it when it would solve the conflict between the error and the uncertainty concepts. Our skepticism about the possibility of dispensing with the concept of the true value is shared by, e.g., Ehrlich (2014), Grégis (2015), and Mari and Giordani (2014). Note that in the International Vocabulary of Metrology (known as VIM) (Joint Committee for Guides in Metrology (JCGM), 2012), although also issued by the JCGM the concept and definition of the true value are explicitly retained.

The use of the term 'uncertainty' in GUM08 seems inconsistent: The general GUM08 concept seems to be that the 'error' has to include all error sources and thus cannot be known; 'uncertainty' is weaker, it is only an estimate of quantifiable errors, excluding the unknown components. This view is supported by the following quotation (GUM08, p. viii) "It is now widely recognized that, when all of the known or suspected components of error have been evaluated and the appropriate corrections have been applied, there still remains an uncertainty about the correctness of the stated result, that is, a doubt about how well the result of the measurement represents the value of the quantity being measured." It is not fully clear what this means. One possible reading is that they use the term 'error' in the redefined sense, viz., as a quantity which measures the actual deviation from the true value. Then this statement would be a mere truism, just saying that after all correction and calibration activities there is still a need for error (in the error concept terminology) estimation. The only other possible reading is that they want to say that, since due to unknown (unrecognized and/or recognized but not quantified[5]) error sources, error estimation will always be incomplete and there remains an additional uncertainty not covered by the error estimation. This often is very true but the use of the term 'uncertainty' would then be inconsistent in their document, because here the connotation of 'uncertainty' is the unknown (unquantified or even unrecognized) part of the error, which by definition cannot be assessed, while in the main part of their document, the

---

[4]see Section 5.2 for a deeper discussion of this issue.

[5]Rigorously speaking, within the concept of subjective probability recognized but unquantified uncertainties should not exist.



connotation of 'uncertainty' seems to be a quantified statistical estimate. In summary, it is not clear if the 'uncertainty' includes the unknown error terms or not.

In GUM08, p. 2/3 it is claimed that the concept of uncertainty "is not inconsistent with other concepts of uncertainty of measurement, such as a measure of the possible error in the estimated value of the measurand as provided by the result of a measurement [or] an estimate characterizing the range

of values within which the true value of the measurand lies[6] (VIM:1984 definition 3.09). Although these two traditional concepts are valid as ideals, they focus on *unknowable* quantities: the "error" of the result of a measurement and the "true value" of the measurand (in contrast to the estimated value), respectively. Nevertheless, whichever *concept* of uncertainty is adopted, an uncertainty component is always *evaluated* using the same data and related information...". It remains unclear how

the concepts can, on the one hand, be consistent, while, on the other hand, it is claimed that the error approach and the uncertainty approach are actually conceptually different and not only with respect to terminology. In GUM08, p.5 it reads "In this Guide, great care is taken to distinguish between the terms "error" and "uncertainty". They are not synonyms, but represent completely different concepts; they should not be confused with one another or misused" Since both concepts, however, are

consistent, it is not clear in what the difference of the concepts consists.

Again, we come back to Kacker et al. (2007) who claim that error estimation and uncertainty analysis are best distinguished in the sense that the former relies on frequentist statistics while the latter is founded on Bayesian statistics. Here the following remarks are in order: (1) Many of the methods presented in GUM08, including their 'Type A evaluation (of uncertainty)', which is the

'method of evaluation of uncertainty by the statistical analysis of series of observations' are from the frequentist toolbox. (2) Kacker et al. (2007) endorse Monte Carlo methods to estimate uncertainty. Monte Carlo uncertainty estimation, however, is in its heart a frequentist method, because it estimates the uncertainty from the frequency distribution of the Monte Carlo samples. And (3) it is astonishing why GUM08, if representing a Bayesian concept, does not in the first place require

to apply the Bayes theorem to convert the likelihood distributions into a posteriori probability distributions. The methodology proposed in GUM is uncertainty propagation. This is a mere forward (or direct) problem: given that $x_{\text{true}}$ is the true value, and a measurement procedure with some error distribution, it returns a probability distribution for values $x_{\text{measured}}$ that might be measured. However, GUM08's definition of uncertainty "parameter, associated with *the result of a measurement,*

*that characterizes the dispersion of the values that could reasonably be attributed to the measurand*" (emphasis added by us), seems associated with another meaning: given a measured value $x_{\text{measured}}$ ("result of a measurement") and a measurement procedure with some error distribution, what is the probability distribution $x_{\text{true}}$ ("values that could reasonably be attributed to the measurand"). This is an inverse problem, for which Bayes theorem is applicable rather than uncertainty propagation.

---

[6]It is not clear how this can be achieved without explicit consideration of the Bayes theorem.



Interestingly enough, early documents of the history of GUM (Kaarls, 1980; Bureau International des Poids et Mésures) provide evidence that the terminological turn from 'error' to 'uncertainty' was triggered only by linguistic arguments, based upon the fact that in common language the term 'uncertainty' is often associated with "doubt, vagueness, indeterminacy, ignorance, imperfect knowledge". These early documents provide no evidence that 'error' and 'uncertainty' were conceived as

two different technical terms connotating different concepts. Any re-interpretation of the terms 'error' and 'uncertainty' as frequentist versus Bayesian terms or operational versus idealistic concepts came later.

In summary, it appears that the uncertainty concept is not essentially different from the error concept. We do, however, not claim that the terms 'error' and 'uncertainty' are fully equivalent;

even in pre-GUM language there might be some subtle linguistic differences. We perfectly know our measurement (even if it is erroneous) and are ignorant with respect to and have imperfect knowledge only about the true value. This suggests that the uncertainty is an attribute of the true value while the error is associated with a measurement or an estimate. Because of the measurement error there is an uncertainty as to what the true value is. The uncertainty thus describes the degree of ignorance

about the true value while the estimated error describes to which degree the measurement is thought to deviate from the true value. In this use of language both terms still relate to the same concept. This notion seems, as far as we can judge, to be consistent with the language widely used in the pre-GUM literature, but this issue deserves a more thorough linguistic assessment that is beyond the scope of this paper. The introduction of the term 'uncertainty of measurement' seems to us a mere linguistic

revision of an established terminology which does not connect to any further insights.

In summary, we have to distinguish between two questions; first, whether the terms 'error' and 'uncertainty' have the same connotation, and second, whether the underlying concepts are indeed different. The first question is contingent upon the underlying stipulation, and any statement about their equivalence or difference without reference to a definition is a futile pseudo-statement. The an-

swer to the second question is less trivial and deserves some scientific discussion. The main question still seems to be how the true value, the error or uncertainty, and the measured value are related with each other. This question will be addressed in the following sections.

## 5   The unknown true value of the measurand

The alleged key problem of the error concept is, in our reading of GUM08, that the value of the true

value of the measurand is not known, and that this true value must appear neither in the definition of any term nor in the recipes to estimate it. To better understand this key problem, we decompose it into four sub-problems.

1. Quantities of which the value cannot determined must not appear in definitions.


2. The error distribution must not be conceived as a probability density distribution of a value to
   be the true value.

3. Nonlinearity issues pose problems on error estimation.

4. On can never know that the uncertainty budget is complete because it can always happen
   that a certain source of uncertainty has been overlooked; thus, the full error estimate is an
   unachievable ideal.

Some of these sub-problems are in some way formulated in GUM08 but it is not exactly specified
there why the fact that the true value of the measurand is unknowable poses a problem to the er-
ror scientist. Others have been formulated by us, serving, as arguments of the Devil's advocate, as
working hypotheses in order to moot the error and uncertainty concepts in the context of indirect
measurements. In the following we will scrutinize these theses one after the other.

**5.1   The operational definition**

GUM08 tries to avoid to use the true value of the measurand in the definition of the term 'un-
certainty'. This strategy is employed because the true value of the measurand is "not knowable"
(GUM08, p. 3). It may be found puzzling why it should be necessary to know the value of a quantity
to use it in the definition of a term. The weight of Thomas Bayes or the body height of David Hume
at a certain time are well-defined quantities although we have no chance to measure them today. Also
we might have a clear physical conception of what the temperature in the center of the sun might
be although it may not be practicable to put a thermometer there, and we even might not be able to
figure out any other, more sophisticated, method to assign an accurate observation-based value to
this quantity. Intuitively, we conceive the definition of a quantity and the assignment of the value to
a quantity as quite different things.

In GUM08 it is claimed that the definition of 'uncertainty' is an operational one (p. 2). An op-
erational definition defines a quantity by stipulating a procedure by which a value is assigned to
this quantity. The concept of operational definitions was suggested by Bridgeman (1927) in order to
give terms in science a clear-cut meaning. This operationalism, at least a narrow conception of it,
has its own problems, has received considerable criticism and has led to deep philosophical discus-
sions (see, e.g., Chang, 2019). To summarize these is beyond the scope of this paper and for here it
must suffice to mention that there are alternatives, such as theoretical definitions or reduction of the
*definiendum* to previously defined terms.

GUM08's claim that the uncertainty concept is based on an operational definition leads to two
further inconsistencies. First, no unambiguous operation is stipulated on which the definition can be
based, but multiple operations are proposed, which might give different uncertainty estimates. Thus,
the definition is void. Our critical attitude with respect to operationalism in the context of GUM is
shared, e.g., by MARI14.



The other problem with the operational definition is the following: In GUM08, pp 2-3, it is claimed

that the uncertainty concept is not inconsistent with the error concept, and a few lines later it reads "an uncertainty component is always *evaluated* using the same data and related information." The latter suggests that within the error concept the same operations are used as within the uncertainty concept. Since the operations define the term and the related concept, the uncertainty concept and the error concept must be the same.

In summary, the fact that the true value of the measurand is unknowable is a problem for the definition of the term 'error' and its statistical estimates only if we commit ourselves to the doctrine of that only operational definitions must be used. If we abandon this dogma, there is nothing wrong with conceiving the estimated error as a statistical estimate between the measured or estimated and the true value, and the problem is restricted to the assignment of a value to this quantity. Related

issues are investigated in the following.

### 5.2    Likelihood, probability, and the base rate fallacy

The argument discussed in the following is not put forward in GUM08, probably because there indirect measurements are not in the focus. However, since we do focus on indirect measurements, and since this argument is of particular importance in this context, we present and discuss this argument.

Counterintuitively, in general, the error bar or the uncertainty estimate do not tell the probability density of the true value with respect to the measured value, unless the prior probability of the measurand has been considered in the retrieval. It is the theorem of Bayes (1763) which makes the difference. The non-consideration of the Bayes theorem goes under the name of 'base rate fallacy'. 50% of people suffering Covid-19 have fever (Robert Koch Institut, 2020), but this does not imply

that the probability that a person suffering fever to have Covid-19 is 50%. To estimate the latter probability requires knowledge of the percentage of people being infected with the Corona virus, and the probability that a person suffers fever for any reason. In metrology the situation is quite analogous. If we define the true value to be $x$, the ideally measured value $f(x) = y_{\text{ideal}}$, and the estimated measurement error in terms of the standard deviation $\sigma_y$, then the probability density of a

certain value $y$ to be measured is given by a pdf with $y_{\text{ideal}}$ mean and $\sigma_y$ spread. This, however, does not imply that, if we measure $y$ with an uncertainty of $\sigma_y$, and propagate $\sigma_y$ through the inversion procedure to get the uncertainty of $\hat{x}$, namely, $\sigma_x$, that the probability of some $x$ being the true value of the measurand is given by the pdf with mean $\hat{x}$ and $\sigma_x$ spread. Again, it is the a priori probability distribution[7] which is missing. There are three ways out of this problem. For now we will defer

the problem of a possibly incomplete error budget to Section 5.4 and assume that the error buget is complete.

The first solution is to apply a Bayesian retrieval scheme. Indeed in many cases, the solution of the inverse problem $\tilde{F}^{-1}$ employs a Bayesian estimator. Examples are found, e.g., in Rodgers (2000) or

---

[7]or the background frequency distribution.



von Clarmann et al. (2020). On the supposition that the error budget is complete, the interpretation

of the error bar as the spread of a distribution representing the probability density that a certain value

is the true value is correct.

The second solution is the application of the principle of indifference. Often there is no firm a
priori knowledge on the value of the measurand available. Gauss (1809) solves this problem by
application of the indifference principle. That is, the same a priori probability is assigned to all

possible values of the measurand. With this, e.g., in the application to a linear inverse problem and
normal distributions of uncertainties, the Bayesian solution collapses back to a simple unconstrained
least squares solution. Due to the assumption of the equidistribution of the a priori probabilities the
estimated uncertainty of the estimate can still be interpreted as the width of the probability density
function of the true value of the measurand. This concept of 'non-informative a priori', however,

has its own problems. Even if we ignore some more trivial problems for the moment, e.g., that
some quantities cannot, by definition, take negative values, this concept can lead to absurdities:
If we assume that we have no knowledge on, say, the volume density of small-particle aerosols
in the atmosphere, and describe this missing knowledge by an equidistribution of probabilities, this
would correspond to a non-equidistribution of the surface densities, due to the non-linear relationship

between surface and volume. It strikes us as absurd that information can be generated just by such
a simple transformation from one domain into another. The principle of indifference, upon which
the concept of non-informative priors is built, is critically discussed, e.g., by Keynes (1921, Chapter
IV). The concept of non-informative priors is still criticized even in the Bayesian community (e.g.
D'Agostini, 2003).

The third solution is the likelihood interpretation, which has been introduced by Fisher (1922). The
likelihood that the true value is $x$ if the measured signal is $y$ equals the probability that $y$ is measured
if the true value is $x$. No prior information is considered. Solution of the inverse problem $\tilde{F}^{-1}$ by
maximizing the likelihood of $x$ does not provide the most probable estimate of $x$, and accordingly
the error bar of the solution must not be interpreted as the width of a probability distribution of

the true value. Application to a linear inverse problem and normal distributions of uncertainties
renders formally the same estimator as the Gaussian least squares solution, but its interpretation has
changed. It can no longer be interpreted as the mean of a pdf of the true value. The distribution
with mean $\hat{x}$ and the standard deviation $\sigma_x$ can still be interpreted as a likelihood distribution of the
true value around the estimate. If need be, in some cases, i.e., if the inverse problem is well-posed

enough to allow an unconstrained solution, the maximum likelihood estimate can be, *post factum*,
transformed into a Bayesian estimate by application of the Bayes theorem. Certainly one might
urge the objection that the true a priori distribution is never known and that this transformation
is thus an idealized concept. This argument is only applicable by the frequentist statistician. The
uncertainty statisticians, however, have already committed themselves to the concept of subjective

probability because otherwise the aggregarion of random uncertainties and systematic effects into





one combined uncertainty distribution would not be possible[8]. For the uncertainty statistician the a priori distribution represents the knowledge of a rational agent, as opposed to the true underlying frequency distribution. Thus this counter-argument is not valid.

In summary, it is true that the error bar does not necessarily represent the width of a distribution
representing the probability density that a certain value is the true value of the measurement. In the case of a Bayesian estimate, however, which is quite frequently chosen in remote sensing, it can be conceived this way. And in the context of maximum likelihood estimates, the estimated error still can be conceived as the width of a distribution representing the likelihood that a certain value is the true value of the measurement. All these statements, however, are contingent upon the assumption
that the error or uncertainty budget is complete. This problem is deferred to Section 5.4.

### 5.3   Nonlinearity issues

The uncertainty concept relies on the possibility of evaluating uncertainties caused by measurement errors and "systematic effects" without knowledge of the true value. This is certainly granted for linear problems. The resulting uncertainty in $\hat{x}$, namely $\sigma_x$, generated by a measurement error statis-
tically characterized by its standard deviation $\sigma_y$ or by a systematic effect, e.g., a not exactly known value of a constant $b$ will be the same for each $\hat{x}$, and no specific relation between the estimate $\hat{x}$ and the true value of the measurand $x$ is required. This is because in the linear case Gaussian error propagation holds,

$$\sigma_{x,noise}^2 = \left(\frac{\partial x}{\partial y}\right)^2 \sigma_{y,noise}^2, \tag{5}$$

and

$$\sigma_{x,b}^2 = \left(\frac{\partial x}{\partial b}\right)^2 \sigma_b^2, \tag{6}$$

or for vectorial quantities

$$\mathbf{S}_{x,noise} = \mathbf{G}\mathbf{S}_y\mathbf{G}^T, \tag{7}$$

and

$$\mathbf{S}_{x,b} = \mathbf{G}_b\mathbf{S}_b\mathbf{G}_b^T, \tag{8}$$

where $\mathbf{S}_{x,noise}$, $\mathbf{S}_{x,b}$ and $\mathbf{S}_b$ are the covariance matrices and $\mathbf{G}$ and $\mathbf{G}_b$ the matrices of partial derivatives $\frac{\partial x_n}{\partial y_m}$ and $\frac{\partial x_n}{\partial b_k}$, respectivly[9].

---

[8]Willink and White (2012) might contradict here and make the claim that even systematic errors can be conceived as drawn from some population of errors, consistent with the frequentist view.

[9]For standard deviations and covariance matrices we use the notation suggested by von Clarmann et al. (2020) where the first subscript indicates the domain to which the uncertainty or error estimate refers, and the optional second subscript the source of te error.





For nonlinear problems the situation is more complicated because Equations (5) to (8) are only valid in approximation. The error scientist can invoke the concept of moderate nonlinearity (Rodgers, 2000). That is to say, $\hat{x}$ is assumed to be a reasonably good approximation of $x$, and the partial derivatives needed are evaluated at $\hat{x}$. If the error in $b$ is small enough ensuring that the resulting $\hat{x} \pm \sigma_x$ is within the range where linear approximation is justifiable, then $\sigma_x$ is a less-than-perfect but far-better-than-useless estimate of the corresponding error component in $x$.

The uncertainty scientist has a problem if they want to stay consistent with their doctrine. Since knowledge of $x$ is denied, the approximation $\hat{x} \approx x$ begs justification and it is not clear how Gaussian error estimation can be applied to systematic effects.

On the face of it, Monte Carlo error estimation or other variants of ensemble-based sensitivity studies can serve as an alternative. These, however, also invoke the nonlinear model $F$ and results thus still depend on the choice of $\hat{x}$, and any choice of this value which is not closely related to the true value $x$ will produce uncertainty estimates which are recalcitrant against any interpretation. Monte Carlo and related methods, however, are apt for the error scientists to estimate the error budget including the systematic effects if $f$ is too nonlinear to justify Gaussian error estimation.

In summary, the evaluation of uncertainties in the case nonlinearity poses a problem to the uncertainty scientist because the uncertainty estimate depends on the assumed value of the measurand, and $\hat{x} \approx x$ must be assumed. Within the framework of error analysis this assumption is allowed and measurement errors as well as systematic effects thus can be evaluated also for nonlinear inverse problems.

### 5.4 Incompleteness of the error budget

The arguments put forward above are based on the supposition that the error budget is complete. Beyond measurement noise, the total error budget includes systematic effects in the measured signal $y$, uncertainties in parameters $b$, and effects due to the chosen inverse scheme $\tilde{F}^{-1}$. If our reading of GUM08 is correct, then the most severe criticism of the 'error concept' by GUM08 is that one can never be sure that the error budget is indeed complete, and that thus the distribution with $\hat{x}$ expectation and $\sigma_x$ standard deviation cannot tell us the probability density that any value of $x$ is the true value.

The precision of a measurement is a well-behaved quantity in a sense that it is testable in a straight forward way: From three sets of collocated measurements of the same quantity, where each set is homogeneous with respect to the expected precision of its measurements, the variances of the differences provide unambigous precision estimates. The situation is more difficult for biases. Biases between different measurement systems do not tell us what the bias of one measurement system with respect to the – unfortunately unknowable – truth is. Even if the number of measurement systems is quite large, it is not guaranteed that the mean bias of all of them is zero. And an infinite number



of measurement systems is out of reach in a real world. Up to that point we concede that a positive proof of the completeness of the error budget is impossible. But this is not the end of the story.

A falsificationist (Popper, 1935) approach is more promising. It follows the rationale that it will never be possible to prove that our assumptions on the bias of a measurement system is correct. Instead, we estimate the bias as well as we can, and use it as a best estimate of the bias until some test provides evidence that the estimate is incorrect. Such a test typically consists of the intercomparison of data sets from different measurement systems. If the bias between these data sets is larger than the combined systematic error estimates, at least one of the systematic error estimates is too low and has to be refuted. Further work is then needed to find out which of the measurement systems is most likely to underestimate its systematic error. Conversely, as long as the mean difference of the measurements of the same measurand can be explained by the combined estimate of the systematic errors of both measurement systems, the systematic error estimates can be maintained, although this is, admittedly, no proof of the correctness of the error estimates. But as long as severe tests as described above are executed and the error estimates cannot be refuted, it is rational to believe that they are sufficiently complete.

We have mentioned above that the uncertainty concept depends on the acceptance of the subjective probability in the sense of degree of rational belief. Without that, an error budget including systematic effects would make no sense because systematic effects cannot easily be conceived as probabilistic in a frequentist sense; that is to say, the resulting error cannot be conceived as a random variable in a frequentist sense. Being forced to adopt the concept of probability as a degree of rational belief, it makes perfectly sense to conceive, after consideration of the Bayes theorem (see Section 5.2) the distribution with expectation $\hat{x}$ and covariance $\sigma_{x,\mathrm{total}}$ as the probability distribution which tells the rational agent the probability of any value to be the true value.

## 6 The applicability of GUM to remote sensing of the atmosphere

In this Section we identify issues where GUM08 clashes with the needs of error or uncertainty estimation in the field of remote sensing of atmospheric constituents and temperature. These issues are (1) since the atmospheric state varies quasi-continuously in space and time, the measurand is not well defined; (2) there are applications of atmospheric data where the total uncertainty estimate alone does not help; (3) Eq 11 in GUM08 is in conflict with the causal arrow, and (4) some GUM08 interpretations commit one to Bayesianism but some assumptions Bayesianism is based on cannot be logically inferred from generally accepted axioms nor traced back to observations.

### 6.1 What if the measurand is not well-defined?

On macroscopic scales, atmospheric state variables vary continuously in space on time. On microscopic scales, the typical target quantities, concentrations or temperature, are not even defined.





A typical example of this problem is be the volume mixing ratio (VMR) of a certain species at a point in the atmosphere (See also, von Clarmann, 2014). The determination of a quantity like this requires a canonical ensemble of air but in the real, inhomogenous, atmosphere, this quantity does not exist. It is an uninstantiated ideal. Due to these inhomogeneities the air volume sounded must be infinitesimally small, i.e., it must approach a point. In the real atmosphere there is either a target molecule at this point ($VMR = 1$) or another molecule ($VMR = 0$) or no molecule at all (undefined $VMR$ due to division by zero). Thus, one measures only averages over finite inhomogeneous air volumes. This approach, supposedly the only possible approach, clashes with the premise of GUM08[10] that the measurand needs to be well defined. Measuring atmospheric state variables requires the specification of the region the average is made over. The relevant toolbox of atmospheric data characterization includes concepts like resolution, averaging kernels etc. (see Rodgers, 2000 for detail). Since this type of measurements is apparently out of the scope of GUM08, the latter is very silent with respect to solutions to the problem of the characterization of measurements of quantities that are not well defined. Broadening the scope and applicability of the GUM08 framework to include less than ideally defined measurands and measurements that demand inverse methods would significantly increase the value and utility of GUM08 approach. Relevant recommendations on data characterization developed within the TUNER activity (von Clarmann et al., 2020) aim at helping to reach this goal.

## 6.2 The combined error

One of the positive aspects of GUM08 is that it breaks with the misled concept of characterizing systematic errors with 'safe bounds' (Kaarls, 1980; Kacker et al., 2007; Bich, 2012). This concept was sometimes endorsed by error statisticians subscribing to frequentism. Within a frequentist concept of probability, a probabilistic treatment of systematic errors was not easily possible because due to its systematic nature a systematic error cannot easily[11] be characterized by a frequency or probability distribution. The concept of subjective probability solves this problem. With the subjectivist's toolbox, it is no longer a problem to assign probability density functions, standard deviations and so forth when characterizing systematic errors. This possibility is a precondition for aggregating systematic and random errors to give the total error. GUM08, however, goes a step further and even denies the necessity to report random and systematic errors independently. Here we have to urge severe objections.

Von Clarmann et al. (2020) explicitly demand that error estimates be classified as random or systematic, In contrast, GUM08 (E.3.3 / E.3.7) state: "In fact, as far as the calculation of the combined standard uncertainties [...] is concerned, there is no need to classify uncertainty components and thus

---

[10]In GUM08 this problem is recognized but no solution is offered, since the term 'definitional uncertainty' is introduced in this context but not applied in practice.

[11]The qualification not *easily* was chosen because frequentists still might sample over multiple universes or apply other measures to squeeze systematic errors in a frequentist concept.





no real need for any classificational scheme." If indeed meant as written, we challenge the claim that
a total combined error budget is sufficient and therefore no classificational scheme is needed at all.
Characterizing the measurement of a unique quantity, e.g. the value of a natural constant ageed upon
by the calibration authorities, by a single error margin might be sufficient. But most measurements,
and particularly those of atmospheric state variables such as temperature, concentrations of trace
species, and so forth, deal with quantities varying as function of time and space. Any sensible use of
the resulting datasets requires a clear distinction between statistical and systematic error budgets. For
example, for time series analysis targeted at the determination of trends, the total error budget is of
no use but the random error budget is needed instead. This is because any purely additive systematic
error component cancels out in this application and their consideration would unduely distort the
weights of the data points available. In summary, the denial of the importance of distinguishing
between random errors and systematic errors does not provide proper guidance, and altogether is a
strong misjudgement.

Benevolent readers of GUM08 take the GUM authors to be saying only that the aggregation
of estimated errors to give the total error budget follows the same rules for systematic and random
errors, and that the criticized statement is not meant to deny the importance of distinguishing between
random and systematic errors beyond the mere aggregation process. If this reading is correct, we
agree, but here GUM08 leaves room for interpretation.

### 6.3   The causal arrow

Putting quantum effects aside, the measured value is unambiguously determined by the true value
via causal processes plus a given but unknown error term. In other words, the causal arrow points
from the atmospheric state to the measured value. The inverse direction, however, is ambiguous even
if we set the problem of the unknown error term aside. Many atmospheric states can cause the same
measurement, even if the noise term is exactly the same. Thus, it seems more adequate to us to
formulate the measurement problem using a function that describes the measurement as a function
of the atmospheric state, *viz.*, the measurand (Eq. 1) and not *vice versa*. The inverse measurement
equation (Eq. 2) which describes the dependence of the atmospheric state variables on the measured
values is not unambigously defined because the regularization term used to solve ill-posed problems
or the norm chosen to provide a solution for over-determined problems, all contained in the term
$\tilde{F}^{-1}$, are based on assumptions, depend on the personal preference of the person performing the
inversion, and need to be considered in the data and uncertainty characterization (See, Section 2,
and von Clarmann et al., 2020 for details). Thus we think that it is essential to appreciate the inverse
nature of the problem, and this is much easier if the measurement equation describes the forward
problem and thus does not suggest an unambiguous determination of the measurand from the mea-
sured quantity. An argument along this line of thought, but in a context wider than that of remote
sensing of the atmosphere, has been put forward by Possolo and Toman (2007).


### 6.4 Bayesian versus non-Bayesian

Some interpretations of GUM08 (e.g. White, 2016; Kacker et al., 2007) associate it with a Bayesian conception of probability. Thus one might suspect that 'uncertainty' is simply the Bayesian replacement of error. But things seem not to be so simple, for two reasons.

(1) It is, however, not quite clear which of Bayes' methods and principles a statistician has to use to be a Bayesian (c.f., e.g., Fienberg, 2006), since the Bayes theorem is accepted also by non-Bayesians, and the use of maximum likelihood methods, introduced by the almost 'militant' frequentist R. A. (Fisher, 1922) does, as far as we can judge, not commit one to use a frequentist definition of the term 'probability'.

(2) Interestingly enough, Willink and White (2012) use the term 'uncertainty' also in a frequentist framework, report that the turn to the new terminology happened already in 1980/81, and make a strong case that various allegedly purely Bayesian concepts of GUM08 can be given a valid frequentist interpretation.

Defenders of the doctrine that 'error' and 'uncertainty' have a different meaning and that the error 650 concept is a purely frequentist concept may refer to (Mayo, 1996, Ch. 13), who admits to frequentism and indeed proposes an 'error-statistical philosophy of science'. Her concept, however, does not deal with measurement errors but with errors in the acceptance or rejection of hypotheses. Thus it cannot be interpreted in a way that measurement errors are a purely frequentist concept.

In the community of remote sensing, both maximum likelihood and Bayesian retrieval schemes 655 are in use. Depending on the measurement type and the anticipated use of the data both have their pro's and con's. In order to avoid to make the rift between the Bayesian and non-Bayesian[12] part of the community even worse, the TUNER consortium has decided not to make a recommendation as to which of these retrieval schemes is thought to be superior. It was considered as more important to provide an adequate scheme for error or uncertainty estimation for any of these retrieval approaches. 660 As a consequence, it is not considered as adequate to custom-taylor uncertainty reporting to the Bayesian philosophy.

White (2016) reports that paradoxes shatter the bedrocks of Bayesian philosophy, namely the likelihood principle that says that all relevant evidence about an unknown quantity obtained from an experiment is contained in the likelihood. Others accept the theoretical validity of the Bayes theorem 665 but challenge its applicability in real life because of the unknown and unknowable prior probabilities. It has been recognized by (Hume, 2003/1739, 1748) that what was valid yesterday might not be valid tomorrow. This implies that a statistic of past events might not provide a reliable prior for future inverse problems. Also the use of the so-called non-informative prior can be challenged: The

---

[12]We challenge the dichotomy 'Bayesian vs. frequentist'. Not every non-frequentist is a whole-hearted Bayesian; not all objective probabilities are frequentist (see, e.g. Popper, 1959). Not everybody who endorses a subjective concept of probability accepts Bayesian tenets on confirmation theory and test theory. Further, subjectivist and objectivist probability concepts are not necessarily in contradiction but can be bridged (Lewis, 1980).





domain in which the prior is expressed is an *ad hoc* decision and any non-linear transformation will
render an informative prior. E.g., a flat, thus apparently non-informative velocity distribution goes
along with a non-flat, thus informative distribution of kinetic energy. Similarly an equidistribution of
droplet diameters goes along with a non-flat, thus informative, distribution of droplet volumes, etc.
This is considered by some as an absurdity brought about by the concept of non-informative priors.

More generally speaking, the Bayesian philosophy relies on a couple of unwarranted assump-
tions, e.g., the likelihood principle and the indifference principle. The proof of the former has been
challenged (Evans, 2013; Mayo, 2013, quoted after White, 2007), and the latter has been criticized
as not deducible from any accepted axioms. Thus a pro or contra Bayesian decision is a purely
philosophical decision, and it does not seem adequate to make such a decision generally binding.

While it is fully agreeable that the concept of error reporting has long relied and still relies on a
subjective, i.e., information-dependent concept of probability[13], this does not commit one to accept
Bayesianism in full.

Coming back to the title question whether error and uncertainty are different things and differ-
ent concepts, and accepting that traditional error analysis was compatible with a degree-of-belief
conception of probability, we are left with two possible interpretations. One is that it is only the en-
dorsement of a subjective concept of probability that allegedly makes uncertainty analysis Bayesian
and defines the difference between error and uncertainty. If so, we raise the objection that classical
error analysis is not a purely frequentist approach. The other interpretation is that there is more to
it, and Bayesian uncertainty analysis is indeed something entirely different. The GUM08 does not
provide a clear reference to such a Bayesian uncertainty analysis method. GUM08 makes reference
to Jeffreys (1983) as an authority of the degree-of-belief-concept of probability. Jeffreys, however,
offers no clue as to what the difference between 'error' and 'uncertainty' might be. In the context of
measurements or observations, Jeffreys always uses the term 'error' (e.g., *op. cit.*, p. 72), and often
we find statements like "the probable error [...] is the uncertainty usually quoted" (*op. cit.*, p. 72), "no
uncertainty beyond the sampling errors" (*op. cit.*, p. 389), or "treat the errors as independent" (*op.
cit.*, p. 443). With the statement that errors are not mistakes (*op. cit.*, p. 13), Jeffreys explicitly con-
tradicts the GUM pioneers (Kaarls, 1980) and GUM08 endorsers Merchant et al. (2017). Also Press
(1989) is referenced by GUM08 only to defend the use of a subjective concept of probability but not
in a context aiming at the clarification of the alleged difference between 'error' and 'uncertainty'.

We concede that Bayesians and frequentists may use the error or uncertainty estimates in a dif-
ferent way. In situations where a hypothesis is to be tested on the basis of measurement data, the
frequentist would rely on Fisherian p-values or Pearsonian rejection limits or a mixture of these ap-
proaches, while the Bayesian would assign a total probability to the hypothesis. The underlying error
or uncertainty estimates, however, are required to support both approaches. We think that a quan-

---

[13]We understand that subjective probability is related to the belief of a rational agent. Since two rational agents having
access to the same information will believe the save, this variant of subjective probability should better be called 'inter-
subjective' probability. This concept is often labelled 'objective Bayesianism'





tity for characterizing the error or uncertainty of a direct or indirect measurement which commits
the user to either a frequentist or a Bayesian use of the measurements is of little use. Reference to
Bayesianism alone cannot explain the claimed difference between 'error' and 'uncertainty'.

## 7 Conclusions

The denial that a valid connotation of the term 'error' is a statistical characterization between a
measured or estimated and the true value of the measurand would be an attempt to brush away
centuries of scientific literature. This is, however, a matter of stipulation or convention and thus
beyond the reach of a scientific argument. We thus take GUM08 to be conceding that both the
concepts, error analysis and uncertainty assessment, aim at providing a statistical characteristic of the
imperfectness of a measurement or an estimate. We understand GUM08 in a sense that the problem
of the error concept is that it conceives the estimated error as a statistical measure of the difference
between the measured or estimated value and the true value. Since the true value is unknowable,
according to GUM08 the term 'error' can neither be defined nor can its value be known.

It has been shown that the problem of the unknown true value of the measurand is a problem for
the definition of terms like 'error' or 'uncertainty' only if the concept of an operational definition is
persued. This concept, however, has its own problems and is by no means without alternative. As
soon as the concept of an operational definition is given up, problems associated with defining the
estimated error as a statistical estimate of the difference between the measurement or estimate and
the true value of the measurand disappear, and the problem remaining is only one of assigning a
reasonable value to this now well-defined quantity.

Since GUM08 did not provide many reasons why, in the context of indirect measurements, the
error allegedly cannot be estimated without knowledge of the true value, or why an uncertainty dis-
tribution does not tell us anything about the true value, we list the most obvious ones one could put
forward to bolster this claim. These are the problem of the base rate fallacy, the problem of non-
linearity, and the problem that one can never know that the error budget is complete. The problem of
the base rate fallacy can be solved by either performing a Bayesian inversion, or by conceiving the
resulting distribution as a likelihood distribution. Astonishingly enough, the GUM08's "dispersion or
range of values that could be reasonably attributed to the measurand" is determined without explicit
consideration of prior probabilities and thus cannot be interpreted in terms of posterior probability.
The problem of nonlinearity can be solved by the error scientist either by assuming that the estimate
is close enough to the true value and linearizing around this poing or by Monte-Carlo-like studies.
The uncertainty scientist who has to avoid referring to the true value is at a loss in the case of nonlin-
earity because any estimate of the uncertainty of the estimate will be correct only when evaluated at
the true value or an approximation of it. The problem of the unknown completeness of the error bud-
get can be tackled by performing comparisons between measurement systems. While this will never



provide a positive proof of the completeness of the error budget, it still justifies rational belief in its
completeness, and if error or uncertainty distributions are conceived as subjective probabilities in the
sense of degrees of rational belief, this is good enough. In summary, if (a) our reading of GUM08
is correct in the sense that the traditional error analysis can connotate a statistical quantity, and that
the key difference between the 'error' and 'uncertainty' concepts is their relation to the true value
of the target quantity and (b), that our list of arguments against the error concept is complete, and
finally, if (c) our refutation of these arguments is conclusive, then the claim that the 'error' concept
and the 'uncertainty' concepts are fundamentally different is untenable[14]. Beyond this, reasons have
been identified that put the applicability of the GUM08 concept to atmospheric measurements into
question. At the very least we can state that GUM08, by presenting their terminological stipulation
about the terms 'error' and 'uncertainty' in the appearance of a factual statement, has triggered a
linguistic discussion that distracted the attention from the more important issues how the principles
of error or uncertainty estimation, whatever one prefers to call it, could be made better applicable to
measurements beyond the idealized cases covered by their document.

*Author contributions.* TvC identified the title problem and provided a draft version of the paper. SC contributed
information on the history of the GUM and on literature on GUM (supportive and critical) and helped to under-
stand some less clear parts of GUM08; FH contributed information to the history of science; TvC contributed
information on the philosophy of science and statistics. All authors co-wrote the final version of the paper.

*Acknowledgements.* We acknowledge the scientific guidance and sponsorship of the World Climate Research
Programme to motivate this work, coordinated in the framework of SPARC, and performed as part of the
TUNER activity. The International Space Science Institute (ISSI) has hosted two team meetings and provided
further support. SC is supported by the EU H2020 project Copernicus Cal/Val Solution (CCVS), grant no.
101004242.

---

[14]Building upon Willink and White (2012), we conclude instead, that 'uncertainty' and 'uncertainty' are (at least) two
different things. This seems to hold at least when a frequentist and a Bayesian use this term. Ambiguities related to the term
'error' thus seem not to be removed but superseded by other ambiguities.





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
