# Peer review of "Truth and Uncertainty. A critical discussion of the error concept versus the uncertainty concept"

_Atmospheric Measurement Techniques, 2021_

## Referee Comment (RC1)

**Truth and Uncertainty at the Crossroads**

Antonio Possolo, *NIST Fellow*

National Institute of Standards and Technology
Gaithersburg, Maryland, USA

July 13, 2021

**Recommendation**

The article's Abstract sums up the central claims accurately that the authors develop and substantiate in their narrative, including what I believe to be the correct conclusion that the "error" and "uncertainty" concepts are not fundamentally different, and may be regarded as alternative and complementary interpretations of the doubt about the true value of the measurand that remains after measurement.

However, my reading of the *Guide to the Expression of Uncertainty in Measurement* (GUM) [Joint Committee for Guides in Metrology (JCGM), 2008] suggests less polarizing views about this issue than the views that the authors of the article under review derive from the same Guide.

Their take on things brings to mind the acerbic discussion of essentially the same issues that took place in meetings of the ISO/TAG-4 Working Group 3, around 1986-87 [Collé, 1987a,b] [Schumacher, 1987].

The article should be published after it will have been shortened and more sharply focused to convey its message most effectively, and after improvements will have been made to deficient passages that are discussed under *Specific Comments*.

The *Technical Corrections* offer an assortment of suggestions concerning English usage that the authors should consider.

**General Comments**

Acknowledgment should be made of an understanding of the relation between measurement uncertainty and measurement error that predates the GUM, and that the authors of the article under review likely will find agreeable:

> The uncertainty of a reported value is meant to be a credible estimate of the likely limits to its actual *error,* i.e., the magnitude and sign of its deviation from the truth [Eisenhart and Collé, 1980].

Churchill Eisenhart was my most illustrious predecessor at NIST, and Ronald Collé, a distinguished and esteemed NIST colleague, served as convener of the working group (ISO-TAG-4/WG3) that laid the groundwork for the creation of the GUM [Collé and Karp, 1987].

A discussion whose tenor places the "error approach" on Mars and the "uncertainty approach" on Venus sounds more like the discussions that inflamed the metrological community thirty-five years ago, than a useful discussion that we can engage in today with the benefit of the experience accumulated in these many intervening years [Eisenhart and Collé, 1980] [Collé, 1987a] [Colclough, 1987] [Schumacher, 1987].

The viewpoint that the authors of the article under review wish to convey, can be conveyed quite simply also by means of an allegory: measurement errors are the "carriers" of measurement uncertainty, in a sense analogous to how photons are the "carriers" of light waves and, more generally, of the electromagnetic force.

Accepting such dualism between errors and uncertainty facilitates the scientific discourse without excluding individual or cultural preferences, and tones down the drama that has been unfolding in the literature and that, at times, this article also exacerbates unnecessarily.

The 26 pages of text of the article under review arguably are overkill to convey this simple, conciliatory message: what they do prompt is a review almost as long as the article itself, thus making this review much too long by any standard.

In fact, the key message of the article will be delivered more effectively, and the article will have greater impact, if the article is shortened and its arguments are streamlined.

The article's length can be reduced at least by deleting those portions that distract more than they add insight: for example, the digressions in section 2 and in subsections 5.2 and 5.3.

The authors may wish to extend their criticism to the *International Vocabulary*

*of Metrology* (VIM) [Joint Committee for Guides in Metrology, 2012], whose Introduction states:

> The change in the treatment of measurement uncertainty from an Error Approach (sometimes called Traditional Approach or True Value Approach) to an Uncertainty Approach necessitated reconsideration of some of the related concepts appearing in the second edition of the VIM.

The authors also seem to be unaware of the critical evaluation of the GUM that Gleser [1998] published shortly after the original, 1993 edition of the GUM was corrected and reprinted, in 1995 [BIPM et al., 1995]. References to suitable portions of this evaluation will add value to the article under review, and will also facilitate shortening it.

The article is very repetitive in the multiple instances where it rehashes the relations between the concepts of *error*, *uncertainty*, *true value*, and *Bayesianism*. Consolidating and refocusing the fragmentary discussion of these relations would make the article much easier to read and would enhance the cogency of its arguments. However, accomplishing this would involve a major rewrite.

In its Annex E (E.5.1) the GUM addresses the issue that is the main focus of the article under review, when it states that

> The focus of this Guide is on the measurement result and its evaluated uncertainty rather than on the unknowable quantities "true" value and error (see Annex D). By taking the operational views that the result of a measurement is simply the value attributed to the measurand and that the uncertainty of that result is a measure of the dispersion of the values that could reasonably be attributed to the measurand, this Guide in effect uncouples the often confusing connection between uncertainty and the unknowable quantities "true" value and error.

The authors of the article under review quite correctly point out that the uncertainty is neither a property of the measured value nor is it *about* the measured value. The uncertainty surrounds or clouds the true value, and qualifies the state of knowledge that the metrologist has of the true value.

To the extent that the target of measurement is the true value, the measurement error is meaningful even if not observable (however, it can be estimated in some cases, as discussed below in relation with Line 180).

The suggestion, made in the aforementioned E.5.1, that "uncertainty," "error," and "true value" should be uncoupled from one another seems at odds with what is actually done in the practice of measurement science. For example, in relation with certified reference materials, "NIST asserts that a certified value provides an estimate of the true value of a defined measurand" [Beauchamp et al., 2020, 1.2.4].

Therefore, the implied understanding of the scientists developing these materials is that the uncertainties reported in the corresponding certificates are informative about the relation between the measured value and the true value, the difference between the former and the latter being the measurement error.

**Specific Comments**

The numbers in boldface refer to line numbers in the version of the preprint made available for discussion on June 29, 2021.

**002 + 245**  Here and elsewhere throughout the article, "GUM8" should be replaced by "GUM" because the GUM and its existing and planned supplements are being rearranged and renumbered, and "GUM8" is already reserved to refer to something other than the current GUM. For similar reasons, "GUM09" is likely to be misinterpreted, and should not be used: in fact, it is not needed at all because the authors use this acronym only in the very same line (245) where they introduce it.

**010** *the term 'error' was used, with some caveats, for designating a statistical estimate of the expected difference between the measured and the true value of a measurand*

The traditional and still customary meaning of "error" in statistical models is of a non-observable difference between the observed and the (generally also non-observable) true value of a quantity [Davison, 2008, Example 1.1].

For example, in the relationship $m = \mu + \varepsilon$ between a measured value, $m$, and the true value, $\mu$, of the mass of a massive entity, $\varepsilon$ is the error.

The error is generally neither known nor observable, but in many situations it can be estimated, with $\widehat{\varepsilon}$ being commonly used to denote the estimate (refer to the discussion of Line 180).

In the discussion of Lines 518 + 738 below, it will become clear how useful the explicit consideration of error can be, by allowing one conceptually to separate contributions made by different sources of uncertainty.

**015** *stipulated a new terminology, where the term 'measurement uncertainty' is used in situations where one would have said 'measurement error'*

The word "error" occurs 131 times throughout the GUM, and not always deprecatingly. For example, in its 2.2.4, the GUM acknowledges that "The definition of uncertainty of measurement [. . . ] is not inconsistent with other concepts of uncertainty of measurement, such as a measure of the possible error in the estimated value of the measurand."

**025** *the error statisticians and the uncertainty statisticians*

This classification of statisticians into these two classes is an invention of the authors that is more reflective of their imagination than of reality. In fact, the principal participants in the debates that took place in and around the aforementioned ISO/TAG-4/WG-3 were not statisticians.

Furthermore, the issue of "systematic" versus "random" errors (which we will discuss below, in relation with Line 597) may have been even more divisive than the issue of "error" versus "uncertainty."

Therefore, I urge the authors to devise a different way of characterizing the two camps they are alluding to here. A reference to Mayo and Spanos [2011] would be appropriate.

**046** *according to Bayesian statistics (Bayes, 1763) the measured value cannot always be interpreted as the most probable value of the measurand*

Since one does not need to invoke Bayesian statistics to reach the same conclusion [Possolo and Iyer, 2017, Page 011301-12], this remark is spurious.

**071** *Recapitulation of the concept of indirect measurements*

I believe that this long foray into inverse problems adds nothing of value to the discussion, hence suggest that section 2 be deleted. The discussion in subsection 6.3, *The causal arrow*, can easily be reformulated, and in the process also shortened, to drive the same points across — refer to specific suggestions below, for Line 618.

**119** *ancient researchers realized that measurement results always have errors*

It is all a matter of perspective, of course, but I am of the opinion that it is unfair to call Gauss or Legendre "ancient." In the context of European history, the word is typically reserved for the period ending with the fall of the Western Roman Empire (around 500 CE).

In addition, and in particular concerning Gauss, with whose works I am more familiar than with Legendre's, I can only say that it is difficult for me to imagine

a person of more luminous modernity, or with a better sense for what is relevant in scientific practice (of his time or contemporary), than Gauss.

**149** *In the case of 'error', its statistical estimate is mostly understood to be a quadratic estimate and thus does not carry any information about the sign of the error.*

The authors may like to replace this awkward sentence with something along the following lines: "In most cases, errors are not estimated individually. Instead, their typical size is summarized by the square root of their mean squared value, or by the median of their absolute value. Such summaries do not preserve information about the signs of any individual errors."

**154** *the term 'error' has commonly been used to signify a statistical estimate of the size of the difference between the measured and the true value of the measurand*

This is repetitive of the material around Line 10 that was discussed above. In both instances, the authors are unnecessarily turning something simple into something complicated.

One thing is the error $\varepsilon$ in the example discussed above, $m = \mu + \varepsilon$. Another thing is how this error may be characterized or quantified.

For example, the possible errors may be characterized by the probability distribution of $\varepsilon$, like when one says: the signal was corrupted by white noise with mean 0 and standard deviation $\sigma$.

The sizes of possible errors may be summarized by the mean squared error (MSE) of the estimator of the measurand, which captures the difference between expected value of the estimator and the true value of the measurand, as well as dispersion around that expected value. Other summaries include the standard deviation of the error distribution, or the now outmoded *probable error*.

The error may also be characterized indirectly, by an expression of the uncertainty surrounding the quantity of interest. For example, Yoshino et al. [1988] reported the measurement result for the absorption cross-section of ozone at 253.65 nm as $1145^{+7.1}_{-14.4} \times 10^{-20} \text{cm}^2/\text{molecule}$, which says that the measurement error has an asymmetric distribution.

**180** *Since the true value is not known, the actual difference between the measured or estimated value and the true value of the measurand cannot be calculated.*

The authors quite correctly point out that this argument lacks cogency. In fact, more can be said further to dismiss this claim as being no more than a myth.

Consider the simplest of cases of statistical estimation, where one has replicated

determinations of the same quantity, $r_1, \ldots, r_m$, which are then combined to obtain an estimate $t = T(r_1, \ldots, r_m)$ of a quantity $\tau$. The estimate $t$ could be as simple as the average or the median of the replicates, or it could be their coefficient of variation (standard deviation divided by the average).

It is then possible, using the statistical jackknife [Mosteller and Tukey, 1977, Chapter 8] or the statistical bootstrap [Efron and Tibshirani, 1993], to estimate not only the standard deviation of $t$ (based on this single set of replicates $\{r_i\}$), but also both the sign and the magnitude of the error $t - \tau$.

**200 + 364** *our reading is that an error distribution is understood as a distribution whose spread is the estimated statistical error and whose expectation value is the true value, while an uncertainty distribution is understood as a distribution whose spread is the estimated uncertainty and whose expectation value is the measured or estimated value / The error distribution must not be conceived as a probability density distribution of a value to be the true value*

In the simple model for a measured mass, $m = \mu + \varepsilon$, the "error distribution" generally refers to the probability distribution of $\varepsilon$, hence the expected value of the "error distribution" will not be $\mu$, which denotes the true value of the measurand. Instead, this expected value will be the *bias*, which is the persistent offset of $m$ from $\mu$. (Refer to the discussion of Line 597, where I explain why I prefer "persistent" to "systematic," and "volatile" to "random.")

Neither "error distribution" nor "uncertainty distribution" are mentioned in the GUM. While the GUM offers considerable guidance about the assignment of distributions to input quantities, $\{x_j\}$, in its first 69 pages (out of a total of 120) all that it provides about the probability distribution of the output quantity, $y$, is an approximation to its standard deviation, in Equations (10) and (13).

Furthermore, the GUM seems to be more concerned with evaluating $u(y)$ than with estimating the measurand optimally, because the "substitution" estimate of the measurand, which is obtained by substituting the $\{x_j\}$ by their best estimates in $y = f(x_1, \ldots, x_n)$, generally will not yield the best estimate of the measurand in the sense of minimizing mean squared error, mean absolute error, or other similar criteria [Possolo and Iyer, 2017, Page 011301-12].

In the course of those initial 69 pages, the GUM touches upon the topic of the distribution of $y$ tangentially — for example when it discusses expanded uncertainty, coverage factors, and coverage interval —, but only in its Annex G (beginning on Page 70) does the GUM venture into a discussion of how to characterize the probability distribution of $y$.

Annex G invokes the Central Limit Theorem based on a first-order Taylor approximation of the measurement function $f$ in $y = f(x_1, \ldots, x_n)$, to claim that

*y*'s distribution may be taken as being approximately Gaussian. This argument can, on occasion, be spectacularly inaccurate [Possolo, 2015, Example E11].

Since $u(y)$ typically is based on finitely many degrees of freedom, the GUM argues (using a slightly different notation) that $(y - \eta)/u(y)$, where $\eta$ denotes *y*'s true value, should have a Student's $t$ distribution approximately, wherefrom coverage intervals then issue readily, thus achieving the goal, stated in its clause 0.5, of providing "an interval about the measurement result that may be expected to encompass a large fraction of the distribution of values that could reasonably be attributed to the quantity subject to measurement."

The meaning of this distribution that the GUM, by hook or by crook assigns to $y$, and, even more importantly, the meaning of the distributions derived for $y$ by application of the Monte Carlo method, and of the coverage intervals based on them, should be the more appropriate and productive targets for critical review, similarly to what Stoudt et al. [2021] have done.

**213** *Frequentist statistics, we understand, is a concept where the term 'probability' is defined via the limit of frequencies for a sample size approaching infinity. This definition is untenable because it involves a circularity*

The authors oversimplify and are unacceptably dismissive.

If the Frequentist interpretation of probability were this "obviously" defective, then none of John Venn, Richard von Mises, Andrey Nikolaevich Kolmogorov, Jerzy Neyman or Jack Kiefer — all intellectual giants in their own right — would have embraced it.

I suggest that the authors avoid embarrassment by considering the excellent overview of the interpretations of probability compiled by Hájek [2007].

**265** *The values the rational agent believes to be true are sufficient in this case, because the error distribution does not tell us anything about the truth anyway but only about the agent's believe of what truth is.*

The simplest measurement error model mentioned above, $m = \mu + \varepsilon$, is meaningful under essentially all paradigms of statistical inference.

In neither the classical (Frequentist) nor in the Bayesian approaches does the probability distribution of $\varepsilon$ convey any information about $\mu$, other than in special cases: for example, when the variance of $\varepsilon$ depends on $\mu$.

Both approaches involve assigning a probability distribution to $\varepsilon$, which then determines the likelihood function. The Bayesian approach involves also assignments of probability distributions to $\mu$ and to any parameters in the distribution of $\varepsilon$ whose values are unknown.

The marginal distribution of $m$ typically will differ in the classical and Bayesian approaches even when the same choice is made for the distribution of $\varepsilon$.

**317** *Monte Carlo uncertainty estimation, however, is in its heart a frequentist method, because it estimates the uncertainty from the frequency distribution of the Monte Carlo samples.*

The authors are quite wrong on this one.

Of course, the extent of how wrong depends on what they mean by "Monte Carlo uncertainty estimation." I assume that they mean it in the sense and context in which it was introduced into uncertainty analysis by Morgan and Henrion [1992], subsequently having been incorporated into the GUM Supplement 1 [Joint Committee for Guides in Metrology, 2008].

In such sense and context, the Monte Carlo method is purely mathematical, and non-denominational (neither Frequentist nor Bayesian), and solves the following problem: given a random vector $X$ whose probability distribution has been fully specified, and a real-valued, measurable function $f$ defined on the range of $X$, determine the probability distribution of $Y = f(X)$.

The Monte Carlo method solves this problem using numerical methods and sampling driven by pseudo-random numbers. It solves it in the sense that it can produce the value of $\Pr(Y \in B)$ to within any specified accuracy, for any measurable subset $B$ in the range of $Y$.

The fact that its accuracy is guaranteed by the Law of Large Numbers does not make it Frequentist because the Law of Large Numbers is neither Frequentist nor Bayesian. The Law of Large Numbers is a mathematical result about sums of random variables based on Kolmogorov's axioms for probability measures [Kolmogorov, 1933].

If the authors' views on the Monte Carlo method were correct, then Markov Chain Monte Carlo sampling, which is the workhorse of contemporary Bayesian inference, would be "in its heart a frequentist method" too!

**318** *it is astonishing why GUM08, if representing a Bayesian concept, does not in the first place require to apply the Bayes theorem*

The authors should reference Gleser [1998] who points out the mixed-bag of viewpoints coexisting in the GUM. Clearly the authors are well entitled to feel astonishment at the GUM not using Bayes rule at all, especially considering the whirlwind of claims about the GUM and its Supplements being Bayesian.

However, in fairness to the GUM, such whirlwind has been more of an afterthought than a consequence of the GUM itself. First, the word "Bayes" is nowhere to be found in the GUM, and the word "Bayesian" occurs exactly once:

in the title of reference [14], on Page 115.

Only in Annex E (E.3.5) does the GUM venture into this controversial territory when it says "In contrast to this frequency-based point of view of probability, an equally valid viewpoint is that probability is a measure of the degree of belief that an event will occur." And then it adds: "Recommendation INC-1 (1980) upon which this Guide rests implicitly adopts such a viewpoint of probability."

The expression "degree of belief" occurs exactly once in the main body of the GUM (3.3.5), where it says:

> Thus a Type A standard uncertainty is obtained from a probability density function (C.2.5) derived from an observed frequency distribution (C.2.18), while a Type B standard uncertainty is obtained from an assumed probability density function based on the degree of belief that an event will occur [often called subjective probability (C.2.1)]. Both approaches employ recognized interpretations of probability.

The same expression occurs in Annex C, and again in Annex E, where E.3.6 comes the closest to advocacy by enumerating "three distinct advantages to adopting an interpretation of probability based on degree of belief."

Therefore, and on the whole, the GUM is far more discreetly or ambiguously Bayesian than it has more recently been heralded to be (surprisingly, mostly by "born again," self-declared Bayesians).

The GUM's alleged Bayesianism in fact reduces to (i) entertaining (subjective) probability distributions for input quantities that are elicited from experts, and (ii) regarding the probability distribution of the measurand as quantification of degrees of belief about the true value of the measurand, even though it is not a Bayesian posterior distribution [Gleser, 1998, 2.2].

**343** *This suggests that the uncertainty is an attribute of the true value while the error is associated with a measurement or an estimate. Because of the measurement error there is an uncertainty as to what the true value is. The uncertainty thus describes the degree of ignorance about the true value while the estimated error describes to which degree the measurement is thought to deviate from the true value*

The authors are quite right. Please consider the following rewrite, which, although allegorical, I believe further enhances the expression of the authors' sentiment — also compare with Possolo [2015, Note 3.2, Page 16]:

> This suggests that measurement uncertainty surrounds the true value

of the measurand like a fog that obfuscates it, while measurement error is both the source of that fog and part and parcel of the measured value. Measurement uncertainty thus describes the doubt about the true value of the measurand, while measurement error quantifies the extent to which the measured value deviates from the true value.

**379** *The weight of Thomas Bayes or the body height of David Hume at a certain time are well-defined quantities although we have no chance to measure them today*

I suggest that, for the sake of propriety and good taste, the authors abstain from referring to properties of the bodies of Thomas Bayes and David Hume, refined and excellent gentlemen both, long deceased, and use instead properties of other notable material entities that are no longer amenable to measurement, like the Colossus of Rhodes or the Lighthouse of Alexandria.

**411** *5.2 Likelihood, probability, and the base rate fallacy*

I believe that this subsection is a digression from the main topic that would best be deleted. A shorter, better focused article will have greater impact than one with multiple digressions that are largely off-topic.

**481** *5.3 Nonlinearity issues*

The same suggestion as for subsection 5.2, for the same reasons.

**518 + 738** *5.4 Incompleteness of the error budget*

This is an important issue that the authors should address in greater generality than in the context of inverse problems. The following example captures the key issues clearly and simply. The authors allude to the same ideas in Line 738.

The values measured in inter-laboratory studies are often modeled as $m_j = \mu + \lambda_j + \varepsilon_j$ for $j = 1,\ldots,n$, where $\mu$ denotes the true value of the quantity of interest, and the $\{\lambda_j\}$ and the $\{\varepsilon_j\}$ are errors of different kinds: the former express laboratory effects [Toman and Possolo, 2009a,b, 2010], which in many cases will be persistent effects attributable to differences between measurement methods or between forms of calibration; the latter are laboratory-specific measurement errors quantified in the uncertainties reported by the participants.

The reality of the $\{\lambda_j\}$ (that is, that they cannot all be zero) becomes apparent only when the measurement results are put on the table and inter-compared. If the measured values are significantly more dispersed than the associated, reported uncertainties intimate that they should be, then this is an indication

that there is some *dark uncertainty* [Thompson and Ellison, 2011] afoot that was not captured in the individual uncertainty budgets.

This dark uncertainty is "carried" (in the sense in which this term was used in the *General Comments*) by the $\{\lambda_j\}$. Refer to Koepke et al. [2017] and to Possolo et al. [2021] for more extended discussions of this concept.

**548**  *We have mentioned above that the uncertainty concept depends on the acceptance of the subjective probability in the sense of degree of rational belief. Without that, an error budget including systematic effects would make no sense because systematic effects cannot easily be conceived as probabilistic in a frequentist sense; that is to say, the resulting error cannot be conceived as a random variable in a frequentist sense.*

These statements are inaccurate.

First, the uncertainty concept may be contingent on a Bayesian perspective, but this perspective need not be subjective: it can be a so-called "objective Bayesian" perspective, which Jeffreys [1946], Bernardo [1979], and Berger [2006], among others, have favored.

Second, the main difficulty facing a Frequentist approach to the characterization of measurement uncertainty concerns what the GUM calls Type B evaluations of uncertainty components, not the recognition of the contributions that persistent ("systematic") effects make to said uncertainty.

In fact, the contributions from some persistent effects can be evaluated by Type A methods (refer to the comments above for line 180), and the contributions from some volatile ("random") effects can be evaluated by Type B methods (for example, the imprecision of a balance that a laboratory technician has great familiarity with).

**597**  *Von Clarmann et al. (2020) explicitly demand that error estimates be classified as random or systematic [. . . ] In summary, the denial of the importance of distinguishing between random errors and systematic errors does not provide proper guidance, and altogether is a strong misjudgment.*

The word "demand" appears to be too strong a descriptor of what von Clarmann et al. [2020] actually did, which was to "formulate *recommendations* with respect to the evaluation and reporting of random errors, systematic errors, and further diagnostic data," where the emphasis on "recommendations" is mine.

We need to discuss two separate issues regarding this point: the first concerns the choice of terms ("systematic" and "random"); the second concerns whether and when to bundle them all into a single expression of uncertainty.

Concerning the first issue — the choice of terms:

My dislike of terms like "systematic" and "random" is that they are metaphysical: they speak to the nature of the errors, which is often elusive and may be shifting.

For example, von Clarmann et al. [2020, R3, Page 4420] recognize that "depending on the application of the data, the same type of error can act as random or systematic error," and many other authors have acknowledged the same. "Random," in particular, is a thorny concept, whose definition seems to be far from settled [Landsman, 2020] [Eagle, 2016] [Bennett, 2011] [Gács, 2005].

For these reasons, I recommend descriptive qualifiers instead, for example *persistent* (instead of "systematic") and *volatile* (instead of "random"). They are less committal and afford greater flexibility, in particular to address cases where a volatile error becomes persistent, or vice versa.

Writing almost thirty-five years ago, Collé [1987a], summarized the two approaches to measurement uncertainty that were then dominant as follows:

> The "*classical*" approach is based on a central distinction between so-called random and systematic uncertainties. The uncertainties are presumably classified by the underlying physical error type [. . . ] and the approach demands that the different uncertainty types be combined by different methods. Causing even further confusion, the uncertainties in these classical treatments are said to depend on one's "perspective" and hey possess chameleon-like properties, and may change from one type to another.

> In contrast, the "*romantic*" approach dispenses with the underlying error distinction, and classifies the uncertainties only on the basis of how the uncertainty estimates were made. All uncertainty components in this approach can be combined by the same general propagation formulae. The romantic approach underlies the BIPM/CIPM Recommendation.

Concerning the second issue — the bundling of contributions from all sources of uncertainty:

While agreeing with the *romantic* approach in principle, I believe that it is advisable to consider how uncertainty evaluations will be used, before deciding whether to combine contributions from all sources of uncertainty into a single evaluation, or not. This is a more nuanced, less extreme approach than either of the two approaches aforementioned.

Consider an inter-laboratory study where several laboratories measure the same quantity independently of one another, or a meta-analysis of results from preexisting studies that were carried out and published independently of one another.

Suppose that the purpose is to blend the corresponding estimates into a consensus value: for example, as was done for the ozone absorption cross-section at 253.65 nm [Hodges et al., 2019].

Typically, the consensus value will be some form of weighted average. Therefore, the errors behind the uncertainties reported by the participants will "average out" in the process to some extent. This may be fine, or it may be inappropriate.

Such "averaging out" will be fine if laboratory-specific persistent errors lead to estimates that are high for some laboratories and low for other laboratories, with the true value lying somewhere in the middle.

But such "averaging out" will be inappropriate if a common bias, unbeknownst to all, affects all results similarly. Reporting separately the evaluation of the contributions made by persistent effects, and by volatile effects, as is commonly done in astrophysics and in particle physics, will then be an appropriate, prudent way to report uncertainty intended for use by a downstream user.

The need for such discretion, and the role that considerations of fitness-for-purpose of uncertainty evaluations should play in deciding what to do and when, is mentioned already in the pre-GUM literature [Ku, 1980].

**618** *6.3 The causal arrow — [. . . ] We think that it is essential to appreciate the inverse nature of the problem, and this is much easier if the measurement equation describes the forward problem and thus does not suggest an unambiguous determination of the measurand from the measured quantity.*

The measurement model in the GUM is only one of many kinds of measurement models to which the principles for uncertainty evaluation that are enunciated in the GUM apply. The GUM-6 [Joint Committee for Guides in Metrology, 2020], published recently, describes several other kinds of measurement models, including statistical measurement models.

Rodgers [2000, 2.3.2] explains how Bayesian statistical models can be used in general to solve inverse problems, and Ganesan et al. [2014] describe an application of hierarchical Bayesian methods to atmospheric trace gas inversions. The Bayesian approach can be fruitful in such settings because the prior distribution acts as a regularization prescription.

Possolo [2015] gives examples of measurements involving models that are quite different from the conventional measurement model in the GUM. In particular, Examples E7 (Thermistor Calibration), E17 (Gas Analysis), and E32 (Load Cell Calibration) concern calibrations that are structurally similar to the thermometer example that the authors mention in Line 109.

Using $x_1, \ldots, x_n$ and $y$ with the same roles that the GUM gives them, a statistical forward model can be formulated simply by saying $x_1, \ldots, x_n \sim L_y$, which is shorthand for "the joint probability distribution of (the random variables whose realized values are) the observable inputs $x_1, \ldots, x_n$ has $y$ as a parameter and likelihood function $L_y$.

A Bayesian formulation will then add $y \sim P$, where $P$ is the prior distribution of $y$, and application of Bayes's rule produces a solution for the inverse problem in the form of the posterior distribution, $Q$, of $y \sim Q_{x_1, \ldots, x_n}$. Compare this formulation with the treatment of calibration via conventional regularization in Hagwood [1992].

**662** *paradoxes shatter the bedrocks of Bayesian philosophy, namely the likelihood principle that says that all relevant evidence about an unknown quantity obtained from an experiment is contained in the likelihood. Others accept the theoretical validity of the Bayes theorem but challenge its applicability in real life because of the unknown and unknowable prior probabilities.*

The paradoxes alluded to often relate more to the adoption of so-called "non-informative" prior distributions than to the acceptance of the likelihood principle, as Cox [2006] points out, in a contribution referenced by White [2016].

All theories of inference have given rise to paradoxes, and nevertheless most often they produce valid and practically useful inferences. Regarding the likelihood principle in particular, at least one well-known "paradox" has been dismissed as a false alarm [Goldstein and Howard, 1991].

In any case, White [2016] does not come even close to suggesting that such paradoxes "shatter the bedrocks of Bayesian philosophy," in particular as applied in measurement science. I know for a fact that Rod White does not object to the use of Bayesian methods when these are warranted and there is genuine prior information that should be taken into account.

The objection, which is also raised by Bayesians [O'Hagan, 2006], is to the systematic reliance on "non-informative" prior distributions just for the sake of going through the motions of the Bayesian machinery or to pay lip service to scientific objectivity.

The Bayesian approach to problems of statistical inference is a choice among many that can be made, similarly to how some people choose to drink lemonade and others bourbon. Different approaches to statistical inference (be they frequentist, fiducial, or Bayesian) all can claim notable successes in solving problems of practical importance.

Bayesian methods, in particular, can boast a long and varied roster of accomplishments that prove beyond reasonable doubt that they are applicable in real life, and that they can be used to solve important practical problems, and that often they do so better than non-Bayesian alternatives [O'Hagan, 2008].

A particularly striking, recent accomplishment of Bayesian methods concerns the use of measurements of $\Delta^{14}CO_2$, in conjuction with atmospheric transport models, to demonstrate that several bottom-up approaches to the estimation of national inventories likely underestimate U.S. fossil fuel $CO_2$ emissions [Basu et al., 2020].

This study, which is based on methodological advances published in this very journal [Basu et al., 2016], includes rigorous, model-based uncertainty evaluations, and also serves to show that the GUM and its supplements have much catching-up to do if they will ever come to play a role in addressing momentous issues like the measurement of greenhouse gas emissions.

The suggestion that Bayesian methods are questionable because prior distributions are "unknown and unknowable" reveals a misconception about prior distributions: they are meant to encapsulate the knowledge that someone has about the quantity of interest, prior to performing an experiment that generates fresh information about it. Therefore, proper, informative, subjective prior distributions are known to who formulates them, by construction.

Of course, the Bayesian can be much mistaken and construct a prior distribution that reflects an erroneous conception of reality, in which case the "knowledge" that the prior encapsulates is false knowledge and its use will lead the inference astray. However, Bayesian methods cannot be blamed for delusions any more than Newton's laws can be blamed for accidental falls.

**674** *the Bayesian philosophy relies on a couple of unwarranted assumptions, e.g., the likelihood principle and the indifference principle.*

The authors convey a wrong impression on both counts.

Adherence to the likelihood principle is a choice that, in most applications, turns out to be a better choice than most alternatives. Still, it is only a choice, among many that can be made. Making such choice is necessary but not sufficient to be Bayesian. Many statisticians, physicists, chemists, and biologists adhering to the likelihood principle are not Bayesian.

Neither is adopting an indifference principle (or, more generally, using an allegedly non-informative prior distribution) necessary to qualify as being Bayesian. In fact, quite the contrary is true: reliance on proper, informative, and suitably elicited subjective prior distributions, are the hallmarks of genuine Bayesian practice. But this, too, is only a choice [Robert, 2007].

**Technical Corrections**

**025** Replace *comes down to the question if and how* with "comes down to the question of whether, and if so how"

**056** Replace *Second we assess to which degree* with "Second, we assess the degree to which"

**068** Replace *we conclude to which degree* with "we conclude the degree to which"

**128** Replace *A rich methodical toolbox* with "A rich methodological toolbox"

**180** Replace *This argument is often used to dispraise* with "This argument is often used to disparage"

**267** Replace *agent's believe* with "agent's belief"

**364** Replace *Quantities of which the value cannot determined* with "Quantities whose values cannot be determined." This suggestion deliberately ignores the antiquated invective against using the possessive *whose* for inanimate objects, consistently with the recommendation in O'Conner [2019, Page 243].

**373** Replace *Others have been formulated by us, serving, as arguments of the Devil's advocate, as working hypotheses in order to moot the error and uncertainty concepts in the context of indirect measurements* with "We have formulated others as Devil's advocates, which are intended to serve as working hypotheses to MOOT the error and uncertainty concepts in the context of indirect measurements," except that "moot" needs to be replaced by a word that is suitable for this passage: maybe "merge" or "reconcile", depending on what the authors wish to express.

**413** Replace *measurements are not in the focus* with either "measurements are not in focus" or "measurements are not the focus," depending on what the authors wish to say exactly.

**420** Replace *the probability that a person suffering fever to have Covid-19 is 50%* with "the probability is 50 % that a person with fever has COVID-19"

**429** Replace *distribution which is missing* with "distribution that is missing"

**470** Replace *the aggregarion of random uncertainties* with "the aggregation of random uncertainties"

**612** Replace *strong misjudgement* with "strong misjudgment" (unless the British spelling be preferred)

**743** The sentence that includes *traditional error analysis can connotate a statistical quantity* is unclear, and should be rewritten, taking into account the fact that the verb *connotate* is obsolete and has been replaced by *connote*. However, a term more generally familiar would be preferable, like *suggest*, possibly.

**Acknowledgments**

I thank my NIST colleagues Blaza Toman and David Newton (both from the Statistical Engineering Division), for the suggestions for improvement that they offered in relation with a draft of this contribution.

I am particularly indebted to Ronald Collé (Radioactivity Group of the Radiation Physics Division of NIST), who generously shared many of his recollections and records of the early stages of the development of the guidance that ultimately found its way into the GUM.

**References**

S. Basu, J. B. Miller, and S. Lehman. Separation of biospheric and fossil fuel fluxes of $CO_2$ by atmospheric inversion of $CO_2$ and $^{14}CO_2$ measurements: Observation System Simulations. *Atmospheric Chemistry and Physics*, 16(9): 5665–5683, 2016. doi: 10.5194/acp-16-5665-2016.

S. Basu, S. J. Lehman, J. B. Miller, A. E. Andrews, C. Sweeney, K. R. Gurney, X. Xu, J. Southon, and P. P. Tans. Estimating US fossil fuel $CO_2$ emissions from measurements of $^{14}C$ in atmospheric $CO_2$. *Proceedings of the National Academy of Sciences*, 117(24):13300–13307, 2020. doi: 10.1073/pnas. 1919032117.

C. R. Beauchamp, J. E. Camara, J. Carney, S. J. Choquette, K. D. Cole, P. C. DeRose, D. L. Duewer, M. S. Epstein, M. C. Kline, K. A. Lippa, E. Lucon, K. W. Phinney, A. Possolo, K. E. Sharpless, J. R. Sieber, B. Toman, M. R. Winchester, and D. Windover. *Metrological Tools for the Reference Materials and Reference Instruments of the NIST Materials Measurement Laboratory*. NIST Special Publication 260-136 (2020 Edition). National Institute of Standards and Technology, Gaithersburg, MD, 2020. doi: 10.6028/NIST.SP.260-136-2020.

D. Bennett. Defining randomness. In P. S. Bandyopadhyay and M. R. Forster, editors, *Philosophy of Statistics*, volume 7 of *Handbook of the Philosophy of*

*Science*, pages 633–639. North-Holland, Amsterdam, 2011. ISBN 978-0-444-51862-0. doi: 10.1016/B978-0-444-51862-0.50020-4.

J. Berger. The case for objective Bayesian analysis. *Bayesian Analysis*, 1(3): 385–402, 2006. doi: 10.1214/06-BA115.

J. M. Bernardo. Reference posterior distributions for Bayesian inference. *Journal of the Royal Statistical Society, Series B (Methodological)*, 41:113–128, 1979. doi: 10.2307/2985028.

BIPM, IEC, IFCC, ISO, IUPAC, IUPAP, and OIML. *Guide to the expression of uncertainty in measurement (GUM)*. International Organization for Standardization (ISO), Geneva, Switzerland, 1995. ISBN 92-67-10188-9. Corrected and Reprinted.

A. R. Colclough. Two theories of experimental error. *Journal of Research of the National Bureau of Standards*, 92(3):167–185, May-June 1987. doi: 10.6028/jres.092.016.

R. Collé. Minutes of the meeting on measurement uncertainties. *NCSL Newsletter*, 27(4):52–55, October 1987a.

R. Collé. Report of the second meeting of the iso/tag-4/wg-3 working group "uncertainties". *NCSL Newsletter*, 27(4):59–62, October 1987b.

R. Collé and P. Karp. Measurement uncertainties: report of an international working group meeting. *Journal of Research of the National Bureau of Standards*, 92:243–244, May 1987. doi: 10.6028/jres.092.021.

D. R. Cox. Frequentist and Bayesian statistics: a critique (Keynote Address). In L. Lyons and M. K. Ünel, editors, *Statistical Problems in Particle Physics, Astrophysics and Cosmology*, pages 3–6, London, UK, 2006. Imperial College Press. ISBN 1-86094-649-6. doi: 10.1142/9781860948985_0001. Proceedings of PHYSTAT05, Oxford, UK, 12-15 September 2005.

A. C. Davison. *Statistical Models*. Cambridge University Press, Cambridge, UK, 2008. ISBN 978-0-521-73449-3. doi: 10.1017/CBO9780511815850.

A. Eagle. Probability and randomness. In A. Hájek and C. Hitchcock, editors, *The Oxford Handbook of Probability and Philosophy*, chapter 21. Oxford University Press, Oxford, UK, 2016. ISBN 978-0199607617. doi: 10.1093/oxfordhb/9780199607617.013.22.

B. Efron and R. J. Tibshirani. *An Introduction to the Bootstrap*. Chapman & Hall, London, UK, 1993.

C. Eisenhart and R. Collé. Postscript to expression of the uncertainties of final results. In C. W. Solomon, R. D. Bograd, and W. R. Tilley, editors, *NBS Communications Manual for Scientific, Technical, and Public Information*, chapter Exhibit 2-E, pages 2–30–2–32. U.S. Dept. of Commerce, National Bureau of Standards, Gaithersburg, MD, 1980. URL https://catalog.hathitrust.org/Record/011389799. Chapter 15 of the NBS Administrative Manual.

P. Gács. Uniform test of algorithmic randomness over a general space. *Theoretical Computer Science*, 341(1):91–137, 2005. doi: 10.1016/j.tcs.2005.03.054.

A. L. Ganesan, M. Rigby, A. Zammit-Mangion, A. J. Manning, R. G. Prinn, P. J. Fraser, C. M. Harth, K.-R. Kim, P. B. Krummel, S. Li, J. Mühle, S. J. O'Doherty, S. Park, P. K. Salameh, L. P. Steele, and R. F. Weiss. Characterization of uncertainties in atmospheric trace gas inversions using hierarchical Bayesian methods. *Atmospheric Chemistry and Physics*, 14(8):3855–3864, 2014. doi: 10.5194/acp-14-3855-2014.

L. J. Gleser. Assessing uncertainty in measurement. *Statistical Science*, 13(3): 277–290, August 1998.

M. Goldstein and J. V. Howard. A likelihood paradox. *Journal of the Royal Statistical Society. Series B (Methodological)*, 53(3):619–628, 1991. doi: 10.2307/2345591.

C. Hagwood. The calibration problem as an ill-posed inverse problem. *Journal of Statistical Planning and Inference*, 31(2):179–185, 1992. doi: 10.1016/0378-3758(92)90028-Q.

A. Hájek. Interpretations of probability. In E. N. Zalta, editor, *The Stanford Encyclopedia of Philosophy*. The Metaphysics Research Lab, Center for the Study of Language and Information, Stanford University, Stanford, California, 2007. URL plato.stanford.edu/archives/win2007/entries/probability-interpret/.

J. Hodges, J. Viallon, P. J. Brewer, B. J. Drouin, V. Gorshelev, C. Janssen, S. Lee, A. Possolo, M.-A. H. Smith, J. Walden, and R. Wielgosz. Recommendation of a consensus value of the ozone absorption cross-section at 253.65 nm based on literature review. *Metrologia*, 53(3):034001, 2019. doi: 10.1088/1681-7575/ab0bdd.

H. Jeffreys. An invariant form for the prior probability in estimation problems. *Proceedings of the Royal Society of London*, 186(1007):453–461, 1946. doi: 10.1098/rspa.1946.0056.

Joint Committee for Guides in Metrology. *Evaluation of measurement data — Supplement 1 to the "Guide to the expression of uncertainty in measurement" — Propagation of distributions using a Monte Carlo method*. International Bureau of Weights and Measures (BIPM), Sèvres, France, 2008. URL www.bipm.org/en/publications/guides/gum.html. BIPM, IEC, IFCC, ILAC, ISO, IUPAC, IUPAP and OIML, JCGM 101:2008.

Joint Committee for Guides in Metrology. *International vocabulary of metrology — Basic and general concepts and associated terms (VIM)*. International Bureau of Weights and Measures (BIPM), Sèvres, France, 3rd edition, 2012. URL https://jcgm.bipm.org/vim/en/. BIPM, IEC, IFCC, ILAC, ISO, IUPAC, IUPAP and OIML, JCGM 200:2012 (2017 version with minor corrections and informative annotations).

Joint Committee for Guides in Metrology. *Guide to the expression of uncertainty in measurement — Part 6: Developing and using measurement models*. International Bureau of Weights and Measures (BIPM), Sèvres, France, 2020. BIPM, IEC, IFCC, ILAC, ISO, IUPAC, IUPAP and OIML, JCGM GUM-6:2020.

Joint Committee for Guides in Metrology (JCGM). *Evaluation of Measurement Data — Guide to the Expression of Uncertainty in Measurement*. International Bureau of Weights and Measures (BIPM), Sèvres, France, 2008. URL www.bipm.org/en/publications/guides/gum.html. BIPM, IEC, IFCC, ILAC, ISO, IUPAC, IUPAP and OIML, JCGM 100:2008, GUM 1995 with minor corrections.

A. Koepke, T. Lafarge, A. Possolo, and B. Toman. Consensus building for interlaboratory studies, key comparisons, and meta-analysis. *Metrologia*, 54(3): S34–S62, 2017. doi: 10.1088/1681-7575/aa6c0e.

A. N. Kolmogorov. *Foundations of the theory of probability*. Chelsea Publishing Co., New York, NY, second edition, 1933. Translation edited by Nathan Morrison.

H. H. Ku. Expressions of imprecision, systematic error, and uncertainty associated with a reported value. In C. W. Solomon, R. D. Bograd, and W. R. Tilley, editors, *NBS Communications Manual for Scientific, Technical, and Public Information*, chapter Exhibit 2-E, pages 2–24–2–29. U.S. Dept. of

Commerce, National Bureau of Standards, Gaithersburg, MD, 1980. URL
https://catalog.hathitrust.org/Record/011389799. Chapter 15 of the
NBS Administrative Manual.

K. Landsman. Randomness? What Randomness? *Foundations of Physics*, 50
(2):61–104, 2020. doi: 10.1007/s10701-020-00318-8.

D. G. Mayo and A. Spanos. Error statistics. In P. S. Bandyopadhyay and M. R.
Forster, editors, *Philosophy of Statistics*, volume 7 of *Handbook of the Phi-
losophy of Science*, pages 153–198. North-Holland, Amsterdam, 2011. ISBN
978-0-444-51862-0. doi: 10.1016/B978-0-444-51862-0.50005-8.

M. G. Morgan and M. Henrion. *Uncertainty — A Guide to Dealing with Uncer-
tainty in Quantitative Risk and Policy Analysis*. Cambridge University Press,
New York, NY, first paperback edition, 1992. 10th printing, 2007.

F. Mosteller and J. W. Tukey. *Data Analysis and Regression*. Addison-Wesley
Publishing Company, Reading, Massachusetts, 1977. ISBN 0-201-04854-X.

P. T. O'Conner. *Woe is I: The Grammarphobe's Guide to Better English in Plain
English*. Riverhead Books, New York, NY, fourth edition, 2019. ISBN 978-
0525533-054.

A. O'Hagan. Science, subjectivity and software (comment on articles by berger
and by goldstein). *Bayesian Analysis*, 1(3):445–450, September 2006. doi:
10.1214/06-BA116G.

A. O'Hagan. The Bayesian Approach to Statistics. In T. Rudas, editor, *Hand-
book of Probability: Theory and Applications*, chapter 6. Sage Publications,
Thousand Oaks, CA, 2008. ISBN 978-1-4129-2714-7. doi: 10.4135/
9781452226620.n6.

A. Possolo. *Simple Guide for Evaluating and Expressing the Uncertainty of
NIST Measurement Results*. National Institute of Standards and Technology,
Gaithersburg, MD, 2015. doi: 10.6028/NIST.TN.1900. NIST Technical Note
1900.

A. Possolo and H. K. Iyer. Concepts and tools for the evaluation of measurement
uncertainty. *Review of Scientific Instruments*, 88(1):011301, 2017. doi: 10.
1063/1.4974274.

A. Possolo, A. Koepke, D. Newton, and M. R. Winchester. Decision tree for key
comparisons. *Journal of Research of the National Institute of Standards and
Technology*, 126:126007, 2021. doi: 10.6028/jres.126.007.

C. P. Robert. *The Bayesian Choice: From Decision-Theoretic Foundations to Computational Implementation*. Springer, New York, NY, second edition, 2007. ISBN 978-0-387-71598-8.

C. D. Rodgers. *Inverse Methods for Atmospheric Sounding: Theory and Practice*, volume 2 of *Atmospheric, Oceanic, and Planetary Physics*. World Scientific, Singapore, 2000. ISBN 981-02-2740-X.

R. B. F. Schumacher. A dissenting position on uncertainties. *NCSL Newsletter*, 27(4):55–59, October 1987.

S. Stoudt, A. Pintar, and A. Possolo. Coverage intervals. *Journal of Research of the National Institute of Standards*, 126:126004, 2021. doi: 10.6028/jres. 126.004.

M. Thompson and S. L. R. Ellison. Dark uncertainty. *Accreditation and Quality Assurance*, 16:483–487, October 2011. doi: 10.1007/s00769-011-0803-0.

B. Toman and A. Possolo. Model-based uncertainty analysis in inter-laboratory studies. In F. Pavese, M. Bär, A. B. Forbes, J. M. Linares, C. Perruchet, and N. F. Zhang, editors, *Advanced Mathematical and Computational Tools in Metrology and Testing: AMCTM VIII*, volume 78 of *Series on Advances in Mathematics for Applied Sciences*, pages 330–343. World Scientific Publishing Company, Singapore, 2009a. ISBN 981-283-951-8.

B. Toman and A. Possolo. Laboratory effects models for interlaboratory comparisons. *Accreditation and Quality Assurance*, 14:553–563, October 2009b. doi: 10.1007/s00769-009-0547-2.

B. Toman and A. Possolo. Erratum to: Laboratory effects models for interlaboratory comparisons. *Accreditation and Quality Assurance*, 15:653–654, 2010. doi: 10.1007/s00769-010-0707-4.

T. von Clarmann, D. A. Degenstein, N. J. Livesey, S. Bender, A. Braverman, A. Butz, S. Compernolle, R. Damadeo, S. Dueck, P. Eriksson, B. Funke, M. C. Johnson, Y. Kasai, A. Keppens, A. Kleinert, N. A. Kramarova, A. Laeng, B. Langerock, V. H. Payne, A. Rozanov, T. O. Sato, M. Schneider, P. Sheese, V. Sofieva, G. P. Stiller, C. von Savigny, and D. Zawada. Overview: Estimating and reporting uncertainties in remotely sensed atmospheric composition and temperature. *Atmospheric Measurement Techniques*, 13(8):4393–4436, 2020. doi: 10.5194/amt-13-4393-2020.

D. R. White. In pursuit of a fit-for-purpose uncertainty guide. *Metrologia*, 53: S107–S124, 2016. doi: 10.1088/0026-1394/53/4/S107.

K. Yoshino, D.E. Freeman, J.R. Esmond, and W.H. Parkinson. Absolute absorption cross-section measurements of ozone in the wavelength region 238-335 nm and the temperature dependence. *Planetary and Space Science*, 36(4): 395–398, 1988. doi: 10.1016/0032-0633(88)90127-4.

---

## Referee Comment (RC3)

**Comments and suggestions for the Authors**

**Manuscript title:** Truth and Uncertainty. A critical discussion of the error concept versus the uncertainty concept

    In this manuscript the authors present an argument that the the "error concept" and the framework put forth by the GUM (deemed the "uncertainty concept") are the same. The major issue I understand that the authors take with the GUM is in the recommendation that uncertainties reported with estimates of measurement need not be specifically interpreted with respect to errors and "true values." The authors refute this claim by seemingly arguing that uncertainties or "estimated errors" cannot be interpreted without reference to true values and therefore the concepts must be the same. To be honest, it was extremely difficult to parse through the unnecessarily lengthy 23 pages of text to come to the understanding that this is (I believe) the authors main argument, and critically I do not believe this argument is effectively made. In general the manuscript is too long with repetitive sections that are often confusing and in some places contradictory. The message is often lost in unnecessary language arguments between the GUM and the authors' definition of 'error' and in generally narrow and misconceived discussions about frequentist vs Bayesian statistical methods. More seriously, the language used in reference to statistical concepts is imprecise and in some areas completely incorrect. The authors should define their terms with equations where applicable and adhere to commonly accepted mathematical/statistical/probabilistic definitions. In several places the statistical interpretations of their "error estimates" in relation to "true" values are overly simplified and likely to be misinterpreted, in particular when models are misspecified. Rather than focusing on an argument that the uncertainties typically reported under the "error concept" can be also be interpreted as under the "uncertainty concept", they seem to miss the point of GUM (as I interpret GUM, but I would also argue more broadly the understanding of these concepts in the field of statistics) that reported uncertainties need not come with inferential statements about how close estimates are to the true value (e.g. actual errors) to be useful for comparison to other estimates. Instead the focus seems to be mainly a language argument that "true value" is the same as "value of the measurand" and so the concepts must be the same – a not particularly useful argument in my opinion.

    Without considerable revision and restructuring I do not believe the manuscript provides a useful contribution to AMT. In fact, I am concerned that publication in its present form would propagate dangerous misconceptions about statistical methods and uncertainty quantification to the community. In addition, the authors do not provide a concise, understandable overview of the "error" and "uncertainty concepts" and the supposed differences, which narrows the manuscript's audience to those who are already well-versed in the GUM and the specific error analysis framework the authors consider. I do agree with the authors that the quantitative methods laid out under the GUM framework are not inconsistent with the traditional error analysis framework and, if properly understood, the interpretations of such quantities under both frameworks are generally in agreement. It is my belief that a useful manuscript would argue these points very concisely, showing that the recommendation in GUM are not inconsistent with traditional methods in the atmospheric remote sensing community, and would focus more attention on addressing how the GUM principles apply to atmospheric retrievals and where GUM may fall short.

**General comment about "true values" and uncertainties**

    The authors need to clearly state what they mean by "true value" in their arguments. Specifically, when discussing true values are they referring to the truth in terms of reality (if such a quantity exists) or the true value in terms of the specified statistical model and resulting theory? The latter are the only

"true values" that have any statistical guarantees in the interpretation of uncertainty estimates and are *only* equivalent to the true value in reality if the statistical model perfectly describes the true data generating process (i.e. is the "correct" model), which we know is unlikely to be the case particularly in atmospheric remote sensing retrievals. As an example, consider maximum likelihood estimation for atmospheric remote sensing retrievals. Measured radiances $\boldsymbol{y}$ are assumed to be generated from a true state of the atmosphere $\boldsymbol{x}$ through a "true" radiative transfer function $\boldsymbol{f}$, and the true data generating process may be idealized as

$$\boldsymbol{y} = \boldsymbol{f}(\boldsymbol{x}, \boldsymbol{b}) + \epsilon$$

assuming correctly specified Gaussian random errors, $\epsilon$. In practice, as the authors point out, $\boldsymbol{f}$ is not fully known and is replaced by a function $\boldsymbol{F}$ that represents the radiative transfer function to the best of the scientists knowledge. In this case, the observation equation used for inference is

$$\boldsymbol{y} = \boldsymbol{F}(\boldsymbol{x}, \boldsymbol{b}) + \epsilon$$

and the statistical model is "misspecified" in relation to the true model. Under both models, the radiances are assumed to be generated from the specified distribution with unknown true state $\boldsymbol{x}_0$, but crucially these "true values" are not the same under both models! The MLE, $\hat{\boldsymbol{x}}$, has the interpretation of the value of $\boldsymbol{x}$ such that the model (correct or misspecified) generates radiances most similar to what is observed (i.e. given $\hat{\boldsymbol{x}}$ the observed radiances are most probable). Assuming regularity conditions hold and reasonably large sample size, the sampling distribution of $\hat{\boldsymbol{x}}$ is approximately Gaussian with mean equal to $\boldsymbol{x}_0$ (under the model) and the standard deviation represents an estimate of the expected deviation of the estimate from that true value, $\boldsymbol{x}_0$. Under the correct model, $\boldsymbol{x}_0$ is interpreted as the true state of the atmosphere, but under the misspecified model $\boldsymbol{x}_0$ is the value of $\boldsymbol{x}$ that minimizes the difference between the true data generating model and the misspecified model. The degree to which this true value matches the true target depends on the degree of misspecification which is not known.

Therefore, statements about unknown true values (reality) based on misspecified models ("all models are wrong") are *inferential* and conditional on all of the assumptions and uncertainties in the measurement system. I do not read GUM as dispensing with the concept of the true value, I understand GUM to recommend that when reporting uncertainties associated with estimates of a value of a measurand (GUM agrees "value of a measurand" can be synonymous with "true" value of the measurand) it is not necessary to make inferential statements about actual errors *specifically when reported uncertainties are meant to be used to assess reliability/consistency with other measurement systems.* That is, if I have two different measurement frameworks providing interval (uncertainties) of plausible values of a measurand, these intervals can be used to compare consistency with each other without needing to know the "true value." In this case, it is only necessary to describe uncertainty estimates as summarizing a range of estimates that would also be plausible for the measurand under the measurement system, which is still consistent with quantities reported under the "error concept." Consider an uncertainty (or 'error') estimate, $\sigma_x$, related to parameter/measurand $\boldsymbol{x}$, e.g. the standard deviation of a sampling distribution of $\hat{\boldsymbol{x}}$ (frequentist) or the posterior standard deviation of the posterior $p(x|y)$ (Bayesian). Under a frequentist approach, $\sigma_x$ describes how much the *estimate* is expected to vary around its statistical expectation $E(\hat{\boldsymbol{x}})$ and represents the spread of values that would also be plausible values of the estimator if the experiment were to be repeated, given the same assumptions in the measurement system. Under a Bayesian paradigm, $\sigma_x$ describes variability around the posterior mean and provides information on the spread of plausible values (estimates) of the measurand that are also consistent with the scientists' knowledge given the observations, assumptions and prior knowledge. Of course, you can argue that $\sigma_x$ also describes the spread of the "error distribution" $\hat{\boldsymbol{x}} - \boldsymbol{x}_0$ but this doesn't describe the expected magnitude of actual errors, the mean of the "error distribution" $E(\hat{\boldsymbol{x}}) - \boldsymbol{x}_0$, unless the estimator is unbiased (see related comment about MSE vs variance below). Given this, I do not understand what the authors' issue with this GUM recommendation is, unless they are simply arguing that "value of a measurand" also means "true value of a measurand" (that the GUM agrees with) in

which case I see this as quibbling about words and not addressing the larger concept of whether it is necessary to make inferential statements of the form e.g. "95% confident that the true value is within some interval" when reporting uncertainties.

**Additional comments**

1. The authors spend several pages (sections) arguing that in addition to the universally accepted statistical definition of error as the difference between measured/estimated and the "truth", a second definition of the word error be accepted (deemed 'error') to refer to statistical estimate of the expected differences between the observed/estimated and true value. This secondary 'error' definition proves confusing in multiple places as it is unclear to which error the authors are referring to, be it actual error or 'error', thus inadvertently making an argument for GUM's choice of separation in language of uncertainty estimates and actual errors. In general, the arguments about language definitions of "uncertainty" and "error" could be summarized much more concisely in about a paragraph, acknowledging that the GUM definition of 'uncertainty (of measurement)' encompasses the same quantities that have have often been shorthandedly referred to with reference to the word error as "error estimates", "error bars", etc. Therefore, I think large portions of sections 3 and 4 are repetitive and could be removed.

2. Page 1, lines 10-11: I find the definition of 'error' as *designating a statistical estimate of the expected difference between the measured and the true value of a measurand* to be not in agreement with standard deviations they later reveal are often use as "error estimates' in remote sensing retrievals (e.g. section 5.3). The authors definition is consistent with statistical summaries of error like root mean squared error (RMSE) which estimates the square root of the expected squared difference of actual errors, or median absolute difference the mean of the absolute value of actual errors. The variance of an estimator is only theoretically equal to the MSE if the estimator is unbiased, and even in that case the variability is around the true *model* parameter of a potentially misspecified model, not necessarily reality. Any inferential statements about the true value in reality and distributions of actual errors are conditioned on all assumptions and uncertainties in the measurement system being reasonably correct. The authors need to clarify their language in regards to what they mean by 'error', "true values", and how these definitions apply to the uncertainty estimates they reference later. Otherwise I am concerned that there is a serious underlying misunderstanding of how to interpret uncertainties they report.

3. Page 1, lines 10-11: This is also the first place in the paper where the failure to use consistent mathematical notation is problematic. Consider the simple statistical model

$$X = \mu + \epsilon,$$

where $\epsilon$ is a random variable representing actual measurement error. What the authors contend is 'error' could be written, $E(X - \mu)$ where $E(\cdot)$ denotes the statistical expectation and $\mu$ is the "true value" of the measurand. This quantity is equivalent to $E(e)$ and would represent measurement bias. Or do they mean this to represent $var(\epsilon)$, that is $E(\epsilon - E(\epsilon))^2$? Or instead do they intend to refer to the same manner of quantities but with respect to an estimator of $\mu$ given a set of observations of $X$, $x_1, \ldots, x_n$, say $\hat{\boldsymbol{x}}$? The latter would be consistent with what the authors presents in Section 2, but it would help immensely if the authors provided some manner of illustrative model, and used it to clarify their ensuing arguments.

4. Page 1, lines 14-16: I do not believe GUM presents a "contrasting" definition of the term error. GUM presents the universally accepted statistical definition of error, and defines "uncertainty" to quantify the spread of plausible values given uncertainties in the system. What do the authors mean here by "measurement error" that the term "measurement uncertainty" is replacing? $Var(\epsilon)$? Then, on page 3, lines 75-76, the authors refer to $\epsilon$ as the actual "measurement error" in the $y$-domain. Is this the

same reference to measurement error as in line 16 or there is 'measurement error' meant to refer to the variance or standard deviation of the actual measurement errors ($\epsilon$), If the latter, this inconsistency makes more of an argument for GUM's separation of language definitions of 'error' and 'uncertainty' than for the authors' definition.

5. Page 1, lines 23-24: The authors state, "The claim is made that the uncertainty concept can be construed without reference to the unknown and unknowable true value while the error concept can not." What specifically is the error concept to which they are referring? I read GUM as saying "errors" (as defined as actual measurement errors) cannot be construed without knowledge of the true value (in the example above, $\mu$), meaning that value of a realization of the random variable $\epsilon$ in the example above can't be known because $\mu$ is unknown, and analogously the resulting difference between and estimate $\hat{x}$ and $\boldsymbol{mu}$. However, the parameters of the distribution of $\epsilon$ (describing its mean and variance for example) can be discussed, reasoned about, and even estimated from data with some additional assumptions. How does the claim in the following sentence (lines 25-26) follow from this?

6. Page 2, lines 25-26: The authors state that the dispute comes down to "the question if and how the error (or uncertainty) distribution is related to the true value of the measurand." Again, here is where imprecise terminology is confusing. By "error distribution" I would assume they mean the distribution of the random measurement errors $\epsilon$, but what do they mean by "uncertainty distribution" (or do they mean that the word "uncertainty" is now a synonym for "error distribution")? So the dispute is about the relationship between $\epsilon$ and $\mu_0$? How? Or by error distribution do the mean some distribution of the estimator - the truth, e.g. $\hat{x} - \mu_0$?

7. Lines 24-27: The distinction between "error" vs "uncertainty" statisticians is artificial, I am not aware of any such distinction nor do I believe any such dispute or "rift" along these lines exists in the statistical community (I am a practicing statistician). Please cite a reference for the existence of this rift, if you have one.

8. Line 128: It has yet to be clearly stated what the authors define to be the debated difference between "error estimation and uncertainty assessment." A concise definition of both and the argued against definitions at the beginning of the paper would greatly improve the presentation.

9. Line 138-148: I can only assume the second meaning of the term 'error' the authors are referring to are shorthand statements that have been historically made in the literature such as "the estimated error of quantity of interest is X." These statements typically use terms like "estimated error" to represent a quantity like a standard deviation of a sampling distribution or a posterior standard deviation, and it is assumed that the reader/community understands this implicit definition (and that is does not provide information about the actual truth without inference). Again, I do not find the argument over whether this specific quantity should be referred to as "uncertainty" or "estimated error" to be particularly compelling, but rather a discussion of the interpretation of these quantities seems to be needed.

10. Line 190: I do not see how this is not quibbling about words. What does referring to "uncertainty" under the GUM definition as 'error' under the error concept provide that the GUM definition does not? Other than what seems to be a generally misused definition regarding the "truth" in the authors' definition of 'error'. I think the authors would agree that the two meanings of error set forth in this manuscript refer to different concepts.

11. Line 199: I cannot find reference to "error distribution" or "uncertainty distribution" in GUM. The definitions provided here are consistent with a sampling distribution of an estimator ("error distribution") and a posterior distribution ("uncertainty distribution"). Is this what is intended? If so, please adhere to well-defined statistical definitions. If not, please clarify.

12. Lines 233-239: I see no reason why uncertainties reported as in GUM, along with assumptions of the statistical model, cannot be used for hypothesis testing. In (frequentist) hypothesis testing an assumption is made about the true state, in which case the truth is assumed known and inference is made based on how reasonable this assumption is given the variability (uncertainty) of plausible estimates under the measurement system. If the assumed value of the truth is outside what the scientist believes to be plausible based on their understanding of the measurement system and uncertainties, then a decision is made that the hypothesized value is unlikely to be the true value. A Bayesian hypothesis test would argue whether or not an assumed value or range of values for the parameter are consistent or not with posterior knowledge (uncertainties).

13. Line 265: Again, what is meant by "error distribution"?

14. Lines 317-318: *Monte Carlo uncertainty estimation, however, is in its heart a frequentist method, because it estimates the uncertainty from the frequency distribution of the Monte Carlo samples.* This statement is fundamentally false. Monte Carlo methods are simply methods to solve numerical problems through sampling and are used in both frequentist and Bayesian statistics.

15. Section 5.1: The GUM definition of "uncertainty" does not dispense with reference to the measurand only to its true value. To this end, GUM is consistent with the authors statement *we conceive the definition of a quantity and the assignment of the value to a quantity as quite different things.* In general this section reads more as a language "gotcha" argument against the GUM's use of the term operational definition rather than in a useful argument about the definition of uncertainty, and as such I'd suggest omitting.

16. Section 5.2: This section should be omitted. It presents incomplete and oversimplified interpretations of Bayesian and frequentist methods that are distracting to the manuscript.

---

## Author Comment (AC1)

**Review #1: 'Truth and Uncertainty at the Crossroads' by Antonio Possolo**

**Comment: Recommendation**
The article's Abstract sums up the central claims accurately that the authors develop and substantiate in their narrative, including what I believe to be the correct conclusion that the "error" and "uncertainty" concepts are not fundamentally different, and may be regarded as alternative and complementary interpretations of the doubt about the true value of the measurand that remains after measurement.
However, my reading of the Guide to the Expression of Uncertainty in Measurement (GUM) [Joint Committee for Guides in Metrology (JCGM), 2008] suggests less polarizing views about this issue than the views that the authors of the article under review derive from the same Guide.
Their take on things brings to mind the acerbic discussion of essentially the same issues that took place in meetings of the ISO/TAG-4 Working Group 3, around 1986-87 [Collé, 1987a,b] [Schumacher, 1987].
The article should be published after it will have been shortened and more sharply focused to convey its message most effectively, and after improvements will have been made to deficient passages that are discussed under Specific Comments.

**Reply:** The authors thank the reviewer for this insightful and thorough review of their manuscript.

**Action:** See below, under specific comments.

**Comment: The Technical Corrections offer an assortment of suggestions concerning English usage that the authors should consider.**

**Reply:** The authors appreciate these corrections.

**Action:** See below for details.

**Comment: General Comments**
Acknowledgment should be made of an understanding of the relation between measurement uncertainty and measurement error that predates the GUM, and that the authors of the article under review likely will find agreeable: "The uncertainty of a reported value is meant to be a credible estimate of the likely limits to its actual error, i.e., the magnitude and sign of its deviation from the truth" [Eisenhart and Collé, 1980]. Churchill Eisenhart was my most illustrious predecessor at NIST, and Ronald Collé, a distinguished and esteemed

NIST colleague, served as convener of the working group (ISO-TAG-4/WG3) that laid the groundwork for the creation of the GUM [Collé and Karp, 1987].

A discussion whose tenor places the "error approach" on Mars and the "uncertainty approach" on Venus sounds more like the discussions that inflamed the metrological community thirty-five years ago, than a useful discussion that we can engage in today with the benefit of the experience accumulated in these many intervening years [Eisenhart and Collé, 1980] [Collé, 1987a] [Colclough, 1987] [Schumacher, 1987]. The viewpoint that the authors of the article under review wish to convey, can be conveyed quite simply also by means of an allegory: measurement errors are the "carriers" of measurement uncertainty, in a sense analogous to how photons are the "carriers" of light waves and, more generally, of the electromagnetic force.

Accepting such dualism between errors and uncertainty facilitates the scientific discourse without excluding individual or cultural preferences, and tones down the drama that has been unfolding in the literature and that, at times, this article also exacerbates unnecessarily.

**Reply:** Agreed.

**Action:** Relevant parts of the paper were reorganized and rewritten. Care has been taken not to give the impression of a split of the community in two blocs in hostile opposition. Relevant references have been included.

**Comment: The 26 pages of text of the article under review arguably are overkill to convey this simple, conciliatory message: what they do prompt is a review almost as long as the article itself, thus making this review much too long by any standard.**

**In fact, the key message of the article will be delivered more effectively, and the article will have greater impact, if the article is shortened and its arguments are streamlined.**

**The article's length can be reduced at least by deleting those portions that distract more than they add insight: for example, the digressions in section 2 and in subsections 5.2 and 5.3.**

**Reply:** We agree to shorten/rewrite particularly Section 2. We are not really convinced that Sections 5.2 and 5.3 are disgressions. The impossibility to evaluate errors/uncertainties in the nonlinear case without approximate knowledge of the true value and the inadequacy to conceive the measured value as the most probable value without consideration of the a priori probability seem essential to us. Instead we shortened the paper by reorganizing it in a way that repetition could better be avoided. Further, we have deleted the Section on Bayesianism versus non-Bayesianism, because the current version of GUM is not fully Bayesian, and this Section was thus connected only loosely to the

main topic of the paper.

**Action:** The paper has been substantially reorganized and shortened.

**Comment: The authors may wish to extend their criticism to the International Vocabulary of Metrology (VIM) [Joint Committee for Guides in Metrology, 2012], whose Introduction states: "The change in the treatment of measurement uncertainty from an Error Approach (sometimes called Traditional Approach or True Value Approach) to an Uncertainty Approach necessitated reconsideration of some of the related concepts appearing in the second edition of the VIM."**

**Reply:** Agreed.

**Action:** Added before the criticism is formulated: "The International Vocabulary of Metrology document (BIPM, 2012) points in the same direction"

**Comment: The authors also seem to be unaware of the critical evaluation of the GUM that Gleser [1998] published shortly after the original, 1993 edition of the GUM was corrected and reprinted, in 1995 [BIPM et al., 1995]. References to suitable portions of this evaluation will add value to the article under review, and will also facilitate shortening it.**

**Reply:** Yes, indeed. We were unaware of this review.

**Action:** Reference to Gleser (1998) has been made.

**Comment: The article is very repetitive in the multiple instances where it rehashes the relations between the concepts of error, uncertainty, true value, and Bayesianism. Consolidating and refocusing the fragmentary discussion of these relations would make the article much easier to read and would enhance the cogency of its arguments. However, accomplishing this would involve a major rewrite.**

**Reply:** Agreed.

**Action:** The paper has been reorganized, and parts of the article have been rewritten, in order to avoid repetition and to make the structure of the argument clearer.

**Comment: In its Annex E (E.5.1) the GUM addresses the issue that is the main focus of the article under review, when it states that "The focus of this Guide is on the measurement result and its evaluated uncertainty rather than on the unknowable quantities "true" value and error (see Annex D). By taking the operational views that the result**

of a measurement is simply the value attributed to the measurand and that the uncertainty of that result is a measure of the dispersion of the values that could reasonably be attributed to the measurand, this Guide in effect uncouples the often confusing connection between uncertainty and the unknowable quantities "true" value and error." The authors of the article under review quite correctly point out that the uncertainty is neither a property of the measured value nor is it about the measured value. The uncertainty surrounds or clouds the true value, and qualifies the state of knowledge that the metrologist has of the true value. To the extent that the target of measurement is the true value, the measurement error is meaningful even if not observable (however, it can be estimated in some cases, as discussed below in relation with Line 180).

**Reply:** Fully agreed.

**Comment:** The suggestion, made in the aforementioned E.5.1, that "uncertainty," "error," and "true value" should be uncoupled from one another seems at odds with what is actually done in the practice of measurement science. For example, in relation with certified reference materials, "NIST asserts that a certified value provides an estimate of the true value of a defined measurand" [Beauchamp et al., 2020, 1.2.4].
Therefore, the implied understanding of the scientists developing these materials is that the uncertainties reported in the corresponding certificates are informative about the relation between the measured value and the true value, the difference between the former and the latter being the measurement error.

**Reply:** We fully agree.

**Action:** None, because the Beauchamp-statement seems to address the uncertainty issue only indirectly. Although this quotation could possibly strengthen our point, its inclusion would imply considerably more text, which we want to avoid for reasons of brevity.

**Comment: Specific Comments**
The numbers in boldface refer to line numbers in the version of the preprint made available for discussion on June 29, 2021.
**002 + 245** Here and elsewhere throughout the article, "GUM8" should be replaced by "GUM" because the GUM and its existing and planned supplements are being rearranged and renumbered, and "GUM8" is already reserved to refer to something other than the current GUM. For similar reasons, "GUM09" is likely to be misinterpreted, and should not be used: in fact, it is not needed at all because the authors use this acronym only in the very same line (**245**)

**where they introduce it.**

**Reply: Thank you for bringing this to our attention.**

**Action:** To be able to still distinguish between GUM in general and the 2008 version, and not to clash with the GUM numbering system, we have changed 'GUM08' to 'GUM-2008'. The abbreviation GUM09 is not used any longer.

**Comment: Specific Comments**
**010** *the term 'error' was used, with some caveats, for designating a statistical estimate of the expected difference between the measured and the true value of a measurand*
**The traditional and still customary meaning of "error" in statistical models is of a non-observable difference between the observed and the (generally also non-observable) true value of a quantity [Davison, 2008, Example 1.1].**
**For example, in the relationship $m = \mu + \epsilon$ between a measured value, m, and the true value, $\mu$, of the mass of a massive entity, $\epsilon$ is the error.**
**The error is generally neither known nor observable, but in many situations it can be estimated, with $\epsilon$ being commonly used to denote the estimate (refer to the discussion of Line 180). In the discussion of Lines 518 + 738 below, it will become clear how useful the explicit consideration of error can be, by allowing one conceptually to separate contributions made by different sources of uncertainty.**

**Reply:** We fully agree.

**Action:** In the course or rewriting the terminology part, we have made clearer that in the traditional terminology, the term 'error' implies an equivocation.

**Comment: 015** *stipulated a new terminology, where the term 'measurement uncertainty' is used in situations where one would have said 'measurement error'*
**The word "error" occurs 131 times throughout the GUM, and not always deprecatingly. For example, in its 2.2.4, the GUM acknowledges that "The definition of uncertainty of measurement [...] is not inconsistent with other concepts of uncertainty of measurement, such as a measure of the possible error in the estimated value of the measurand."**

**Reply**: In the definitions part, GUM defines error as the difference between the measured value and the measurand. There is no instance in GUM where they acknowledge the equivocation that the term 'error' can also refer to a statistical estimate of this difference. On page 5, Note 3, GUM says "In this Guide, great care is taken to distinguish between the terms "error" and "uncertainty". They

are not synonyms, but represent completely different concepts; they should not be confused with one another or misused." This, we think, justifies our statement.

**Action:** We have not taken any action in reply to this specific comment, but our rewriting of the introductory sections in reply to the general comment should have made our point clearer.

**Comment: 025** *the error statisticians and the uncertainty statisticians*
**This classification of statisticians into these two classes is an invention of the authors that is more reflective of their imagination than of reality. In fact, the principal participants in the debates that took place in and around the aforementioned ISO/TAG-4/WG-3 were not statisticians.**

**Reply:** Agreed.

**Action:** In the new version of the manuscript, the terms "error statistician" and "uncertainty statistician" are no longer used.

**Comment: Furthermore, the issue of "systematic" versus "random" errors (which we will discuss below, in relation with Line 597) may have been even more divisive than the issue of "error" versus "uncertainty." Therefore, I urge the authors to devise a different way of characterizing the two camps they are alluding to here. A reference to Mayo and Spanos [2011] would be appropriate.**

**Reply:** We use a definition of random vs. systematic errors which is based fully on observational grounds. This terminology has been agreed by the entire TUNER consortium. To avoid confusion, we do not want to change the terminology again.

**Action:** We mention that our random errors correspond to the volatile errors and that our systematic errors correspond to the persistent errors. However, we use the terms 'volatile' and 'persistent' as purely descriptive terms, not as new technical terms.

**Comment: 046** *according to Bayesian statistics (Bayes, 1763) the measured value cannot always be interpreted as the most probable value of the measurand*
**Since one does not need to invoke Bayesian statistics to reach the same conclusion [Possolo and Iyer, 2017, Page 011301-12], this remark is spurious.**

**Reply:** Here we have to respectfully disagree. The argument of Possolo and Iyer, 2017, Page 011301-12 is based on the assumption that the error correlations between multiple measured values are not known or not considered when

a higher-level data product is produced. Under this assumption the conclusion that the resulting value is not the most probable one is correct. However, von Clarmann et al. (2020) offer a method to estimate the higher-level data product (here: trace gas mixing ratios) under consideration of the full measurement error covariance matrix. Even if the latter does include all uncertainties and covariances, and the Possolo an Iyer argument thus does not apply, still the base rate problem is there, and without consideration of the a priori probabilities the estimate will **not** render the most probable mixing ratios but only the most likely ones. Thus, we do not see what is spurious about the base rate argument.

**Comment: 071 Recapitulation of the concept of indirect measurements I believe that this long foray into inverse problems adds nothing of value to the discussion, hence suggest that section 2 be deleted. The discussion in subsection 6.3, The causal arrow, can easily be reformulated, and in the process also shortened, to drive the same points across — refer to specific suggestions below, for Line 618.**

**Reply:** We agree in part. This section contained unnecessary formalism and detail, and it was too long. However, to understand how a measured or estimated value is probabilistically related to the true value, we still think that it is important to highlight the inverse nature of a measurement process.

**Action:** This section has been merged with other sections and has been considerably shortened. All unnecessary formalism has been removed.

**Comment: 119** *ancient researchers realized that measurement results always have errors*
**It is all a matter of perspective, of course, but I am of the opinion that it is unfair to call Gauss or Legendre "ancient." In the context of European history, the word is typically reserved for the period ending with the fall of the Western Roman Empire (around 500 CE). In addition, and in particular concerning Gauss, with whose works I am more familiar than with Legendre's, I can only say that it is difficult for me to imagine a person of more luminous modernity, or with a better sense for what is relevant in scientific practice (of his time or contemporary), than Gauss.**

**Reply:** The term 'ancient' was by no means meant in any dismissive way. We realize that it can be understood in this way and replace it.

**Action:** New wording: "Investigators realized already in the 19th century that measurement results always have errors."

**Comment: 149** *In the case of 'error', its statistical estimate is mostly understood to be a quadratic estimate and thus does not carry any information about the sign of the error.*

The authors may like to replace this awkward sentence with something along the following lines: "In most cases, errors are not estimated individually. Instead, their typical size is summarized by the square root of their mean squared value, or by the median of their absolute value. Such summaries do not preserve information about the signs of any individual errors."

**Reply:** We agree to reword this statement:

**Action:** New wording: In the case of 'error', its statistical estimate is mostly understood to be the square root of the variance of the probability density function of the error and thus does not carry any information about the sign of the error.

**Comment: 154** *the term 'error' has commonly been used to signify a statistical estimate of the size of the difference between the measured and the true value of the measurand*
This is repetitive of the material around Line 10 that was discussed above. In both instances, the authors are unnecessarily turning something simple into something complicated. One thing is the error $\epsilon$ in the example discussed above, $m = \mu + \epsilon$. Another thing is how this error may be characterized or quantified.

**Reply:** Agreed; however, we prefer to leave this part and avoid the repetition elsewhere in the paper.

**Action:** The text has been restructured and shortened in order to make the arguments clearer and to avoid repetition. The new structure is intended to make our arguments w.r.t. terminological versus structural arguments clearer.

**Comment:** For example, the possible errors may be characterized by the probability distribution of $\epsilon$, like when one says: the signal was corrupted by white noise with mean 0 and standard deviation $\sigma$.
The sizes of possible errors may be summarized by the mean squared error (MSE) of the estimator of the measurand, which captures the difference between expected value of the estimator and the true value of the measurand, as well as dispersion around that expected value. Other summaries include the standard deviation of the error distribution, or the now outmoded probable error.
The error may also be characterized indirectly, by an expression of the uncertainty surrounding the quantity of interest. For example, Yoshino et al. [1988] reported the measurement result for the absorption cross-section of ozone at **253.65 nm** as $1145^{+7.1}_{-144}10^{-20}$ cm$^2$/molecule, which says that the measurement error has an asymmetric distribution.

**Reply:** Agreed. Most such cases in our context are caused by the non-linearity of Beer's law. Symmetrically distributed errors in the transmission measurements cause an asymmetric distribution of the inferred cross section errors. This highlights how important the non-linearity issue actually can be.

**Action:** Added: "Nonlinear error propagation may in some cases make asymmetric error estimates adequate."

**Comment: 180** *Since the true value is not known, the actual difference between the measured or estimated value and the true value of the measurand cannot be calculated.*
**The authors quite correctly point out that this argument lacks cogency. In fact, more can be said further to dismiss this claim as being no more than a myth. Consider the simplest of cases of statistical estimation, where one has replicated determinations of the same quantity, $r_1, \ldots, r_m$, which are then combined to obtain an estimate $t = T(r_1, \ldots, r_m)$ of a quantity $\tau$. The estimate t could be as simple as the average or the median of the replicates, or it could be their coefficient of variation (standard deviation divided by the average). It is then possible, using the statistical jackknife [Mosteller and Tukey, 1977, Chapter 8] or the statistical bootstrap [Efron and Tibshirani, 1993], to estimate not only the standard deviation of t (based on this single set of replicates $r_i$), but also both the sign and the magnitude of the error $t - \tau$.**

**Reply:** We are happy that we agree on this important point. With respect to the example presented in the review, however, GUM defenders would probably object that these methods provide a handle on the random part of the error but not on what they call "systematic" effects.

**Action:** None, for the sake of brevity.

**Comment: 200 + 364** *our reading is that an error distribution is understood as a distribution whose spread is the estimated statistical error and whose expectation value is the true value, while an uncertainty distribution is understood as a distribution whose spread is the estimated uncertainty and whose expectation value is the measured or estimated value / The error distribution must not be conceived as a probability density distribution of a value to be the true value*
**In the simple model for a measured mass, $m = \mu + \epsilon$, the "error distribution" generally refers to the probability distribution of $\epsilon$, hence the expected value of the "error distribution" will not be $\mu$, which denotes the true value of the measurand. Instead, this expected value will be the *bias*, which is the persistent offset of $m$ from $\mu$. (Refer to the discussion of Line 597, where I explain why I prefer "persistent" to "systematic," and "volatile" to "random.")**
**Neither "error distribution" nor "uncertainty distribution" are men-**

tioned in the GUM. While the GUM offers considerable guidance about the assignment of distributions to input quantities, $x_j$, in its first 69 pages (out of a total of 120) all that it provides about the probability distribution of the output quantity, $y$, is an approximation to its standard deviation, in Equations (10) and (13).

Furthermore, the GUM seems to be more concerned with evaluating $u(y)$ than with estimating the measurand optimally, because the "substitution" estimate of the measurand, which is obtained by substituting the $x_j$ by their best estimates in $y = f(x_1, \ldots, x_n)$, generally will not yield the best estimate of the measurand in the sense of minimizing mean squared error, mean absolute error, or other similar criteria [Possolo and Iyer, 2017, Page 011301-12].

In the course of those initial 69 pages, the GUM touches upon the topic of the distribution of y tangentially – for example when it discusses expanded uncertainty, coverage factors, and coverage interval –, but only in its Annex G (beginning on Page 70) does the GUM venture into a discussion of how to characterize the probability distribution of y. Annex G invokes the Central Limit Theorem based on a first-order Taylor approximation of the measurement function $f$ in $y = f(x_1, \ldots, x_n)$, to claim that $y$'s distribution may be taken as being approximately Gaussian. This argument can, on occasion, be spectacularly inaccurate [Possolo, 2015, Example E11].

Since $u(y)$ typically is based on finitely many degrees of freedom, the GUM argues (using a slightly different notation) that $(y - \nu)/u(y)$, where $\nu$ denotes $y$'s true value, should have a Student's t distribution approximately, wherefrom coverage intervals then issue readily, thus achieving the goal, stated in its clause 0.5, of providing "an interval about the measurement result that may be expected to encompass a large fraction of the distribution of values that could reasonably be attributed to the quantity subject to measurement."

The meaning of this distribution that the GUM, by hook or by crook assigns to $y$, and, even more importantly, the meaning of the distributions derived for $y$ by application of the Monte Carlo method, and of the coverage intervals based on them, should be the more appropriate and productive targets for critical review, similarly to what Stoudt et al. [2021] have done.

**Reply:** The example discussed by [Possolo and Iyer, 2017, Page 011301-12] inquires into a case where a higher level data product (here: the area of a rectancle) is calculated from the direct measurements of the lengths of the vertices. Possolo and Iyer demonstrate that the most probable area cannot be determined without knowledge of the error correlation of the length measurements of both vertices. Von Clarmann et al. (2020) concede that retrieved values of atmospheric state variables are not optimal as long as the measurement error covariance matrix includes only measurement noise but not the other error sources, mapped into the measurement space. The problem mentioned by

Possolo and Iyer falls in this category of problems. In the paper under review, we indeed forgot to consider this problem. But independent from this, our argument invoking the base rate still holds.

Gaussian error propagation (extended to consider covariances) holds, regardless of the error distributions of the ingoing quantities, as long as the function through which the errors are propagated is sufficiently linear. In particular, it is not required that the errors of the ingoing quantities follow a Gaussian distribution. Monte Carlo methods are needed either if the function is too nonlinear for Gaussian error propagation (we tackle this issue in old Section 5.3, new Section 3.3), if more information than only the expectation value and the variance shall be inferred (beyond the scope of our paper) or if the measurement error covariance matrix used for the inversion does not include all error components. In the latter case MC methods can be used to correct the result, but that is somewhat beyond the issue of error estimation. However, MC methods do not solve the problem of the base rate fallacy.

**Action:**. Added "[...yield the probability distribution of any value to be the true value.] This holds even if the error distribution is extended to include also systematic effects, and if all error correlations are adequately taken into account in the case of multi-dimensional measurements."

**Comment: 213** *Frequentist statistics, we understand, is a concept where the term 'probability' is defined via the limit of frequencies for a sample size approaching infinity. This definition is untenable because it involves a circularity* **The authors oversimplify and are unacceptably dismissive. If the Frequentist interpretation of probability were this "obviously" defective, then none of John Venn, Richard von Mises, Andrey Nikolaevich Kolmogorov, Jerzy Neyman or Jack Kiefer – all intellectual giants in their own right – would have embraced it.**
**I suggest that the authors avoid embarrassment by considering the excellent overview of the interpretations of probability compiled by Hájek [2007].**

**Reply:** If this argument was conclusive, then we should either believe in the ether theory or we should deny Lorentz and Poincaré the status of an intellectual giant. Scientific knowledge is approximately accumulative, and thus it is not astonishing that later generations know more than earlier generations. Stegmüller's argument runs as follows: Hypothetical frequentism (i.e. frequentism involving an infinite reference class; this seems necessary to be able to extend the probability concept to probability density functions of continuous random variables) rely on the large number theorem. Both the weak and the strong version of the large number theorem involve probabilities. Thus the definition is circular or involves an infinite regress. We do not see any flaw in his argument, and this argument is refuted neither by Hájek [2007] nor by any other literature of our knowledge.

It is important to note that the distinction between frequentism and other conceptions of probability such as subjective probability is about how the term 'probability' is defined, and not about the methods used. Methods still can work well, even if the definition of the underlying key term is flawed.

**Action:** As a compromise, we have replaced "is untenable" with "challenged".

**Comment: 265** *The values the rational agent believes to be true are sufficient in this case, because the error distribution does not tell us anything about the truth anyway but only about the agent's believe of what truth is.*
**The simplest measurement error model mentioned above, $m = \mu + \epsilon$, is meaningful under essentially all paradigms of statistical inference. In neither the classical (Frequentist) nor in the Bayesian approaches does the probability distribution of $\epsilon$ convey any information about $\mu$, other than in special cases: for example, when the variance of $\epsilon$ depends on $\mu$.**
**Both approaches involve assigning a probability distribution to $\epsilon$, which then determines the likelihood function. The Bayesian approach involves also assignments of probability distributions to $\mu$ and to any parameters in the distribution of $\epsilon$ whose values are unknown. The marginal distribution of $m$ typically will differ in the classical and Bayesian approaches even when the same choice is made for the distribution of $\epsilon$.**

**Reply:** Agreed for the mathematical part but we are afraid that our original text did not make clear the point we intended to make. Thus we reword our statement for clarity.

**Action:** Reworded: "In GUM the error concept is discarded because the capability of conducting an error estimate allegedly depends on the knowledge of the true value. However, once having invoked the concept of subjective probability, no objective knowledge of the unknowable true value is needed any longer. The subjectivist can work with the value they believes to be true. This solves the alleged problem of the error concept, namely, that the true value is unknown."

**Comment: 317** *Monte Carlo uncertainty estimation, however, is in its heart a frequentist method, because it estimates the uncertainty from the frequency distribution of the Monte Carlo samples.*
**The authors are quite wrong on this one.**
**Of course, the extent of how wrong depends on what they mean by "Monte Carlo uncertainty estimation." I assume that they mean it in the sense and context in which it was introduced into uncertainty analysis by Morgan and Henrion [1992], subsequently having been incorporated into the GUM Supplement 1 [Joint Committee for Guides in Metrology, 2008]. In such sense and context, the Monte Carlo method is purely mathematical, and non-denominational (neither Frequentist nor Bayesian), and solves the following problem:**

given a random vector $X$ whose probability distribution has been fully specified, and a real-valued, measurable function $f$ defined on the range of $X$, determine the probability distribution of $Y = f(X)$. The Monte Carlo method solves this problem using numerical methods and sampling driven by pseudo-random numbers. It solves it in the sense that it can produce the value of $Pr(Y \in B)$ to within any specified accuracy, for any measurable subset $B$ in the range of $Y$. The fact that its accuracy is guaranteed by the Law of Large Numbers does not make it Frequentist because the Law of Large Numbers is neither Frequentist nor Bayesian. The Law of Large Numbers is a mathematical result about sums of random variables based on Kolmogorov's axioms for probability measures [Kolmogorov, 1933]. If the authors' views on the Monte Carlo method were correct, then Markov Chain Monte Carlo sampling, which is the workhorse of contemporary Bayesian inference, would be "in its heart a frequentist method" too!

**Reply:** We still think that, by employing samples, MC methods do use the frequentist toolbox. A Monte Carlo sample is a sample where values considered as more probable occur more frequently. But since we think that the use of the frequentist toolbox does not make a subjectivist a frequentist, and since this statement is not needed to support our stance, we decided to withdraw our argument on MC methods.

**Action:** Argument deleted.

**Comment: 318** *it is astonishing why GUM08, if representing a Bayesian concept, does not in the first place require to apply the Bayes theorem*
The authors should reference Gleser [1998] who points out the mixed-bag of viewpoints coexisting in the GUM. Clearly the authors are well entitled to feel astonishment at the GUM not using Bayes rule at all, especially considering the whirlwind of claims about the GUM and its Supplements being Bayesian.
However, in fairness to the GUM, such whirlwind has been more of an afterthought than a consequence of the GUM itself. First, the word "Bayes" is nowhere to be found in the GUM, and the word "Bayesian" occurs exactly once: $n$ the title of reference [14], on Page 115.
Only in Annex E (E.3.5) does the GUM venture into this controversial territory when it says "In contrast to this frequency-based point of view of probability, an equally valid viewpoint is that probability is a measure of the degree of belief that an event will occur." And then it adds: "Recommendation INC-1 (1980) upon which this Guide rests implicitly adopts such a viewpoint of probability."
The expression "degree of belief" occurs exactly once in the main body of the GUM (3.3.5), where it says: "Thus a Type A standard

uncertainty is obtained from a probability density function (C.2.5) derived from an observed frequency distribution (C.2.18), while a Type B standard uncertainty is obtained from an assumed probability density function based on the degree of belief that an event will occur [often called subjective probabil- ity (C.2.1)]. Both approaches employ recognized interpretations of probability.

The same expression occurs in Annex C, and again in Annex E, where E.3.6 comes the closest to advocacy by enumerating "three distinct advantages to adopting an interpretation of probability based on degree of belief." Therefore, and on the whole, the GUM is far more discreetly or ambiguously Bayesian than it has more recently been heralded to be (surprisingly, mostly by "born again," self-declared Bayesians).

The GUM's alleged Bayesianism in fact reduces to (i) entertaining (subjective) probability distributions for input quantities that are elicited from experts, and (ii) regarding the probability distribution of the measurand as quantification of degrees of belief about the true value of the measurand, even though it is not a Bayesian posterior distribution [Gleser, 1998, 2.2].

**Reply:** We do not claim that GUM is Bayesian. However, we recognize that there exist readings of GUM that see it as Bayesian. We do not say that we **are** astonished that the Bayes theorem plays no role in GUM but we say we should be astonished that the Bayes theorem plays no role in GUM if GUM really was Bayesian. Our argument follows the form of a reduction to the absurd: As a working hypothesis we assume that GUM is Bayesian, we then find that this assumptions lead to inconsistencies, and finally we conclude that the purported Bayesian turn cannot explain the difference between the error concept and the uncertainty concept.

**Action:** We have reworded the related text to make the argument clearer, and to make it more obvious that our argument form is a *reductio ad absurdum.*

**Comment: 343** *This suggests that the uncertainty is an attribute of the true value while the error is associated with a measurement or an estimate. Because of the measurement error there is an uncertainty as to what the true value is. The uncertainty thus describes the degree of ignorance about the true value while the estimated error describes to which degree the measurement is thought to deviate from the true value*

The authors are quite right. Please consider the following rewrite, which, although allegorical, I believe further enhances the expression of the authors' sentiment – also compare with Possolo [2015, Note 3.2, Page 16]: This suggests that measurement uncertainty surrounds the true value of the measurand like a fog that obfuscates it, while measurement error is both the source of that fog and part and parcel of the measured value. Measurement uncertainty thus describes the

**doubt about the true value of the measurand, while measurement error quantifies the extent to which the measured value deviates from the true value.**

**Reply:** Agreed.

**Action:** Footnote added.

**Comment: 379** *The weight of Thomas Bayes or the body height of David Hume at a certain time are well-defined quantities although we have no chance to measure them today*
**I suggest that, for the sake of propriety and good taste, the authors abstain from referring to properties of the bodies of Thomas Bayes and David Hume, refined and excellent gentlemen both, long deceased, and use instead properties of other notable material entities that are no longer amenable to measurement, like the Colossus of Rhodes or the Lighthouse of Alexandria.**

**Reply:** Agreed.

**Action:** Changed as suggested; footnote added with reference to the originator of this idea for this illustrative example.

**Comment: 411** *5.2 Likelihood, probability, and the base rate fallacy*
**I believe that this subsection is a digression from the main topic that would best be deleted. A shorter, better focused article will have greater impact than one with multiple digressions that are largely off-topic.**

**Reply:** Since we conceive measurements (and their analysis) as estimation of the true value involving an inverse process, we think that this argument is quite essential. We do, however, agree that this section was too long, and that the original structure of the paper did not make our argument sufficiently clear.

**Action:** This concern has been considered when the manuscript was reorganized.

**Comment: 481** *5.3 Nonlinearity issues*
**The same suggestion as for subsection 5.2, for the same reasons.**

**Reply:** The denial of the approximate knowledge of the true value poses serious problems particular in the case of non-linearity. In linear cases the statistical error can be estimated without knowledge of the true value because the Jacobian does not depend on the measurand. This is not the case if there is non-linearity. We do not think that this is a digression or off-topic, but a serious challenge to an uncertainty concept that pretends to be able to avoid the concept of the

true value.

**Comment: 518 + 738** *5.4 Incompleteness of the error budget*
This is an important issue that the authors should address in greater generality than in the context of inverse problems. The following example captures the key issues clearly and simply. The authors allude to the same ideas in Line **738**.
The values measured in inter-laboratory studies are often modeled as $m_j = \mu + \lambda_j + \epsilon_j$ for $j = 1, \ldots, n$, where $\mu$ denotes the true value of the quantity of interest, and the $\lambda_j$ and the $\epsilon_j$ are errors of different kinds: the former express laboratory effects [Toman and Possolo, 2009a,b, 2010], which in many cases will be persistent effects attributable to differences between measurement methods or between forms of calibration; the latter are laboratory-specific measurement errors quantified in the uncertainties reported by the participants.
The reality of the $\lambda_j$ (that is, that they cannot all be zero) becomes apparent only when the measurement results are put on the table and inter-compared.
If the measured values are significantly more dispersed than the associated, reported uncertainties intimate that they should be, then this is an indication that there is some dark uncertainty [Thompson and Ellison, 2011] afoot that was not captured in the individual uncertainty budgets.
This dark uncertainty is "carried" (in the sense in which this term was used in the General Comments) by the $\lambda_j$. Refer to Koepke et al. [2017] and to Possolo et al. [2021] for more extended discussions of this concept.

**Reply:** The rationale behind limiting our discussion to inverse problems is that this manuscript has been written for an AMT special issue that deals exclusively with error reporting in remote sensing contexts. There are certainly many people who are better qualified than we are to discuss these issues in the context of general metrology. We see it as our task to scrutinize the applicability and usefulness of the BIPM guidelines to remote sensing of the atmosphere.
By the way: Those passages in our manuscript where the incompleteness of the error budget is discussed do not refer explicitly to inverse problems.

**Comment: 548** *We have mentioned above that the uncertainty concept depends on the acceptance of the subjective probability in the sense of degree of rational belief. Without that, an error budget including systematic effects would make no sense because systematic effects cannot easily be conceived as probabilistic in a frequentist sense; that is to say, the resulting error cannot be conceived as a random variable in a frequentist sense.*
These statements are inaccurate.
First, the uncertainty concept may be contingent on a Bayesian perspective, but this perspective need not be subjective: it can be

**a so-called "objective Bayesian" perspective, which Jeffreys [1946], Bernardo [1979], and Berger [2006], among others, have favored.**

**Reply:** The uncertainty concept explicitly invokes subjective probability (Sect 3.3.5). We think that the uncertainty concept is contingent on the concept of subjective probability (i.e., a degree of belief concept of probability) but not necessarily on Bayesianism. While objective probability (both frequentist probability and Popper's propensity) is a characteristic of the event (the object), subjective probability is a characteristic of the knowledge of the agent (the subject) dealing with this event. Any concept of probability that conceives errors, uncertainties etc as a degree of ignorance and characterize them with a degree of belief are thus based on subjective probability, because the ignorance is a characteristic of the agent (the subject), not of the value (object).

Thus we think that even objective Bayesians employ the concept of subjective probability. That is to say, also objective Bayesianism depends on a concept of probability which describes the information (and its uncertainty) an agent (the 'subject') has. Thus one could argue that the term "objective Bayesianism" is a misnomer, or at least is misleading.

Our use of the terms 'subjective probability' and our understanding of 'objective Bayesianism' seem to be fully consistent with D. R. White (2016 Metrologia 53 S107) who states: "There are two main branches of Bayesian statistics, objective and subjective, both of which are founded on three main principles: the use of subjective probability, the use of Bayes' theorem to invert conditional probabilities, and the likelihood principle" (his Section 3.2). Thus, the contradiction between subjective probability and objective Bayesianism is only apparent but not real. Anyway, since GUM invokes subjective probability but not Bayesianism, the discussion of objective Bayesianism is not relevant to the paper.

**Action:** (Only indirectly linked to this comment) We have deleted the section on Bayesianism versus non-Bayesianism.

**Comment: Second, the main difficulty facing a Frequentist approach to the characterization of measurement uncertainty concerns what the GUM calls Type B evaluations of uncertainty components, not the recognition of the contributions that persistent ("systematic") effects make to said uncertainty.**

**Reply:** We must distinguish between methods that use frequency distributions as estimators of probability distributions and the way how the concept of 'probability' is defined. Type B evaluations work well even for frequency distributions. Take a finite sample of parameters $b_i$, calculate the variance, propagate the variance through the system using linear theory; and in parallel, propagate each parameter $b_i$ through the system and calculate the variance in the result space. If the system is sufficiently linear, both variances will be approximately the same (and exactly the same in the linear case). We see no difficulty to apply Type B evaluations to frequency distributions.

The problem solved by GUM by abolishing frequentism and turning towards a subjective concept of probability is that systematic effects cannot be characterized by a probability distribution at all. In a frequentist's world it would simply make no sense to assign an error distribution to a systematic effect, because the systematic effect is only one number. To assign an error distribution to a systematic effect requires to conceive the error distribution as a characterization of the knowledge or belief of the rational agent instead of a frequency of events. And this is exactly what the subjective concept of probability does. For a frequentist it is absurd to assign a distribution to a systematic error.

**Comment: In fact, the contributions from some persistent effects can be evaluated by Type A methods (refer to the comments above for line 180), and the contributions from some volatile ("random") effects can be evaluated by Type B methods (for example, the imprecision of a balance that a laboratory technician has great familiarity with).**

**Reply:** We agree that there is no clear correspondence between Type A vs. Type B evaluation of uncertainty on the one hand and random vs. systematic error components on the other hand. TUNER recommends to evaluate all error components via Type B analysis. This seems necessary to disentangle the different error components (i.e. error from the different sources). For example, the error due to measurement noise is estimated by propagating known measurement noise through the retrieval. The same holds for, e.g., volatile parameter uncertainties. Type A evaluation is recommended to test the validity of the Type B estimates.

**Comment: 597** *Von Clarmann et al. (2020) explicitly demand that error estimates be classified as random or systematic [. . . ] In summary, the denial of the importance of distinguishing between random errors and systematic errors does not provide proper guidance, and altogether is a strong misjudgment.*
**The word "demand" appears to be too strong a descriptor of what von Clarmann et al. [2020] actually did, which was to "formulate *recommendations* with respect to the evaluation and reporting of random errors, systematic errors, and further diagnostic data," where the emphasis on "recommendations" is mine.**
**We need to discuss two separate issues regarding this point: the first concerns the choice of terms ("systematic" and "random"); the second concerns whether and when to bundle them all into a single expression of uncertainty.**
**Concerning the first issue – the choice of terms:**
**My dislike of terms like "systematic" and "random" is that they are metaphysical:**
**they speak to the nature of the errors, which is often elusive and may be shifting.**
**For example, von Clarmann et al. [2020, R3, Page 4420] recognize that "depending on the application of the data, the same type of er-**

ror can act as random or systematic error," and many other authors have acknowledged the same.

"Random," in particular, is a thorny concept, whose definition seems to be far from settled [Landsman, 2020] [Eagle, 2016] [Bennett, 2011] [Gács, 2005].

For these reasons, I recommend descriptive qualifiers instead, for example persistent (instead of "systematic") and volatile (instead of "random"). They are less committal and afford greater flexibility, in particular to address cases where a volatile error becomes persistent, or vice versa. Writing almost thirty-five years ago, Collé [1987a], summarized the two approaches to measurement uncertainty that were then dominant as follows: The "classical" approach is based on a central distinction between so-called random and systematic uncertainties. The uncertainties are presumably classified by the underlying physical error type [...] and the approach demands that the different uncertainty types be combined by different methods. Causing even further confusion, the uncertainties in these classical treatments are said to depend on one's "perspective" and hey possess chameleon-like properties, and may change from one type to another. In contrast, the "romantic" approach dispenses with the underlying error distinction, and classifies the uncertainties only on the basis of how the uncertainty estimates were made. All uncertainty components in this approach can be combined by the same general propagation formulae. The romantic approach underlies the BIPM/CIPM Recommendation.

**Reply:** We agree to replace "demand" with "recommend". We are reluctant to change terminology in favor of 'volatile' and 'persistent' errors because this would lead to inconsistent terminology within the AMT special issue this paper is written for. However the connotation of our terms 'random errors' and 'systematic' errors, as defined in von Clarmann et al. (2020) matches that of the 'volatile' and 'persistent' errors of Possolo. There, systematic errors are defined as bias-generating errors while random errors are defined as variance-generating errors. They can be distinguished fully on observational grounds.

**Action:** "demand" replaced with "recommend". Further we have added the following footnote: "In this context it is important to note that, in contrast to some older conceptions, von Clarmann et al. (2020) define 'systematic errors' as bias-generating errors and 'random errors' as variance-generating errors. To avoid confusion with the older conceptions, one can use instead the descriptive terms 'persistent' and 'volatile' errors as suggested by Possolo (2021). This is not done here to maintain consistency with von Clarmann et al. (2020).

**Comment: Concerning the second issue – the bundling of contributions from all sources of uncertainty:**
While agreeing with the romantic approach in principle, I believe that

it is advisable to consider how uncertainty evaluations will be used, before deciding whether to combine contributions from all sources of uncertainty into a single evaluation, or not. This is a more nuanced, less extreme approach than either of the two approaches aforementioned.

Consider an inter-laboratory study where several laboratories measure the same quantity independently of one another, or a meta-analysis of results from preexisting studies that were carried out and published independently of one another.

Suppose that the purpose is to blend the corresponding estimates into a consensus value: for example, as was done for the ozone absorption cross-section at 253.65 nm [Hodges et al., 2019].

Typically, the consensus value will be some form of weighted average. There- fore, the errors behind the uncertainties reported by the participants will "average out" in the process to some extent. This may be fine, or it may be inappropriate. Such "averaging out" will be fine if laboratory-specific persistent errors lead to estimates that are high for some laboratories and low for other laboratories, with the true value lying somewhere in the middle. But such "averaging out" will be inappropriate if a common bias, unbeknownst to all, affects all results similarly. Reporting separately the evaluation of the contributions made by persistent effects, and by volatile effects, as is commonly done in astrophysics and in particle physics, will then be an appropriate, prudent way to report uncertainty intended for use by a downstream user.

The need for such discretion, and the role that considerations of fitness-for-purpose of uncertainty evaluations should play in deciding what to do and when, is mentioned already in the pre-GUM literature [Ku, 1980].

**Reply:** We fully agree that the choice if errors of different type shall be bundled or not depends on the application. However, we still uphold our criticism that the denial of the importance of distinguishing between random errors and systematic errors does not provide proper guidance, and altogether is a strong misjudgment. The data provider does, in general, not know what the data user will do with the data, and since the data user may use the data for a purpose where it is essential to distinguish between systematic and random errors, the information which error component contributes to which category must be provided.

**Action:** Added: "[...is a strong misjudgment]. The data users must be provided with all information required to taylor the relevant error budget to the given application of the data."

**Comment: 618** *6.3 The causal arrow – [. . . ] We think that it is essential to appreciate the inverse nature of the problem, and this is much easier if the mea-*

*surement equation describes the forward problem and thus does not suggest an unambiguous determination of the measurand from the measured quantity.*

The measurement model in the GUM is only one of many kinds of measurement models to which the principles for uncertainty evaluation that are enunciated in the GUM apply. The GUM-6 [Joint Committee for Guides in Metrology, 2020], published recently, describes several other kinds of measurement models, including statistical measurement models.

Rodgers [2000, 2.3.2] explains how Bayesian statistical models can be used in general to solve inverse problems, and Ganesan et al. [2014] describe an application of hierarchical Bayesian methods to atmospheric trace gas inversions.

The Bayesian approach can be fruitful in such settings because the prior distribution acts as a regularization prescription.

Possolo [2015] gives examples of measurements involving models that are quite different from the conventional measurement model in the GUM. In particular, Examples E7 (Thermistor Calibration), E17 (Gas Analysis), and E32 (Load Cell Calibration) concern calibrations that are structurally similar to the thermometer example that the authors mention in Line 109.

Using $x_1, \ldots, x_n$ and $y$ with the same roles that the GUM gives them, a statistical forward model can be formulated simply by saying $x_1, \ldots, x_n \sim L_y$, which is shorthand for "the joint probability distribution of (the random variables whose realized values are) the observable inputs $x_1, \ldots, x_n$ has $y$ as a parameter and likelihood function $L_y$.

A Bayesian formulation will then add $y \sim P$, where $P$ is the prior distribution of $y$, and application of Bayes's rule produces a solution for the inverse problem in the form of the posterior distribution, $Q$, of $y \sim Q x_1, \ldots, x_n$. Compare this formulation with the treatment of calibration via conventional regularization in Hagwood [1992].

**Reply:** Our point is that for a given measurand and a given realization of the measurement error, the measured signal is, putting quantum effects aside, which can be embraced by the measurement error, unambiguously determined. Conversely, for a given measured signal, the most likely or most probable value of the measurand is **not** unambiguously determined. Since traditionally a mathematical function (which is the theoretical core of the measurement model) maps an independent variable $x$ unambiguously to a dependent variable $y$, while the opposite direction may be ambiguous, we find the GUM (2008) notation counter-intuitive.

**Action:** We have inserted: "Many conceptions exist of measurement models, which relate the measured value to the true value, and depending on the context, one can be more adequate than another (Possolo, 2015). GUM recommends a model that conceives the estimate of the true value of the measurand as a function of the measured value. Since in remote sensing of the atmosphere

multiple atmospheric states can cause the same set of measurements, and the measurement function thus would be ambiguous, we prefer a different concept, as outlined in the following. Further, we have deleted the section on Bayesianism.

**Comment: 662** *paradoxes shatter the bedrocks of Bayesian philosophy, namely the likelihood principle that says that all relevant evidence about an unknown quantity obtained from an experiment is contained in the likelihood. Others accept the theoretical validity of the Bayes theorem but challenge its applicability in real life because of the unknown and unknowable prior probabilities.*

The paradoxes alluded to often relate more to the adoption of so-called "non- informative" prior distributions than to the acceptance of the likelihood principle, as Cox [2006] points out, in a contribution referenced by White [2016].

All theories of inference have given rise to paradoxes, and nevertheless most often they produce valid and practically useful inferences. Regarding the likelihood principle in particular, at least one well-known "paradox" has been dismissed as a false alarm [Goldstein and Howard, 1991].

In any case, White [2016] does not come even close to suggesting that such paradoxes "shatter the bedrocks of Bayesian philosophy," in particular as applied in measurement science. I know for a fact that Rod White does not object to the use of Bayesian methods when these are warranted and there is genuine prior information that should be taken into account.

The objection, which is also raised by Bayesians [O'Hagan, 2006], is to the systematic reliance on "non-informative" prior distributions just for the sake of going through the motions of the Bayesian machinery or to pay lip service to scientific objectivity.

The Bayesian approach to problems of statistical inference is a choice among many that can be made, similarly to how some people choose to drink lemonade and others bourbon. Different approaches to statistical inference (be they frequentist, fiducial, or Bayesian) all can claim notable successes in solving problems of practical importance.

Bayesian methods, in particular, can boast a long and varied roster of accomplishments that prove beyond reasonable doubt that they are applicable in real life, and that they can be used to solve important practical problems, and that often they do so better than non-Bayesian alternatives [O'Hagan, 2008].

A particularly striking, recent accomplishment of Bayesian methods concerns the use of measurements of $\Delta 14\,\mathrm{CO_2}$, in conjunction with atmospheric transport models, to demonstrate that several bottom-up approaches to the estimation of national inventories likely underestimate U.S. fossil fuel $\mathrm{CO_2}$ emissions [Basu et al., 2020].

This study, which is based on methodological advances published in this very journal [Basu et al., 2016], includes rigorous, model-based

uncertainty evaluations, and also serves to show that the GUM and its supplements have much catching-up to do if they will ever come to play a role in addressing momentous issues like the measurement of greenhouse gas emissions. The suggestion that Bayesian methods are questionable because prior distributions are "unknown and unknowable" reveals a misconception about prior distributions: they are meant to encapsulate the knowledge that someone has about the quantity of interest, prior to performing an experiment that generates fresh information about it. Therefore, proper, informative, subjective prior distributions are known to who formulates them, by construction.

Of course, the Bayesian can be much mistaken and construct a prior distribution that reflects an erroneous conception of reality, in which case the "knowledge" that the prior encapsulates is false knowledge and its use will lead the inference astray. However, Bayesian methods cannot be blamed for delusions any more than Newton's laws can be blamed for accidental falls.

**Reply:** We do not deny the benefits provided by the Bayesian toolbox but we reported related critical issues. The aim was to provide an argument why we consider it as inadequate to commit the community to a full-blown Bayesian philosophy. The context of this argument are the question (a) if the "Bayesian turn" can explain the alleged difference between the error concept and the uncertainty concept, and (b), if so, if it is adequate to anchor Bayesianism in a normative document. Our reply to both these questions is negative. However, since the current version of GUM is not fully Bayesian, we now consider the section on Bayesianism as irrelevant for the title topic.

**Action:** The section on Bayesianism has been deleted.

**Comment: 674** *the Bayesian philosophy relies on a couple of unwarranted assumptions, e.g., the likelihood principle and the indifference principle.*
The authors convey a wrong impression on both counts. Adherence to the likelihood principle is a choice that, in most applications, turns out to be a better choice than most alternatives. Still, it is only a choice, among many that can be made. Making such choice is necessary but not sufficient to be Bayesian. Many statisticians, physicists, chemists, and biologists adhering to the likelihood principle are not Bayesian.

Neither is adopting an indifference principle (or, more generally, using an allegedly non-informative prior distribution) necessary to qualify as being Bayesian.

In fact, quite the contrary is true: reliance on proper, informative, and suitably elicited subjective prior distributions, are the hallmarks of genuine Bayesian practice. But this, too, is only a choice [Robert, 2007].

**Reply:** We did not want in this paper to argue about which to positions make a user of the Bayesian toolbox a "real Bayesian" and which of the unwarranted assumptions belong to the bedrocks of Bayesian philosophy. We also did not want to judge if Bayesianism is good or bad. Instead, our point was that a philosophy that is challenged by a non-negligible part of the community shall not be made generally binding. However, since GUM itself (contrary to some of its readings) does not promote Bayesianism, we now consider this issue as irrelevant for the paper.

**Action:** As said above, the section on Bayesianism has been deleted.

**Comment: Technical Corrections**
**025 Replace comes down to the question if and how with "comes down to the question of whether, and if so how"**

**Reply:** Agreed.

**Action:** Changed as suggested.

**Comment: 056 Replace Second we assess to which degree with "Second, we assess the degree to which"**

**Reply:** Agreed.

**Action:** Corrected.

**Comment: 068 Replace** *we conclude to which degree* **with "we conclude the degree to which"**

**Reply:** Agreed.

**Action:** Corrected as suggested. In the same spirit we have changed "to which degree the measurement is thought..." to "the degree to which the measurement is thought..."

**Comment: 128 Replace A rich methodical toolbox with "A rich methodological toolbox"**

**Reply:** Agreed

**Action:** Corrected.

**Comment: 180 Replace** *This argument is often used to dispraise* **with "This argument is often used to disparage"**

**Reply:** Agreed but no longer relevant.

**Action:** None. because in the shortened manuscript this statement does no longer appear.

**Comment: 267 Replace** *agent's believe* **with "agent's belief"**

**Reply:** Thanks for spotting!

**Action:** Corrected.

**Comment: 364 Replace** *Quantities of which the value cannot determined* **with "Quantities whose values cannot be determined." This suggestion deliberately ignores the antiquated invective against using the possessive whose for inanimate objects, consistently with the recommendation in O'Conner [2019, Page 243].**

**Reply:** Agreed; the Copernicus editorial team usually does a great job with respect to language issues, and we trust that they will remove any language-related inconsistencies and outdatedness.

**Action:** Changed as suggested.

**Comment: 373 Replace** *Others have been formulated by us, serving, as arguments of the Devil's advocate, as working hypotheses in order to moot the error and uncertainty concepts in the context of indirect measurements* **with "We have formulated others as Devil's advocates, which are intended to serve as working hypotheses to MOOT the error and uncertainty concepts in the context of indirect measurements," except that "moot" needs to be replaced by a word that is suitable for this passage: maybe "merge" or "reconcile", depending on what the authors wish to express.**

**Reply:** Agreed.

**Action:** The new sentence now reads: "We have formulated others as Devil's advocates, which are intended to serve as working hypotheses to critically discuss the error and uncertainty concepts in the context of indirect measurements."

**Comment: 413 Replace** *measurements are not in the focus* **with either "measurements are not in focus" or "measurements are not the focus," depending on what the authors wish to say exactly.**

**Reply:** Agreed, but obsolete.

**Action:** None, because this sentence does no longer appear in the revised version of the manuscript.

**Comment: 420 Replace** *the probability that a person suffering fever to have Covid-19 is 50%* **with "the probability is 50% that a person with fever has COVID-19"**

**Reply:** Agreed.

**Action:** Corrected as suggested.

**Comment: 429 Replace** *distribution which is missing* **with "distribution that is missing"**

**Reply:** Agreed but obsolete.

**Action:** None, because this sentence does no longer appear in the revised version of the manuscript.

**Comment: 470 Replace** *the aggregation of random uncertainties* **with "the aggregation of random uncertainties"**

**Reply:** Agreed, thanks for spotting.

**Action:** corrected as suggested.

**Comment: 612 Replace** *strong misjudgement* **with "strong misjudgment" (unless the British spelling be preferred)**

**Reply:** Agreed.

**Action:** Corrected: "misjudgment".

**Comment: 743 The sentence that includes** *traditional error analysis can connotate a statistical quantity* **is unclear, and should be rewritten, taking into account the fact that the verb connotate is obsolete and has been replaced by connote. However, a term more generally familiar would be preferable, like suggest, possibly.**

**Reply:** Agreed.

**Action:** "[...] that the traditional error analysis can **deal with** a statistical quantity, and that [...]"

---

## Author Comment (AC2)

**Review #2:**

**Comment: Summary:**
This work presents a thorough, although perhaps at times exhaustive, discussion on the definitions and distinctions of the terms 'error' and 'uncertainty' and how their viewpoint, namely that the two are the same, differs from that put forth in "Evaluation of measurement data – Guide to the expression of uncertainty in measurement", issued by the Joint Committee for Guides in Metrology, colloquially referred to as GUM. On the whole, the topic of this paper is somewhat subjective and more a matter of the definition of terminology. While I would, at first, question its appropriateness for this journal, I think the topic is useful if not necessary and find that this journal is likely the most appropriate option for its target audience. I mostly agree with the authors' viewpoint and rationale throughout but would argue that it likely should be reorganized and made more concise.

**Reply:** The authors thank the reviewer for their open- mindedness with respect to a topic that is somewhat off the mainstream of the usual AMT articles.

**Comment: General Comments:**
I find this paper unique amongst all those I have reviewed in the past in the sense that its premise borders almost on the philosophical rather than the purely scientific even though most of its arguments are rooted soundly in established mathematics and statistics. Much of the paper hinges on the authors' interpretation of GUM and addressing what they believe to be logical fallacies and inconsistencies. However, it is clear from reading both this work and the review of Referee #1 and his colleagues, which demonstrates an extensive familiarity with GUM, that there are differing interpretations of GUM. I must admit that I am only somewhat aware of the content of GUM and have not read it in its entirety, which unfortunately impedes my ability to thoroughly evaluate this paper. For this reason, I must defer much of the debate on the meaning / intention of GUM to what will be subsequent interactions between the authors, Referee #1, and the editor, but overall this supports an underlying premise of the paper that further clarification of the nomenclature is necessary.

**Reply:** We do agree that some of our interpretations are subjective, but since GUM is in some places somewhat ambiguous and leaves room for interpretation, this subjectivity seems unavoidable to us.

**Comment:** The authors go into detail in Sections 3 and 4 of what they interpret GUM is saying on the meanings of error and uncertainty. However, at the end of Section 4 (paragraph starting at Line 338) the authors more clearly state what error and uncertainty are,

though it is not clear if this is the authors' interpretation of what they believe GUM is actually saying (as opposed to what the authors interpret GUM intends to say) or if it is what the authors believe. If the former, it should be better clarified. If the latter, this seems rather subjective (as opposed to definitive) because to attribute any meaning to either of these words will always be a matter of opinion, even if a strict definition is likely necessary. If the authors wish to firmly establish their own definitions of error and uncertainty, then they must make this a clearer priority of this paper. Another alternative is that might be a good topic for a companion paper.

**Reply:** The paragraph starting with line 338 (in the original manuscript) describes hoe the terms 'error' and 'uncertainty' were understood in the pre-GUM language. Some confusion is generated by bringing this point not until here. In the restructured manuscript, we redistribute our arguments into a terminological section and a conceptual section. We start the subsection on terminology with the connotation of these terms in common language. Restructuring and rewriting hopefully has made this issue clearer in the revised version.

**Action:** The manuscript has been restructured and partly rewritten as described above.

**Comment: While Sections 5.2 and 5.3 are interesting, they do not appear to be necessary to make the authors' main arguments pointing out the inconsistencies in GUM. However, as mentioned previously, if the authors wish to firmly establish their own definitions of error and uncertainty, particularly as they apply to remote sensing and inversion schemes, then these sections become more useful.**

**Reply:** Section 5.2: In the restructured version of the manuscript, we make more clearly that we conceive measurement (along with its evaluation) as an inverse process, and we put all the relevant arguments, which were distributed over the paper in the original manuscript, in one subsection. We trust that this action has made our point clearer, and that it is now more obvious, why the base-rate-fallacy issue is relevant.
Section 5.3: If the relation between the value of the measurand and the measured signal is nonlinear, the error in the estimate of the true value depends on the true value. We still think that this is a serious challenge for the claim that uncertainties can be estimated without referring to the true value. Our rewriting of Section 5.2 should have paved the way towards a better understanding of the relevance of (old) Section 5.3.

**Action:** The manuscript was restructured and partly rewritten.

**Comment: Normally I would avoid addressing the content of another referee's review, but given the extensive nature of the review**

it seems appropriate to avoid being repetitive and ease the subsequent burden on the authors to respond. Additionally, while I have some opinions on the philosophical debates on frequentists versus Bayesians and other topics within this work, it does not appear that my opinions on the matter are as strongly held as written in either the paper or the review so I will abstain so as not to add complication to the review process. However, I would like to chime in on the issue of "systematic" versus "random" uncertainties. I understand the Referee's point-of-view regarding the nomenclature (i.e., preferring "persistent" and "volatile"), and while technically more accurate it might be simpler and more relatable to the general reader to stick with the more familiar terms. Additionally, I would like to second the authors' desire to try to classify uncertainties in this fashion (i.e., the "classical" approach according to Colle as discussed by Referee #1) as it is important for the user to know the nature of the associated uncertainty.

**Reply:** We agree that the terms "systematic error" and "random error" should not be superseded by a new terminology.

**Action:** We stick to the established terminology but we add a descriptive statement that systematic errors are persistent and that random errors are volatile.

**Comment: I suppose this is somewhat expected given the nature of the topic of this work, but in general I find the content to be filled with jargon and philosophical tangents. I would imagine that the average reader that may not have an extensive knowledge of statistics might find this paper to be unapproachable. It would be in the authors' best interests to cut unnecessary content (e.g., many of the tangents regarding the comparisons between frequentists and Bayesians outside of Section 6.4) to the extent possible to both shorten the paper and appeal to as wide an audience as possible, particularly if they choose to further discuss their own preferred definitions of error and uncertainty.**

**Reply:** We agree that the original version of the paper contained unnecessary material that made it hard to read.

**Action:** We have restructured and shortened the paper and have removed unnecessary formalism.

---

## Author Comment (AC3)

**Review #3:**

Comment: Manuscript title: Truth and Uncertainty. A critical discussion of the error concept versus the uncertainty concept

In this manuscript the authors present an argument that the the the "error concept" and the framework put forth by the GUM (deemed the "uncertainty concept") are the same. The major issue I understand that the authors take with the GUM is in the recommendation that uncertainties reported with estimates of measurement need not be specifically interpreted with respect to errors and "true values." The authors refute this claim by seemingly arguing that uncertainties or "estimated errors" cannot be interpreted without reference to true values and therefore the concepts must be the same. To be honest, it was extremely difficult to parse through the unnecessarily lengthy 23 pages of text to come to the understanding that this is (I believe) the authors main argument, and critically I do not believe this argument is effectively made. In general the manuscript is too long with repetitive sections that are often confusing and in some places contradictory.

**Reply:** We agree that the structure of the manuscript was not sufficiently clear and contained unnecessary repetition.

**Action:** We have restructured and partly rewritten the paper in order to make the arguments clearer, to avoid repetition, and to shorten the manuscript.

Comment: The message is often lost in unnecessary language arguments between the GUM and the authors' definition of 'error' and in generally narrow and misconceived discussions about frequentist vs Bayesian statistical methods.

**Reply:** We agree that language arguments and conceptual arguments should not be merged. For the frequentist vs. Bayesian discussion, see below.

**Action:** The manuscript has been restructured. Terminological issues and conceptual issues are now discussed separately.

Comment: More seriously, the language used in reference to statistical concepts is imprecise and in some areas completely incorrect. The authors should define their terms with equations where applicable and adhere to commonly accepted mathematical/statistical/probabilistic definitions. In several places the statistical interpretations of their "error estimates" in relation to "true" values are overly simplified and likely to be misinterpreted, in particular when models are misspecified.

**Reply:** Often the point is not how quantities are mathematically specified but

how the involved terms are interpreted, i.e., to which entities in te real world the quantities refer. For example, the variance of a frequentist pdf and the variance of a pdf representing personal belief are calculated according to exactly the same formalism, but they connote an entirely different thing.

**Action:** During rewriting, we have taken care to use an unambiguous language.

**Comment: Rather than focusing on an argument that the uncertainties typically reported under the "error concept" can be also be interpreted as under the "uncertainty concept", they seem to miss the point of GUM (as I interpret GUM, but I would also argue more broadly the understanding of these concepts in the field of statistics) that reported uncertainties need not come with inferential statements about how close estimates are to the true value (e.g. actual errors) to be useful for comparison to other estimates.**

**Reply:** If the purpose of uncertainty reporting is not to provide an idea how close the estimate is to the true value, then we do not understand what the purpose of uncertainty reporting is. And even worse: Then we need a different concept that provides this. An estimate how close the estimate is to the true value is exactly what the data user needs.

**Comment: Instead the focus seems to be mainly a language argument that "true value" is the same as "value of the measurand" and so the concepts must be the same – a not particularly useful argument in my opinion.**

**Reply:** We disagree. The bulk of our argument is conceptual, not terminological.

**Action:** In the revised version, we separate language arguments and conceptual arguments. The terminological part is only a small fraction of the paper.

**Comment: Without considerable revision and restructuring I do not believe the manuscript provides a useful contribution to AMT. In fact, I am concerned that publication in its present form would propagate dangerous misconceptions about statistical methods and uncertainty quantification to the community.**

**Reply:** We agree that the original structure did not optimally support our argument.

**Action:** The paper has been restructured and partly rewritten.

**Comment: In addition, the authors do not provide a concise, understandable overview of the "error" and "uncertainty concepts" and**

the supposed differences, which narrows the manuscript's audience to those who are already well-versed in the GUM and the specific error analysis framework the authors consider.

**Reply:** It is exactly the main problem of GUM that they state that the error concept and the uncertainty concept are different but fail to clearly specify in what the difference consists. This is what we criticize. Since GUM is vague with respect to what the difference is, we posit working hypotheses what the difference might be.

**Action:** In the revised version we hope to have made our argument form – to posit working hypotheses and to refute them – clearer.

**Comment: I do agree with the authors that the quantitative methods laid out under the GUM framework are not inconsistent with the traditional error analysis framework and, if properly understood, the interpretations of such quantities under both frameworks are generally in agreement. It is my belief that a useful manuscript would argue these points very concisely, showing that the recommendation in GUM are not inconsistent with traditional methods in the atmospheric remote sensing community, and would focus more attention on addressing how the GUM principles apply to atmospheric retrievals and where GUM may fall short.**

**Reply:** Our conclusion is that we do not see relevant differences between the error concept and the uncertainty concept. Thus, the agreement of the interpretations follows a fortiori. Problem areas where GUM falls short were already discussed in Section 6 of the original manuscript.

**Action:** The Section on the applicability of GUM-2008 to remote sensing of the atmosphere (Section 4 in the revised version) has been partly rewritten, and has become shorter and more focused in the course of restructuring the paper.

**Comment: General comment about "true values" and uncertainties The authors need to clearly state what they mean by "true value" in their arguments. Specifically, when discussing true values are they referring to the truth in terms of reality (if such a quantity exists)...**

**Reply:** Of course we cannot prove the existence of an external truth. However, the purpose of the entire endeavor of taking measurements is simply to create a link between our mind and the external reality. If we deny the existence of an external reality, then we need no measurements. Since the journal AMT has the term "measurement" in its title, we think that the existence of an external reality is uncontroversial among the AMT readership. In the spirit of reviewer #2, who sees the risk that the paper drifts too far towards philosophy, we prefer

not to dwell on this issue in the paper[1].

**Comment: ... or the true value in terms of the specified statistical model and resulting theory? The latter are the only "true values" that have any statistical guarantees in the interpretation of uncertainty estimates and are only equivalent to the true value in reality if the statistical model perfectly describes the true data generating process (i.e. is the "correct" model), which we know is unlikely to be the case particularly in atmospheric remote sensing retrievals.**

**Reply:** Among all concepts of truth (logical truth, analytical truth, factual truth, see R. Carnap, New York, 1966 "Philosophical foundations of physics", for a deeper discussion on this), only the latter, the factual truth, is relevant. Among the theories of factual truth, the correspondence theory is the most intuitive one, and is probably accepted by most empirical scientists. Correspondence theory follows largely the pattern "The statement 'snow is white' is true if and only if snow is white". However, we see no conflict of our paper with the coherence theory of truth. In order to avoid disgressions and to drift away too far towards philosophical aspects not relevant to the paper, we prefer not to include in the paper the discussion of different theories of truth.
We think that in the context of measurements it is sufficiently clear that the true value refers to reality. We are not aware that in the context of measurements the term "true value" is used for anything else. Values resulting from a specified statistical model and resulting theory are not usually called true values. The fact that models usually do not fully represent reality is exactly the reason why in the TUNER project the need of the consideration of model errors is highlighted. Since in our paper the true value bears the attribute "unknowable", it should be clear what the connotation of "true value" is in our paper. The attribute "unknowable" would simply make no sense for any other connotation of the term "truth".

**Action:** To remove all residual ambiguity, we have changed in the introduction "the true value of the measurand" to "the true value the measurand has in reality". For the sake of shortness, we avoid any discussion on what 'reality' is.

**Comment: As an example, consider maximum likelihood estimation for atmospheric remote sensing retrievals. Measured radiances y are assumed to be generated from a true state of the atmosphere x through a "true" radiative transfer function f , and the true data generating process may be idealized as**

$$y = f(x, b) + \epsilon$$
* * *
[1]Strictly speaking, the solipsist denial of external truth would make the reviewing of papers absurd. If a paper to be reviewed would not be part of the external truth but a product of the reviewer's thinking, then it remains unintelligible why the paper needs to be reviewed.

assuming correctly specified Gaussian random errors, $\epsilon$. In practice, as the authors point out, f is not fully known and is replaced by a function $F$ that represents the radiative transfer function to the best of the scientists knowledge. In this case, the observation equation used for inference is

$$y = F(x, b) + \epsilon$$

and the statistical model is "misspecified" in relation to the true model. Under both models, the radiances are assumed to be generated from the specified distribution with unknown true state $x_0$, but crucially these "true values" are not the same under both models!

**Reply:** Not quite. $F$ and $f$ will not produce the same $y$, even when applied to the same $x$. There is only one true $x$. Results inferred from $y = F(x, b) + \epsilon$ are estimates of $x$ but not the true $x$. The true $x$ is not model dependent.

**Comment:** The MLE, $\hat{x}$, has the interpretation of the value of $x$ such that the model (correct or misspecified) generates radiances most similar to what is observed (i.e. given $\hat{x}$ the observed radiances are most probable). Assuming regularity conditions hold and reasonably large sample size, the sampling distribution of $\hat{x}$ is approximately Gaussian with mean equal to $x_0$ (under the model) and the standard deviation represents an estimate of the expected deviation of the estimate from that true value, $x_0$. Under the correct model, $x_0$ is interpreted as the true state of the atmosphere, but under the misspecified model $x_0$ is the value of $x$ that minimizes the difference between the true data generating model and the misspecified model. The degree to which this true value matches the true target depends on the degree of misspecification which is not known.

**Reply:** Up to this point we agree.

**Comment:** Therefore, statements about unknown true values (reality) based on misspecified models ("all models are wrong") are inferential and conditional on all of the assumptions and uncertainties in the measurement system. I do not read GUM as dispensing with the concept of the true value, I understand GUM to recommend that when reporting uncertainties associated with estimates of a value of a measurand (GUM agrees "value of a measurand" can be synonymous with "true" value of the measurand) it is not necessary to make inferential statements about actual errors specifically when reported uncertainties are meant to be used to assess reliability/consistency with other measurement systems. That is, if I have two different measurement frameworks providing interval (uncertainties) of plausible values of a measurand, these intervals can be used to compare consistency with each other without needing to know the "true value."

**In this case, it is only necessary to describe uncertainty estimates as summarizing a range of estimates that would also be plausible for the measurand under the measurement system, which is still consistent with quantities reported under the "error concept." Consider an uncertainty (or 'error') estimate, $\sigma_x$, related to parameter/measurand $x$, e.g. the standard deviation of a sampling distribution of $\hat{x}$ (frequentist) or the posterior standard deviation of the posterior $p(x|y)$ (Bayesian).**

**Reply:** The conventional error estimate may also include systematic effects. Thus, $\sigma_x$ is not necessarily the standard deviation of a sampling distribution of $\hat{x}$. It has long been recognized that the sample standard deviation covers only the random part of the total error.

**Comment: Under a frequentist approach, $\sigma_x$ describes how much the estimate is expected to vary around its statistical expectation $E(\hat{x})$ and represents the spread of values that would also be plausible values of the estimator if the experiment were to be repeated, given the same assumptions in the measurement system. Under a Bayesian paradigm, $\sigma_x$ describes variability around the posterior mean and provides information on the spread of plausible values (estimates) of the measurand that are also consistent with the scientists' knowledge given the observations, assumptions and prior knowledge. Of course, you can argue that $\sigma_x$ also describes the spread of the "error distribution" $\hat{x} - x_0$ but this doesn't describe the expected magnitude of actual errors, the mean of the "error distribution" $E(\hat{x}) - x_0$, unless the estimator is unbiased (see related comment about MSE vs variance below). Given this, I do not understand what the authors' issue with this GUM recommendation is, unless they are simply arguing that "value of a measurand" also means "true value of a measurand" (that the GUM agrees with) in which case I see this as quibbling about words and not addressing the larger concept of whether it is necessary to make inferential statements of the form e.g. "95% confident that the true value is within some interval" when reporting uncertainties.**

**Reply:** We broadly agree, and the whole dispute seems to be based on the misunderstanding that we take the sample standard deviation as the estimate of the total error. Instead, our error estimate is the standard deviation of the density function representing the estimated total error.
Our main point is, that GUM is not clear about why it is a problem that the true value of the measurand is unknown and unknowable. Without this piece of information, however, it is unintelligible what the difference between an estimated error and uncertainty is. Many interpretations are possible, yours is one of them. In the literature we quote, we find some more. We formulate working hypotheses how the unknownness of the true value might affect error/uncertainty estimation and discuss them. We still do not see what should be wrong with this approach.

Interestingly, GUM explicitly supports level-of-confidence approaches (e.g. p. viii). Thus the dismissal of these cannot be quoted as a key difference between the error concept and the uncertainty concept.

**Action:** Terminological and conceptual issues have been separated in the revised version.

**Comment: Additional comments**
**1. The authors spend several pages (sections) arguing that in addition to the universally accepted statistical definition of error as the difference between measured/estimated and the "truth", a second definition of the word error be accepted (deemed 'error') to refer to statistical estimate of the expected differences between the observed/estimated and true value. This secondary 'error' definition proves confusing in multiple places as it is unclear to which error the authors are referring to, be it actual error or 'error', thus inadvertently making an argument for GUM's choice of separation in language of uncertainty estimates and actual errors.**

**Reply:** We have provided considerable evidence of the use of the term "error" as a statistical quantity in the literature. These examples prove that the "statistical definition" is not so universally accepted as the only meaning of the term "error". We do agree that the implied equivocation may in some cases cause confusion.

**Action:** In the revised version we take care not to use the term 'error' without the attributes 'estimated' (for the unsigned statistical estimate) or 'actual' (for the realization of the respective random variable), where relevant. This removes all ambiguity.

**Comment: In general, the arguments about language definitions of "uncertainty" and "error" could be summarized much more concisely in about a paragraph, acknowledging that the GUM definition of 'uncertainty (of measurement)' encompasses the same quantities that have have often been shorthandedly referred to with reference to the word error as "error estimates", "error bars", etc. Therefore, I think large portions of sections 3 and 4 are repetitive and could be removed.**

**Reply:** We agree that the original version of the paper was repetitive.

**Action:** Terminological issues have been separated from the conceptional stuff and the related text has been shortened and partly rewritten.

**Comment: 2. Page 1, lines 10-11: I find the definition of 'error' as des-**

ignating a statistical estimate of the expected difference between the measured and the true value of a measurand to be not in agreement with standard deviations they later reveal are often use as "error estimates" in remote sensing retrievals (e.g. section 5.3). The authors definition is consistent with statistical summaries of error like root mean squared error (RMSE) which estimates the square root of the expected squared difference of actual errors, or median absolute difference the mean of the absolute value of actual errors. The variance of an estimator is only theoretically equal to the MSE if the estimator is unbiased, and even in that case the variability is around the true model parameter of a potentially misspecified model, not necessarily reality. Any inferential statements about the true value in reality and distributions of actual errors are conditioned on all assumptions and uncertainties in the measurement system being reasonably correct. The authors need to clarify their language in regards to what they mean by 'error', "true values", and how these definitions apply to the uncertainty estimates they reference later. Otherwise I am concerned that there is a serious underlying misunderstanding of how to interpret uncertainties they report.

**Reply:** When we write 'standard deviation' we do not mean a sample standard deviation. In accordance with GUM we conceive the standard deviation as a characteristic of a distribution representing subjective probabilities without any link to multiple measurements. These standard deviations can (and should, if supposed to characterize the total error) – also in agreement with GUM – also include the effects of the systematic effects and model deficiencies, as specified, e.g., in von Clarmann et al., Atmos. Meas. Tech., 13, 4393-4436, https://doi.org/10.5194/amt-13-4393-2020, 2020.
In the criticized Section 5.3, however, the covariance matrices explicitly represent only isolated components of the total error budget. Thus the criticism seems not applicable here.

**Action:** To avoid such kind of misunderstanding, we have added a footnote, "When we use variances and standard deviations, we do not mean sample variances and sample standard variations but simply the second central moment of a distribution or its square root. In accordance with GUM-2008, this distribution can represent a probability in the sense of personal belief, and thus can include also systematic effects.".

**Comment: 3. Page 1, lines 10-11:** This is also the first place in the paper where the failure to use consistent mathematical notation is problematic. Consider the simple statistical model

$$X = \mu + \epsilon,$$

where $\epsilon$ is a random variable representing actual measurement error. What the authors contend is 'error' could be written, $E(X - \epsilon)$ where

$E()$ denotes the statistical expectation and $\mu$ is the "true value" of the measurand. This quantity is equivalent to $E(e)$ and would represent measurement bias.

Or do they mean this to represent var$(\epsilon)$, that is $E(\epsilon - E(\epsilon))^2$ ? Or instead do they intend to refer to the same manner of quantities but with respect to an estimator of $\mu$ given a set of observations of $X, x_1, \ldots, x_n$, say $\hat{x}$? The latter would be consistent with what the authors presents in Section 2, but it would help immensely if the authors provided some manner of illustrative model, and used it to clarify their ensuing arguments.

**Reply:** We refer to neither of the three mentioned quantities. The options suggested in the review rely on sample standard deviations from multiple measurements. We understand that the estimated error is the square root of the second moment of a distribution that characterizes the personal belief of an agent. This is in accordance with GUM.

**Action:** Part of the problem should have been solved by the footnote mentioned above. We further add: "The estimate of the total error includes both measurement noise and all known components of further errors, random or systematic, caused by uncertainties in the measurement and data analysis system"

**Comment: 4. Page 1, lines 14-16: I do not believe GUM presents a "contrasting" definition of the term error. GUM presents the universally accepted statistical definition of error, and defines "uncertainty" to quantify the spread of plausible values given uncertainties in the system.**

**Reply:** Agreed that 'contrasting' is not the adequate term. We disagree that the GUM definition is universally accepted as the only meaning of the term 'error'. The literature we quote furnishes evidence of the contrary.

**Action:** 'Contrasting' replaced by 'narrower'.

**Comment: What do the authors mean here by "measurement error" that the term "measurement uncertainty" is replacing? $Var(\epsilon)$?**

**Reply:** We do agree that the term 'measurement error' is misleading because TUNER is interested in the errors/uncertainties whatsoever of the retrieved value $\hat{x}$, while the term 'measurement error' can be understood as errors in the measured signal $y$. But the mathematical definition does not help us here, because the quantity $Var(\epsilon)$ can mean two very different things. Conceived as sample variance it would include only the random, or volatile, part of the error, while within the concept of a personal-belief-probability it would include also the systematic (persistent) parts.

**Action:** In the introduction, "measurement errors in satellite date" is replaced by "errors in estimates of atmospheric state variables retrieved from satellite measurements". Due to the restructuring of the manuscript, the statement at issue has been moved to new Section 2.1, after a statement that the estimate of the total error includes both measurement noise and all known components of further errors, random or systematic, caused by uncertainties in the measurement and data analysis system.

**Comment: Then, on page 3, lines 75-76, the authors refer to $\epsilon$ as the actual "measurement error" in the y-domain. Is this the same reference to measurement error as in line 16 or there is 'measurement error' meant to refer to the variance or standard deviation of the actual measurement errors ($\epsilon$), If the latter, this inconsistency makes more of an argument for GUM's separation of language definitions of 'error' and 'uncertainty' than for the authors' definition.**

**Reply:** In the original version of the paper, $\epsilon$ was consistently used for the actual error in the y-domain. A problem in the old version was not so much that there was an ambiguity between error as a statistical description of a random variable and error as an actual realization of a random variable. This was quite clear from the context. Instead, the problem was that we used the term 'measurement error' both for the error in the measured signal $y$ and the error in the inferred state variable $x$. GUM does not help here, because GUM is about direct measurements where the x-domain and the y-domain need not to be distinguished.

**Action:** On suggestion of Reviewer #1 the formal part (where the epsilons appeared) has been shortened, and the statements at issue do no longer appear. Further, we use, wherever relevant, the terms error only with attributes 'estimated' or 'actual'. And we have taken care to reserve the term measurement errors to general contexts, for direct measurements, or in the case of indirect measurements in the $y$-domain but not for errors in estimates resulting from analyses invoking inversion.

**Comment: 5. Page 1, lines 23-24: The authors state, "The claim is made that the uncertainty concept can be construed without reference to the unknown and unknowable true value while the error concept can not." What specifically is the error concept to which they are referring?**

**Reply:** This is an excellent question that GUM fails to answer. Our paper is an endeavour to find out what they mean.

**Comment: I read GUM as saying "errors" (as defined as actual measurement errors) cannot be construed without knowledge of the true value (in the example above, $\mu$), meaning that value of a realization**

of the random variable $\epsilon$ in the example above can't be known because $\mu$ is unknown, and analogously the resulting difference between and estimate $\hat{x}$ and $\mu$. However, the parameters of the distribution of $\epsilon$ (describing its mean and variance for example) can be discussed, reasoned about, and even estimated from data with some additional assumptions. How does the claim in the following sentence (lines 25-26) follow from this?

**Reply:** It is trivially true that the actual error cannot be inferred because the true value is not known. But this is not what conventional error estimation aimed at. Conventional error estimation has always aimed at providing statistical error estimates. This is why we distinguish in the revised version of the paper between terminological issues and conceptual issues. The claim in (old) lines 25-26 is quoted directly from GUM.

**Action:** Terminological and conceptual issues are now discussed separately.

**Comment: 6. Page 2, lines 25-26: The authors state that the dispute comes down to "the question if and how the error (or uncertainty) distribution is related to the true value of the measurand." Again, here is where imprecise terminology is confusing. By "error distribution" I would assume they mean the distribution of the random measurement errors $\epsilon$, but what do they mean by "uncertainty distribution" (or do they mean that the word "uncertainty" is now a synonym for "error distribution")? So the dispute is about the relationship between $\epsilon$ and $\mu_0$? How? Or by error distribution do the mean some distribution of the estimator - the truth, e.g. $\hat{x} - \mu_0$?**

**Reply:** Again, the misunderstanding arises because the reviewer conceives 'distribution' in a frequentist sense, as obtained from a sample of multiple measurements. We conceive 'distribution' in a wider sense (as GUM does!) as a probability distribution where the probability represents the degree of belief of an agent.

**Action:** The footnote mentioned above should solve this issue.

**Comment: 7. Lines 24-27: The distinction between "error" vs "uncertainty" statisticians is artificial, I am not aware of any such distinction nor do I believe any such dispute or "rift" along these lines exists in the statistical community (I am a practicing statistician). Please cite a reference for the existence of this rift, if you have one.**

**Reply:** As agreed with reviewer #1, we no longer use these terms.

**Action:** The terms 'error statistician' and 'uncertainty statistician' do no longer occur in the revised manuscript. We no longer refer to this rift and have toned

down our statements made in this context.

**Comment: 8. Line 128: It has yet to be clearly stated what the authors define to be the debated difference between "error estimation and uncertainty assessment." A concise definition of both and the argued against definitions at the beginning of the paper would greatly improve the presentation.**

**Reply:** We disagree. The missing definition what error estimation is (in contrast to uncertainty estimation) is exactly what we criticize in GUM. Our paper tries to find out what this difference might be. We find no relevant difference in the concepts (only in the terminology).

**Action:** In order to avoid misunderstandings we now state that the total error includes both measurement noise and all known components of further errors, random or systematic, caused by uncertainties in the measurement and data analysis system.

**Comment: 9. Line 138-148: I can only assume the second meaning of the term 'error' the authors are referring to are shorthand statements that have been historically made in the literature such as "the estimated error of quantity of interest is X." These statements typically use terms like "estimated error" to represent a quantity like a standard deviation of a sampling distribution or a posterior standard deviation, and it is assumed that the reader/community understands this implicit definition (and that is does not provide information about the actual truth without inference). Again, I do not find the argument over whether this specific quantity should be referred to as "uncertainty" or "estimated error" to be particularly compelling, but rather a discussion of the interpretation of these quantities seems to be needed.**

**Reply:** We think that our separation of the manuscript in a (shorter) section on terminological issues and a (longer) one on conceptual aspects solves this issue. Further, we have made clear that the estimated error, as we use this term, does not only include the random part of the error that shows up in the sample standard deviation.

**Action:** Manuscript reorganized as described above.

**Comment: 10. Line 190: I do not see how this is not quibbling about words. What does referring to "uncertainty" under the GUM definition as 'error' under the error concept provide that the GUM definition does not? Other than what seems to be a generally misused definition regarding the "truth" in the authors' definition of 'error'. I think the authors would agree that the two meanings of error set**

**forth in this manuscript refer to different concepts.**

**Reply:** We do not see where a misused definition of "truth" comes into play. The two meanings of error refer to the same concept. The error is a random variable, and the term 'error' is, depending on the context, used for a specific actual realization as well as for a statistical characterization of its distribution. There is no concept that tries to estimate the error by subtracting the measurement from the true value.

**Action:** Terminological issues are now confined to one section, the remainder of the paper is on conceptual issues.

**Comment: 11. Line 199: I cannot find reference to "error distribution" or "uncertainty distribution" in GUM. The definitions provided here are consistent with a sampling distribution of an estimator ("error distribution") and a posterior distribution ("uncertainty distribution"). Is this what is intended? If so, please adhere to well-defined statistical definitions. If not, please clarify.**

**Reply:** In general, we mean with 'error distribution' the probability distribution in terms of degree of belief, as endorsed by GUM on p. 57. However, in this particular context the term 'distribution' is indeed unnecessary.

**Action:** "...how the error (or uncertainty) distribution is related..." has been replaced by "how the measured or estimated value along with the estimated error (or uncertainty) are related ..."

**Comment: 12. Lines 233-239: I see no reason why uncertainties reported as in GUM, along with assumptions of the statistical model, cannot be used for hypothesis testing. In (frequentist) hypothesis testing an assumption is made about the true state, in which case the truth is assumed known and inference is made based on how reasonable this assumption is given the variability (uncertainty) of plausible estimates under the measurement system. If the assumed value of the truth is outside what the scientist believes to be plausible based on their understanding of the measurement system and uncertainties, then a decision is made that the hypothesized value is unlikely to be the true value. A Bayesian hypothesis test would argue whether or not an assumed value or range of values for the parameter are consistent or not with posterior knowledge (uncertainties).**

**Reply:** If the uncertainty does not state a statistical relation between the measured state and the true state, then it cannot be judged how consistent the measurement is with the assumption. The wording "what the scientist believes to be plausible" brings in the concept of the true value through the back door. And if the uncertainty provides a (statistical, estimated, whatsoever) relation

between the measured value and the true value, what is then the difference between the uncertainty concept and the error concept?

**Comment: 13. Line 265: Again, what is meant by "error distribution"?**

**Reply:** The criticized sentence does no longer appear in the revised version.

**Comment: 14. Lines 317-318: Monte Carlo uncertainty estimation, however, is in its heart a frequentist method, because it estimates the uncertainty from the frequency distribution of the Monte Carlo samples. This statement is fundamentally false. Monte Carlo methods are simply methods to solve numerical problems through sampling and are used in both frequentist and Bayesian statistics.**

**Reply:** Monte Carlo methods realize probability distributions as frequency distribution and finally interpret resulting frequencies as probabilities. That is all we intended to say. But since this argument is not necessary for our case, we have decided to remove it.

**Action:** This argument has been deleted.

**Comment: 15. Section 5.1: The GUM definition of "uncertainty" does not dispense with reference to the measurand only to its true value. To this end, GUM is consistent with the authors statement we conceive the definition of a quantity and the assignment of the value to a quantity as quite different things. In general this section reads more as a language "gotcha" argument against the GUM's use of the term operational definition rather than in a useful argument about the definition of uncertainty, and as such I'd suggest omitting.**

**Reply:** In (old) Section 5.1 (new Section 3) we try to understand why the fact that the true value of the measurand might cause problems for error estimation. We find that it is a legitimate question, how and why the unknownness of the true value might cause problems. The answer is not as trivial as one might think, because, although according to GUM the only meaning of the term "error" is the actual difference between the measured value and the true value of the measurand, the conventional concept of error estimation has never been to try to calculate this actual difference.

**Action:** During the restructuring/rewriting we have tried to make our argument clearer.

**Comment: 16. Section 5.2: This section should be omitted. It presents incomplete and oversimplified interpretations of Bayesian and frequentist methods that are distracting to the manuscript.**

**Reply:** This section is not primarily about Bayesian vs. frequentist methods. The Bayes theorem is accepted both by frequentists and Bayesians and can be inferred directly from the Kolmogorov axioms, without invoking any particular interpretation of probability. We find our base-rate-fallacy argument essential. It is most probably the ONLY cogent argument why error bars around the estimated value must not be considered as descriptors of a pdf that tells one the probability of a given value to be the true value.

**Action:** The section has been completely reorganized. We have instead deleted the Section "Bayesian versus non-Bayesian", because the current GUM is not clearly Bayesian; only some of its interpretations are.

---

## Referee Report (RR1)

**AMT-2021-157: 2nd Review**

Antonio Possolo

January 15th, 2022

**1 Summary**

The article under review, *Truth and Uncertainty. A critical discussion of the error concept versus the uncertainty concept*, by Thomas von Clarmann, Steven Compernolle, and Frank Hase, deserves to be published because it is provocative and questions what has become uncritically "accepted wisdom" among many who follow the guidance in the GUM and in its supplements.

I am of the opinion that reviewers ought not to attempt to force authors to express only views that the reviewers share. It should be even less so in the case of a contribution like this one, of von Clarmann's *et al.*, which is more an opinion piece than a technical article, welcome nonetheless.

Below I offer assorted comments that do not add up to a thorough and complete review, which the authors and the editor may like to consider, and, in dialog with one another, possibly reach a settlement about next steps. I believe that the review I submitted of the original submission was quite thorough and detailed, and I appreciate the attention that the authors have obviously given to it.

I will abstain from making detailed comments about how the article is written in English, or offering specific suggestions for how it can be improved on this count. But I will make the following general remarks: the facts and arguments the authors present can, by and large, generally be understood by anyone with a modicum of familiarity with the relevant subjects; however, the article is far from being written in good scholarly English, the punctuation being particularly defective, which occasionally actually hinders understanding.

**2   Comments**

**2.1** L078 *The error has been conceived as an attribute of a measurement or an estimate, while the term 'uncertainty' has been used as an attribute of the true state*

> I would say that the uncertainty is an attribute of the relationship between the person making the measurement and the true value of the measurand, characterizing the incomplete knowledge that person is left with following the measurement.

**2.2** L104 *The estimated error is understood as a measure of the width of a distribution around the measured value which tells the data user the probability density of a certain value to be measured if the value actually measured was the true value.*

> The simplest measurement model expresses the measured value as $x = \mu + \varepsilon$, where $\mu$ denotes the true value of the measurand and $\varepsilon$ denotes measurement error. The error can be positive, negative, or null, and it is modeled as a (non-observable) random variable. The spread of the corresponding probability distribution, gauged using the standard deviation or any other similar metric, is indicative of the typical magnitude of the error, but it is not the error itself.

**2.3** L244 *In GUM the error concept is discarded because the capability of conducting an error estimate allegedly depends on the knowledge of the true value.*

> The theory of statistical estimation [Lehmann and Casella, 1998] has never shied away from entertaining quantities whose values are unknown, be they parameters of interest, or non-observable errors. There are statistical procedures that provide estimates of individual errors, not just summary estimates of their typical magnitude: for example, cross-validation [Stone, 1978] of a (linear or nonlinear) regression model.

**2.4** L419 *Some interpretations of GUM-2008 [...] seem to suggest that error estimation and uncertainty analysis are best distinguished in the sense that the former relies on frequentist statistics while the latter is founded on Bayesian statistics.*

The authors are quite right when they dismiss this myth, variously and repeatedly throughout the article.

**2.5** L462 *The uncertainty concept relies on the possibility of evaluating uncertainties caused by measurement errors and "systematic effects" without knowledge of the true value. This is certainly granted for linear problems. Here the uncertainty estimates do not depend on the value of the measurand. This is because in the linear case Gaussian error propagation holds.*

I do not understand what the authors mean when they say "This is certainly granted for linear problems." And regarding the third sentence: there are nonlinear models where the typical size of the errors does not depend on the value of the measurand, and there are linear models where it does. Finally, what is "Gaussian error propagation?"

**2.6** L504 *The situation is more difficult for biases. Biases between different measurement systems do not tell us what the bias of one measurement system with respect to the — unfortunately unknowable — truth is. Even if the number of measurement systems is quite large, it is not guaranteed that the mean bias of all of them is zero.*

Bias (differently from how the VIM [Joint Committee for Guides in Metrology, 2012] defines it) is the difference between the mathematical expectation of an estimator and the quantity being estimated.

The situation is not as desperate as the authors paint it: the statistical jacknife [Mosteller and Tukey, 1977] [Efron, 1982], the statistical bootstrap [Efron and Tibshirani, 1993], and cross-validation [Mosteller and Tukey, 1977] can provide useful estimates of bias without assuming that the target of estimation is known.

As the authors note around L513, by intercomparing independent measurement results for the same measurand, one can ascertain at least whether all important sources of uncertainty have been accounted for in the respective uncertainty budgets or not.

If there should be lurking uncertainty components that manifest themselves only when measurement results (measured values and their uncertainties) are compared — so-called *dark uncertainty* [Thompson and Ellison, 2011] —, then this may be attributed to yet undetected and still unexplained effects, which can be persistent — that is, biases —, or volatile.

The so-called *Hubble Tension* illustrates such situation, in relation with which Riess et al. [2019] makes the following remark: "pinpointing the cause of the tension requires further improvement in the local measurements, with continued focus on precision, accuracy, and experimental design to control systematics."

**References**

B. Efron. *The jackknife, the bootstrap, and other resampling plans*, volume 38 of *CBMS-NSF Regional Conferences in Applied Mathematics*. Society for Industrial and Applied Mathematics (SIAM), Philadelphia, PA, 1982. ISBN 0-89871-179-7.

B. Efron and R. J. Tibshirani. *An Introduction to the Bootstrap*. Chapman & Hall, London, UK, 1993.

Joint Committee for Guides in Metrology. *International vocabulary of metrology — Basic and general concepts and associated terms (VIM)*. International Bureau of Weights and Measures (BIPM), Sèvres, France, 3rd edition, 2012. URL `https://jcgm.bipm.org/vim/en/`. BIPM, IEC, IFCC, ILAC, ISO, IUPAC, IUPAP and OIML, JCGM 200:2012 (2017 version with minor corrections and informative annotations).

E. L. Lehmann and G. Casella. *Theory of Point Estimation*. Springer Texts in Statistics. Springer-Verlag, New York, NY, 2nd edition, 1998. ISBN 978-1-4419-3130-6.

F. Mosteller and J. W. Tukey. *Data Analysis and Regression*. Addison-Wesley Publishing Company, Reading, Massachusetts, 1977. ISBN 0-201-04854-X.

A. G. Riess, S. Casertano, W. Yuan, L. M. Macri, and D. Scolnic. Large Magellanic Cloud Cepheid Standards provide a 1% foundation for the determination of the Hubble Constant and stronger evidence for physics beyond ΛCDM. *The Astrophysical Journal*, 876(1):85, May 2019. doi: 10.3847/1538-4357/ab1422.

M. Stone. Cross-validation:a review. *Series Statistics*, 9(1):127–139, 1978. doi: 10.1080/02331887808801414.

M. Thompson and S. L. R. Ellison. Dark uncertainty. *Accreditation and Quality Assurance*, 16:483–487, October 2011. doi: 10.1007/s00769-011-0803-0.